# Transformers are Minimax Optimal Nonparametric In-Context Learners

**Juno Kim[1,2]***    **Tai Nakamaki[1]**    **Taiji Suzuki[1,2]**
[1]University of Tokyo    [2]Center for Advanced Intelligence Project, RIKEN
*junokim@g.ecc.u-tokyo.ac.jp

## Abstract

In-context learning (ICL) of large language models has proven to be a surprisingly effective method of learning a new task from only a few demonstrative examples. In this paper, we study the efficacy of ICL from the viewpoint of statistical learning theory. We develop approximation and generalization error bounds for a transformer composed of a deep neural network and one linear attention layer, pretrained on nonparametric regression tasks sampled from general function spaces including the Besov space and piecewise $\gamma$-smooth class. We show that sufficiently trained transformers can achieve – and even improve upon – the minimax optimal estimation risk in context by encoding the most relevant basis representations during pretraining. Our analysis extends to high-dimensional or sequential data and distinguishes the *pretraining* and *in-context* generalization gaps. Furthermore, we establish information-theoretic lower bounds for meta-learners w.r.t. both the number of tasks and in-context examples. These findings shed light on the roles of task diversity and representation learning for ICL.

## 1 Introduction

Large language models (LLMs) have demonstrated remarkable capabilities in understanding and generating natural language data. In particular, the phenomenon of *in-context learning* (ICL) has recently garnered widespread attention. ICL refers to the ability of pretrained LLMs to perform a new task by being provided with a few examples within the context of a prompt, without any parameter updates or fine-tuning. It has been empirically observed that few-shot prompting is especially effective in large-scale models (Brown et al., 2020) and requires only a couple of examples to consistently achieve high performance (García et al., 2023). In contrast, Raventos et al. (2023) demonstrate that sufficient pretraining task diversity is required for the emergence of ICL. However, we still lack a comprehensive understanding of the statistical foundations of ICL and few-shot prompting.

A vigorous line of research has been directed towards understanding ICL of single-layer linear attention models pretrained on the query prediction loss of linear regression tasks (Garg et al., 2022; Akyürek et al., 2023; Zhang et al., 2023; Ahn et al., 2023; Mahankali et al., 2023; Wu et al., 2024). It has been shown that the global minimizer of the $L^2$ pretraining loss implements one step of GD on a least-squares linear regression objective (Mahankali et al., 2023) and is nearly Bayes optimal (Wu et al., 2024). Moreover, risk bounds with respect to the context length (Zhang et al., 2023) and number of tasks (Wu et al., 2024) have been obtained.

Other works have examined ICL of more complex multi-layer transformers. Bai et al. (2023); von Oswald et al. (2023) give specific transformer constructions which simulate GD in context, however it is unclear how such meta-algorithms may be learned. Another approach is to study learning *with representations*, where tasks consist of a fixed nonlinear feature map composed with a varying linear function. Guo et al. (2023) empirically found that trained transformers exhibit a separation where lower layers transform the input and upper layers perform linear ICL. Recently, Kim and Suzuki

([2024](#)) analyzed a model consisting of a shallow neural network followed by a linear attention layer and proved that the MLP component learns to encode the true features during pretraining. However, they assumed the infinite task and sample size limit and did not study generalization capabilities.

**Our contributions.**   In this paper, we analyze the optimality of ICL from the perspective of statistical learning theory. Our object of study is a transformer consisting of a deep neural network with $N$-dimensional output followed by one linear attention layer. The model is pretrained on $n$ input-output samples from $T$ nonparametric regression tasks, generated from a suitably decaying distribution on a general function space. Compared to previous works, we take a crucial step towards understanding practical multi-layer transformers by incorporating the representation learning capabilities of the DNN module. From a more abstract perspective, this work can also be situated as a nonlinear extension of meta-learning. Our contributions are highlighted below.

- We develop a general framework for upper bounding the in-context estimation error of the empirical risk minimizer in terms of the approximation error of the neural network and separate *in-context* and *pretraining* generalization gaps depending on $n, T$, respectively.

- In the Besov space setting, we show that **ICL achieves nearly minimax optimal risk** $n^{-\frac{2\alpha}{2\alpha+d}}$ when $T$ is sufficiently large. Since LLMs are pretrained on vast amounts of data in practice, $T$ can be taken to be nearly infinite, justifying the emergence of ICL at large scales. We extend the optimality guarantees to nearly dimension-free rates in the anisotropic Besov space and also to learning sequential data with deep transformers in the piecewise $\gamma$-smooth function class.

- We show that ICL can **improve upon the a priori optimal rate** when the task class basis resides in a coarser Besov space by learning to encode informative basis representations, emphasizing the importance of pretraining on diverse tasks.

- We also derive **information-theoretic lower bounds** for the minimax risk in both $n, T$, rigorously confirming that ICL is jointly optimal when $T$ is large (Besov space setting), while any meta-learning method is jointly suboptimal when $T$ is small (coarser space setting). This separation aligns with empirical observations of a *task diversity threshold* ([Raventos et al.](#), [2023](#)).

The paper is structured as follows. In Section [2](#), the regression tasks and transformer model are defined in an abstract setting. In Section [3](#), we present the general framework for estimating the ICL approximation and generalization error. In Section [4](#), we specialize to the Besov-type and piecewise $\gamma$-smooth class settings and show that transformers can achieve or exceed the minimax optimal rate in context. In Section [5](#), we derive minimax lower bounds. All proofs are deferred to the appendix; moreover, we provide **numerical experiments** validating our results in Appendix [E](#).

## 1.1   Other Related Works

**Meta-learning.**   The theoretical setting of ICL is closely related to *meta-learning*, where the goal is to infer a shared representation $\psi^\circ$ with samples from a set of transformed tasks $\beta_i^\top \psi^\circ$. When $\psi^\circ$ is linear, fast rates have been established by [Tripuraneni et al.](#) ([2020](#)); [Du et al.](#) ([2021](#)), while the nonlinear case has been studied by [Meunier et al.](#) ([2023](#)) where $\psi^\circ$ is a feature projection into a reproducing kernel Hilbert space. Our results can be viewed as extending this body of work to function spaces of generalized smoothness with a specific deep transformer architecture.

**Optimal rates for DNNs.**   Our analysis extends established optimality results for classes of DNNs in ordinary supervised regression settings to ICL. [Suzuki](#) ([2019](#)) has shown that deep feedforward networks with the ReLU activation can efficiently approximate functions in the Besov space and thus achieve the minimax optimal rate. This has been extended to the anisotropic Besov space ([Suzuki and Nitanda](#), [2021](#)), convolutional neural networks for infinite-dimensional input ([Okumoto and Suzuki](#), [2022](#)), and transformers for sequence-to-sequence functions ([Takakura and Suzuki](#), [2023](#)). We remark that a work in progress ([Imaizumi](#), [2024](#)) also studies the sample complexity of ICL of transformers in a Sobolev space setting.

**Pretraining dynamics for ICL.**   While how ICL arises from optimization is not fully understood, there are encouraging developments in this direction. [Zhang et al.](#) ([2023](#)) has shown for one layer of linear attention that running GD on the population risk always converges to the global optimum. This was extended to incorporate a linear output layer by [Zhang et al.](#) ([2024](#)), and to softmax attention

by Huang et al. (2023); Li et al. (2024); Chen et al. (2024). Kim and Suzuki (2024) considered a compound transformer equivalent to ours with a shallow MLP component and proved that the loss landscape becomes benign in the mean-field limit, deriving convergence guarantees for the corresponding gradient dynamics. These analyses indicate that the attention mechanism, while highly nonconvex, may possess structures favorable for gradient-based optimization.

## 2 Problem Setup

### 2.1 Nonparametric Regression

In this paper, we analyze the ability of a transformer to solve nonparametric regression problems in context when pretrained on examples from a *family* of regression tasks, which we describe below. Let $\mathcal{X} \subseteq \mathbb{R}^d$ be the input space ($d$ is allowed to be infinite), $\mathcal{P}_{\mathcal{X}}$ a probability distribution on $\mathcal{X}$, and $(\psi_j^\circ)_{j=1}^\infty$ a fixed countable subset of $L^2(\mathcal{P}_{\mathcal{X}})$. A regression function $F_\beta^\circ : \mathcal{X} \to \mathbb{R}$ is randomly generated for each task by sampling the sequence of coefficients $\beta \in \mathbb{R}^\infty$ from a distribution $\mathcal{P}_\beta$ on $\mathscr{B}(\mathbb{R}^\infty)$; the class of tasks $\mathcal{F}^\circ$ is defined as

$$\mathcal{F}^\circ = \left\{ F_\beta^\circ = \sum_{j=1}^\infty \beta_j \psi_j^\circ \,\middle|\, \beta \in \operatorname{supp} \mathcal{P}_\beta \right\},$$

endowed with the induced distribution. Each task prompt contains $n$ example input-response pairs $\{(x_k, y_k)\}_{k=1}^n$. The covariates $x_k$ are i.i.d. drawn from $\mathcal{P}_{\mathcal{X}}$ and the responses are generated as

$$y_k = F_\beta^\circ(x_k) + \xi_k, \quad 1 \le k \le n, \tag{1}$$

where the noise $\xi_k$ is assumed to be i.i.d. with mean zero and $|\xi_k| \le \sigma$ almost surely.[1] In addition, we independently generate a query token $\tilde{x}$ and corresponding output $\tilde{y}$ in the same manner.

We proceed to state our assumptions for the regression model. Informally, we suppose a relaxed version of sparsity and orthonormality of $\psi_j^\circ$ and suitable decay rates for the basis expansion. These will be subsequently verified for specific function spaces of interest with their natural decay rate.

**Assumption 1** (relaxed sparsity and orthonormality of basis functions). *For $N \in \mathbb{N}$, there exist integers $\underline{N} < \bar{N} \lesssim N$ with $\bar{N} - \underline{N} + 1 = N$ such that $\psi_{\underline{N}}^\circ, \cdots, \psi_{\bar{N}}^\circ$ are independent and $\psi_1^\circ, \cdots, \psi_{\underline{N}-1}^\circ$ are all contained in the linear span of $\psi_{\underline{N}}^\circ, \cdots, \psi_{\bar{N}}^\circ$. Moreover, there exist $r, C_1, C_2, C_\infty > 0$ such that $\Sigma_{\Psi,N} := \left( \mathbb{E}_{x \sim \mathcal{P}_{\mathcal{X}}}[\psi_j^\circ(x)\psi_k^\circ(x)] \right)_{j,k=\underline{N}}^{\bar{N}}$ satisfies $C_1 \mathbf{I}_N \preceq \Sigma_{\Psi,N} \preceq C_2 \mathbf{I}_N$ and*

$$\left\| \sum_{j=\underline{N}}^{\bar{N}} (\psi_j^\circ)^2 \right\|_{L^\infty(\mathcal{P}_{\mathcal{X}})}^{1/2} \le C_\infty N^r. \tag{2}$$

Denoting the $\bar{N}$-basis approximation of $F_\beta^\circ$ as $F_{\beta,\bar{N}}^\circ := \sum_{j=1}^{\bar{N}} \beta_j \psi_j^\circ$, by Assumption 1 there exist 'aggregated' coefficients $\bar{\beta}_{\underline{N}}, \cdots, \bar{\beta}_{\bar{N}}$ uniquely determined by $\beta$ such that $F_{\beta,\bar{N}}^\circ = \sum_{j=\underline{N}}^{\bar{N}} \bar{\beta}_j \psi_j^\circ$. We define two types of coefficient covariance matrices

$$\Sigma_{\beta,\bar{N}} := \left( \mathbb{E}_\beta[\beta_j \beta_k] \right)_{j,k=1}^{\bar{N}} \in \mathbb{R}^{\bar{N} \times \bar{N}} \quad \text{and} \quad \Sigma_{\bar{\beta},N} := \left( \mathbb{E}_\beta[\bar{\beta}_j \bar{\beta}_k] \right)_{j,k=\underline{N}}^{\bar{N}} \in \mathbb{R}^{N \times N}.$$

**Assumption 2** (decay of $\beta$). *For $s, B > 0$ it holds that $\|F_\beta^\circ\|_{L^\infty(\mathcal{P}_{\mathcal{X}})} \le B$ for all $F_\beta^\circ \in \mathcal{F}^\circ$ and*

$$\|F_\beta^\circ - F_{\beta,N}^\circ\|_{L^2(\mathcal{P}_{\mathcal{X}})}^2 \lesssim N^{-2s} \tag{3}$$

*uniformly over $\beta \in \operatorname{supp} \mathcal{P}_\beta$. Furthermore, $\operatorname{Tr}(\Sigma_{\bar{\beta},N})$ is bounded for all $N$ and*

$$0 \prec \Sigma_{\beta,\bar{N}} \precsim \operatorname{diag} \left[ (j^{-2s-1}(\log j)^{-2})_{j=1}^{\bar{N}} \right]. \tag{4}$$

**Remark 2.1.** In the simple case where $(\psi_j^\circ)_{j=1}^\infty$ is a basis for $L^2(\mathcal{P}_{\mathcal{X}})$, we may set $\underline{N} = 1, \bar{N} = N$ so that the dependency condition of Assumption 1 is trivially satisfied, moreover, $\Sigma_{\bar{\beta},N} = \Sigma_{\beta,N}$ and boundedness of $\operatorname{Tr}(\Sigma_{\bar{\beta},N})$ automatically follows from (4). However, the assumptions in the stated form also allow for hierarchical bases with dependencies such as wavelet systems. We also note that (3) and (4) entail basically the same rate but are not equivalent: the *uniform* bound $|\beta_j|^2 \lesssim j^{-2s-1}(\log j)^{-2}$ along with Assumption 1 implies (3). The $(\log j)^{-2}$ term can be replaced with any $g(j)$ such that $\sum_{j=1}^\infty j^{-1} g(j)$ is convergent.

---

[1] This implies $\operatorname{Var} \xi_k \le \sigma^2$. We require boundedness since the values $y_k$ form part of the prompt input, and we wish to utilize sup-norm covering number estimates for the attention map; this technicality can be removed with more careful analysis. However, for the information-theoretic lower bound we assume Gaussian noise.

## 2.2 In-Context Learning

We now describe our transformer model, which takes $n$ context pairs $\mathbf{X} = (x_1, \cdots, x_n) \in \mathbb{R}^{d \times n}$, $\boldsymbol{y} = (y_1, \cdots, y_n)^\top \in \mathbb{R}^n$ and a query token $\tilde{x}$ as input and returns a prediction for the corresponding output. The covariates are first passed through a nonlinear representation or feature mapping $\phi : \mathcal{X} \to \mathbb{R}^N$, which we assume belongs to a sufficiently powerful class of estimators $\mathcal{F}_N$. Specifically:

**Assumption 3** (expressivity of $\mathcal{F}_N$). $\|\phi(x)\|_2 \leq B_N'$ for some $B_N' > 0$ for all $x \in \mathcal{X}$, $\phi \in \mathcal{F}_N$. Moreover for some $\delta_N > 0$, there exist $\phi_{\underline{N}}^*, \cdots, \phi_{\overline{N}}^* \in \mathcal{F}_N$ satisfying

$$\max_{\underline{N} \leq j \leq \overline{N}} \|\psi_j^\circ - \phi_j^*\|_{L^\infty(\mathcal{P}_\mathcal{X})} \leq \delta_N.$$

By choosing $\mathcal{F}_N$ and $\delta_N$ to satisfy the above assumption, we will be able to utilize established approximation and generalization guarantees for families of deep neural networks in Section 4.

The extracted representations $\phi(\mathbf{X}) = (\phi(x_1), \cdots, \phi(x_n))$ are then mapped to a scalar output via a linear attention layer parametrized by a matrix $\Gamma \in \mathcal{S}_N$ for $\mathcal{S}_N \subset \mathbb{R}^{N \times N}$,

$$\check{f}_\Theta(\mathbf{X}, \boldsymbol{y}, \tilde{x}) := \frac{1}{n} \sum_{k=1}^n y_k \phi(x_k)^\top \Gamma^\top \phi(\tilde{x}) = \left\langle \frac{\Gamma \phi(\mathbf{X}) \boldsymbol{y}}{n}, \phi(\tilde{x}) \right\rangle, \quad \text{where } \Theta = (\Gamma, \phi) \in \mathcal{S}_N \times \mathcal{F}_N.$$

Finally, the output is constrained to lie on $[-\bar{B}, \bar{B}]$ by applying $\text{clip}_{\bar{B}}(u) := \max\{\min\{u, \bar{B}\}, -\bar{B}\}$, yielding $f_\Theta(\mathbf{X}, \boldsymbol{y}, \tilde{x}) := \text{clip}_{\bar{B}}(\check{f}_\Theta(\mathbf{X}, \boldsymbol{y}, \tilde{x}))$. We set $\mathcal{S}_N = \{\Gamma \in \mathbb{R}^{N \times N} \mid 0 \preceq \Gamma \preceq C_3 \mathbf{I}_N\}$ for some $C_3 > 0$ and fix $\bar{B} = B$ for simplicity.

The above setup is a restricted reparametrization of linear attention widely used in theoretical analyses (see e.g. Zhang et al., 2023; Wu et al., 2024, for more details), where the values only refer to $\boldsymbol{y}$ and the query and key matrices are consolidated into one matrix $\Gamma$. The form is equivalent to one step of GD with matrix step size and has been shown to be optimal for a single layer of linear attention for linear regression tasks (Ahn et al., 2023; Mahankali et al., 2023). The placement of the attention layer after the DNN module $\phi$ is justified by the observation that lower layers of trained transformers act as data representations on top of which upper layers perform ICL (Guo et al., 2023).

During pretraining, the model is presented with $T$ prompts $\{(\mathbf{X}^{(t)}, \boldsymbol{y}^{(t)}, \tilde{x}^{(t)})\}_{t=1}^T$ where the tasks $F_{\beta^{(t)}}^\circ \in \mathcal{F}^\circ$, $\beta^{(t)} \sim \mathcal{P}_\beta$ and tokens $\mathbf{X}^{(t)} = (x_1^{(t)}, \cdots, x_n^{(t)})$, $\boldsymbol{y}^{(t)} = (y_1^{(t)}, \cdots, y_n^{(t)})^\top$, $\tilde{x}^{(t)}$ and $\tilde{y}^{(t)}$ are independently generated as described in Section 2.1, and is trained to minimize the empirical risk

$$\widehat{\Theta} = \underset{\Theta \in \mathcal{S}_N \times \mathcal{F}_N}{\arg\min} \widehat{R}(\Theta), \quad \widehat{R}(\Theta) = \frac{1}{T} \sum_{t=1}^T \left( \tilde{y}^{(t)} - f_\Theta(\mathbf{X}^{(t)}, \boldsymbol{y}^{(t)}, \tilde{x}^{(t)}) \right)^2.$$

Our goal is to verify the efficiency of ICL as a learning algorithm and show that learning the optimal $\widehat{\Theta}$ allows the transformer to solve new random regression problems $y = F_\beta^\circ(x) + \xi$ for $F_\beta^\circ \in \mathcal{F}^\circ$ in context. To this end, we evaluate the convergence of the mean-squared risk or estimation error,

$$\bar{R}(\widehat{\Theta}) := \mathbb{E}_{(\mathbf{X}^{(t)}, \boldsymbol{y}^{(t)}, \tilde{x}^{(t)}, \tilde{y}^{(t)})_{t=1}^T}[R(\widehat{\Theta})], \quad R(\Theta) := \mathbb{E}_{\mathbf{X}, \boldsymbol{y}, \tilde{x}, \beta} \left[ (F_\beta^\circ(\tilde{x}) - f_\Theta(\mathbf{X}, \boldsymbol{y}, \tilde{x}))^2 \right].$$

Note that we do not study whether the transformer always converges to $\widehat{\Theta}$; the training dynamics of a DNN is already a very difficult problem. For the attention layer, see the discussion in Section 1.1.

## 3 Risk Bounds for In-Context Learning

In this section, we outline our framework for analyzing the in-context estimation error $\bar{R}(\widehat{\Theta})$. Some additional definitions are in order. The $\epsilon$-covering number $\mathcal{N}(\mathcal{C}, \rho, \epsilon)$ of a metric space $\mathcal{C}$ equipped with a metric $\rho$ for $\epsilon > 0$ is defined as the minimal number of balls in $\rho$ with radius $\epsilon$ needed to cover $\mathcal{C}$ (van der Vaart and Wellner, 1996). The $\epsilon$-*covering entropy* or *metric entropy* is given as $\mathcal{V}(\mathcal{F}, \rho, \epsilon) := \log \mathcal{N}(\mathcal{F}, \rho, \epsilon)$. The $\epsilon$-*packing number* $\mathcal{M}(\epsilon, \mathcal{C}, \rho)$ is given as the maximal cardinality of a $\epsilon$-separated set $\{c_1, \ldots, c_M\} \subseteq \mathcal{C}$ such that $\rho(c_i, c_j) \geq \delta$ for all $i \neq j$. The transformer model class is defined as $\mathcal{T}_N := \{f_\Theta \mid \Theta \in \mathcal{S}_N \times \mathcal{F}_N\}$.

To bound the overall risk, we first decompose into the approximation and generalization gaps.

**Theorem 3.1** (Schmidt-Hieber (2020), Lemma 4, adapted). *There exists a universal constant $C$ such that for any $\epsilon > 0$ such that $\mathcal{V}(\mathcal{T}_N, \|\cdot\|_{L^\infty}, \epsilon) \geq 1$,*

$$\bar{R}(\widehat{\Theta}) \leq 2 \inf_{\Theta \in \mathcal{S}_N \times \mathcal{F}_N} R(\Theta) + C\left(\frac{B^2 + \sigma^2}{T}\mathcal{V}(\mathcal{T}_N, \|\cdot\|_{L^\infty}, \epsilon) + (B + \sigma)\epsilon\right).$$

*Proof.* The convergence rate of the empirical risk minimizer is established for a fixed regression problem $y = f^\circ(z) + \xi$ in Schmidt-Hieber (2020) when $\xi$ is Gaussian; we modify the proof to incorporate bounded noise in Appendix B.1. The ICL setup can be reduced to the ordinary case as follows. We consider the entire batch $(\beta, \mathbf{X}, \xi_{1:n}, \tilde{x})$ including the hidden coefficient $\beta$ as a single datum $z$ with output $\tilde{y}$. The true function is given as $f^\circ(z) = F_\beta^\circ(\tilde{x})$ and the model class is taken to be $\mathcal{T}_N$ implicitly concatenated with the generative process $(\beta, \mathbf{X}, \xi_{1:n}, \tilde{x}) \mapsto (\mathbf{X}, \boldsymbol{y}, \tilde{x})$. Then $R(\Theta)$, $\mathcal{V}(\mathcal{T}_N, \|\cdot\|_{L^\infty}, \epsilon)$ and $T$ agree with the ordinary $L^2$ risk, model class entropy and sample size. $\square$

Here, the second term is the *pretraining* generalization error dependent on the number of tasks $T$; the *in-context* generalization error dependent on the prompt length $n$ manifests as part of the first term. This separation allows us to compare the relative difficulty of the two types of learning.

**Bounding approximation error.**   In order to bound the first term, we analyze the risk of the choice

$$\Theta^* = (\Gamma^*, \phi^*) := \left(\left(\Sigma_{\Psi,N} + \tfrac{1}{n}\Sigma_{\bar{\beta},N}^{-1}\right)^{-1}, \phi_{\underline{N}:\bar{N}}^*\right)$$

where $\phi^*$ is given as in Assumption 3 for a suitable $\delta_N$ to be determined. The definition of $\Gamma^*$ approximately generalizes the global optimum $\Gamma = \left((1 + \tfrac{1}{n})\Lambda + \tfrac{1}{n}\text{tr}(\Lambda)\mathbf{I}_d\right)^{-1}$ for the Gaussian linear regression setup where $x \sim \mathcal{N}(0, \Lambda)$ (Zhang et al., 2023). Since $\Sigma_{\Psi,N} \succeq C_1\mathbf{I}_N$ we have $\Gamma^* \preceq C_1^{-1}\mathbf{I}_N$ and hence we may assume $\Gamma^* \in \mathcal{S}_N$ by replacing $C_3$ with $C_3 \vee C_1^{-1}$ if necessary.

**Proposition 3.2.** *Under Assumptions 1-3, it holds that*

$$\inf_{\Theta \in \mathcal{S}_N \times \mathcal{F}_N} R(\Theta) \leq R(\Theta^*) \lesssim \frac{N^{2r}}{n}\log N + \frac{N^{4r}}{n^2}\log^2 N + \frac{N}{n} + N^{-2s} + N^2\delta_N^4 + N^{2r+1}\delta_N^2.$$

The proof is presented throughout Appendix A. The overall scheme is to approximate $F_\beta^\circ(\tilde{x})$ by its truncation $F_{\beta,\bar{N}}^\circ(\tilde{x})$ and the finite basis $\psi_{\underline{N}:\bar{N}}^\circ$ by $\phi_{\underline{N}:\bar{N}}^*$, which incurs errors $N^{-2s}$ and the terms pertaining to $\delta_N$, respectively. The first three terms arise from the concentration of the $n$ token representations $\psi_{\underline{N}:\bar{N}}^\circ(x_k)$. All hidden constants are at most polynomial in problem parameters.

**Bounding generalization error.**   To estimate the metric entropy of $\mathcal{T}_N$, we first reduce to the metric of the representation class $\mathcal{F}_N$. Here, $\|\cdot\|_{L^\infty}$ refers to the essential supremum over the support of $\mathcal{P}_\mathcal{X}$ and also over all $N$ components for $\mathcal{F}_N$. The proof is given in Appendix B.2.

**Lemma 3.3.** *Under Assumptions 1-3, there exists $D > 0$ such that for all $\epsilon$ sufficiently small,*

$$\mathcal{V}(\mathcal{T}_N, \|\cdot\|_{L^\infty}, \epsilon) \lesssim N^2 \log\frac{B_N'^2}{\epsilon} + \mathcal{V}\left(\mathcal{F}_N, \|\cdot\|_{L^\infty}, \frac{\epsilon}{DB_N'\sqrt{N}}\right).$$

## 4   Minimax Optimality of In-Context Learning

### 4.1   Besov Space and DNNs

We now apply our theory to study the sample complexity of ICL when $\mathcal{F}_N$ consists of (clipped, see (6)) deep neural networks. These can be also seen as simplified transformers with attention layers and skip connections removed. To be precise, we define the set of DNNs with depth $L$, width $W$, sparsity $S$, norm bound $M$ and ReLU activation $\eta(x) = x \vee 0$ (applied element-wise) as

$$\mathcal{F}_{\text{DNN}}(L, W, S, M) = \Big\{(\mathbf{W}^{(L)}\eta + b^{(L)}) \circ \cdots \circ (\mathbf{W}^{(1)}x + b^{(1)})\Big| \mathbf{W}^{(1)} \in \mathbb{R}^{W \times d}, \mathbf{W}^{(\ell)} \in \mathbb{R}^{W \times W},$$

$$\mathbf{W}^{(L)} \in \mathbb{R}^W, b^{(\ell)} \in \mathbb{R}^W, b^{(L)} \in \mathbb{R}, \sum_{\ell=1}^{L}\|\mathbf{W}^{(\ell)}\|_0 + \|b^{(\ell)}\|_0 \leq S, \max_{1 \leq \ell \leq L}\|\mathbf{W}^{(\ell)}\|_\infty \vee \|b^{(\ell)}\|_\infty \leq M\Big\}.$$

The *Besov space* is a very general class of functions including the Hölder and Sobolev spaces which captures spatial inhomogeneity in smoothness, and provides a natural setting in which to study the expressive power of deep neural networks (Suzuki, 2019). Here, we fix $\mathcal{X} = [0,1]^d$ for simplicity.

**Definition 4.1** (Besov space). For $2 \leq p \leq \infty, 0 < q \leq \infty$, fractional smoothness $\alpha > 0$ and $r = \lfloor \alpha \rfloor + 1$, the $r$th modulus of $f \in L^p(\mathcal{X})$ is defined using the difference operator $\Delta_h^r, h \in \mathbb{R}^d$ as

$$w_{r,p}(f,t) := \sup_{\|h\|_2 \leq t} \|\Delta_h^r(f)\|_p, \quad \Delta_h^r(f)(x) = 1_{\{x,x+rh \in \mathcal{X}\}} \sum_{j=0}^r \binom{r}{j} (-1)^{r-j} f(x+jh).$$

Also, the Besov (quasi-)norm is given as $\|\cdot\|_{B_{p,q}^\alpha} = \|\cdot\|_{L^p} + |\cdot|_{B_{p,q}^\alpha}$ where

$$|f|_{B_{p,q}^\alpha} := \begin{cases} \left( \int_0^\infty t^{-q\alpha} w_{r,p}(f,t)^q \frac{dt}{t} \right)^{1/q} & q < \infty \\ \sup_{t>0} t^{-\alpha} w_{r,p}(f,t) & q = \infty \end{cases}$$

and the Besov space is defined as $B_{p,q}^\alpha(\mathcal{X}) = \{f \in L^p(\mathcal{X}) \mid \|f\|_{B_{p,q}^\alpha} < \infty\}$. We write $\mathbb{U}(B_{p,q}^\alpha(\mathcal{X}))$ for the unit ball in $(B_{p,q}^\alpha(\mathcal{X}), \|\cdot\|_{B_{p,q}^\alpha})$.

We have that the Hölder space $C^\alpha(\mathcal{X}) = B_{\infty,\infty}^\alpha(\mathcal{X})$ for order $\alpha > 0, \alpha \notin \mathbb{N}$ and the Sobolev space $W_2^m(\mathcal{X}) = B_{2,2}^m(\mathcal{X})$ for $m \in \mathbb{N}$ as well as the embeddings $B_{p,1}^m(\mathcal{X}) \hookrightarrow W_p^m(\mathcal{X}) \hookrightarrow B_{p,\infty}^m(\mathcal{X})$; if $\alpha > d/p$, $B_{p,q}^\alpha(\mathcal{X})$ compactly embeds into the space of continuous functions on $\mathcal{X}$. See Triebel (1983); Giné and Nickl (2015) for more details. The difficulty of learning a regression function in the Besov is quantified by the minimax risk; the following rate is classical.

**Proposition 4.2** (Donoho and Johnstone (1998)). *The minimax risk for an estimator $\widehat{f}_n$ with $n$ i.i.d. samples $\mathcal{D}_n = \{(x_i, y_i)\}_{i=1}^n$ over $\mathbb{U}(B_{p,q}^\alpha(\mathcal{X}))$ satisfies*

$$\inf_{\widehat{f}_n:\mathcal{D}_n \to \mathbb{R}} \sup_{f^\circ \in \mathbb{U}(B_{p,q}^\alpha(\mathcal{X}))} \mathbb{E}_{\mathcal{D}_n}[\|f^\circ - \widehat{f}_n\|_{L^2(\mathcal{X})}^2] \asymp n^{-\frac{2\alpha}{2\alpha+d}}.$$

A natural basis system for $B_{p,q}^\alpha(\mathcal{X})$ is formed by the *B-splines*, which can be seen as a type of wavelet decomposition or multiresolution analysis (DeVore and Popov, 1988). As B-splines are piecewise polynomials, they can be efficiently approximated by DNNs with at most log depth (Suzuki, 2019).

**Definition 4.3** (B-spline wavelet basis). The tensor product B-spline of order $m \in \mathbb{N}$ satisfying $m > \alpha + 1 - 1/p$, at resolution $k \in \mathbb{Z}_{\geq 0}^d$ and location $\ell \in I_k^d = \prod_{i=1}^d [-m : 2^{k_i}]$ is

$$\omega_{k,\ell}^d(x) = \prod_{i=1}^d \iota_m(2^{k_i} x_i - \ell_i), \quad \text{where} \quad \iota_m(x) = (\underbrace{\iota * \iota * \cdots * \iota}_{m+1})(x), \quad \iota(x) = 1_{\{x \in [0,1]\}}.$$

When $k_1 = \cdots = k_d$, we abuse notation and write $\omega_{k,\ell}^d$ for $k \in \mathbb{Z}_{\geq 0}$ in place of $\omega_{(k,\cdots,k),\ell}^d$.

## 4.2 Estimation Error Analysis

To apply our framework, we set the task class as the unit ball $\mathcal{F}^\circ = \mathbb{U}(B_{p,q}^\alpha(\mathcal{X}))$ and take as basis $\{\psi_j^\circ \mid j \in \mathbb{N}\} = \{2^{kd/2} \omega_{k,\ell}^d \mid k \in \mathbb{Z}_{\geq 0}, \ell \in I_k^d\}$ the set of all B-spline wavelets ordered primarily by increasing $k$ and scaled to counteract the dilation in $x$. Abusing notation, we also write $\beta_{k,\ell}$ to denote the coefficient in $\beta$ corresponding to $2^{kd/2} \omega_{k,\ell}^d$. The set of B-splines at each resolution are independent, while those of lower resolution can always be decomposed into a linear sum of B-splines of higher resolution satisfying certain decay rates, which we prove in Proposition C.10.

From this setup, in Appendix C.1.1, we verify Assumptions 1 and 2 with $r = 1/2, s = \alpha/d$ under:

**Assumption 4.** $\mathcal{F}^\circ = \mathbb{U}(B_{p,q}^\alpha(\mathcal{X}))$, $\alpha > d/p$ and $\mathcal{P}_\mathcal{X}$ has positive Lebesgue density $\rho_\mathcal{X}$ bounded above and below on $\mathcal{X}$. Also, all coefficients $\beta_{k,\ell}$ are independent and

$$\mathbb{E}_\beta[\beta_{k,\ell}] = 0, \quad 0 < \mathbb{E}_\beta[\beta_{k,\ell}^2] \lesssim 2^{-k(2\alpha+d)} k^{-2}, \quad \forall k \geq 0, \ \ell \in I_k^d. \tag{5}$$

We can check that we have not given ourselves an easier learning problem with (5): the assumed variance decay rate is tight (up to the logarithmic factor $k^{-2}$) in the sense that any $f \in \mathbb{U}(B_{p,q}^\alpha(\mathcal{X}))$ can indeed be expanded into a sum of wavelets with the same coefficient decay when averaged over $\ell \in I_k^d$. See Lemma C.1 and the following discussion. We also obtain the following in-context approximation and entropy bounds in Appendix C.1.2.

**Lemma 4.4.** *For any $\delta_N > 0$, Assumption 3 is satisfied by taking*

$$\mathcal{F}_N = \{\Pi_{B'_N} \circ \phi \mid \phi = (\phi_j)_{j=1}^N, \phi_j \in \mathcal{F}_{\mathrm{DNN}}(L, W, S, M)\} \tag{6}$$

*where $\Pi_{B'_N}$ is the projection in $\mathbb{R}^N$ to the centered ball of radius $B'_N = O(\sqrt{N})$ and each $\phi_j$ is a ReLU network such that $L = O(\log N + \log \delta_N^{-1})$ and $W, S, M = O(1)$. Also, the metric entropy of $\mathcal{F}_N$ is bounded as $\mathcal{V}(\mathcal{F}_N, \|\cdot\|_{L^\infty}, \epsilon) \lesssim N \log \frac{N}{\delta_N \epsilon}$.*

Hence Assumptions 1-3 all follow from Assumption 4, and we conclude in Appendix C.1.3:

**Theorem 4.5** (minimax optimality of ICL in Besov space). *Under Assumption 4, if $n \gtrsim N \log N$,*

$$\bar{R}(\widehat{\Theta}) \lesssim N^{-\frac{2\alpha}{d}} \left( \begin{smallmatrix} \text{DNN} \\ \text{approximation} \\ \text{error} \end{smallmatrix} \right) + \frac{N \log N}{n} \left( \begin{smallmatrix} \text{in-context} \\ \text{generalization} \\ \text{error} \end{smallmatrix} \right) + \frac{N^2 \log N}{T} \left( \begin{smallmatrix} \text{pretraining} \\ \text{generalization} \\ \text{error} \end{smallmatrix} \right).$$

*Hence if $T \gtrsim n^{\frac{2\alpha+2d}{2\alpha+d}}$ and $N \asymp n^{\frac{d}{2\alpha+d}}$, in-context learning achieves the minimax optimal rate $n^{-\frac{2\alpha}{2\alpha+d}}$ up to a log factor.*

The first term arises from the $N$-term truncation and oracle approximation error of the DNN module, and is equal to the $N$-term optimal error (Dũng, 2011a). The second and third term each correspond to the in-context and pretraining generalization gap. With regard to $N$, we see that $n = \widetilde{\Omega}(N)$ is enough to learn the basis expansion in context, while $T = \widetilde{\Omega}(N^2)$ is necessary to learn the attention layer. However if $T/N = o(n)$, the third term dominates and the overall complexity scales suboptimally as $T^{-\frac{\alpha}{\alpha+d}}$, illustrating the importance of sufficient pretraining. This also aligns with the task diversity threshold observed by Raventos et al. (2023). Since the amount of training data for LLMs is practically infinite in practice, our result justifies the effectiveness of ICL at large scales with only a small number of in-context samples.

**A limitation of ICL.** In the regime $1 \leq p < 2$, the approximation error is strictly worse without an *adaptive* representation scheme and the resulting rate is suboptimal (see Remark C.3). While DNNs can adapt to task smoothness in supervised settings (Suzuki, 2019), ICL and any other meta-learning methods are fundamentally constrained to non-adaptive representations since they cannot update at inference time, and hence are bounded below by the best linear approximation rate or Kolmogorov width, which is strictly worse than the minimax optimal rate when $p < 2$. Indeed, for any $N$-dimensional subspace $\mathcal{L}_N \subset B_{p,q}^\alpha(\mathcal{X})$ it holds that (Vybíral, 2008)

$$\inf_{\mathcal{L}_N} \sup_{f^\circ \in \mathbb{U}(B_{p,q}^\alpha(\mathcal{X}))} \inf_{\ell_n \in \mathcal{L}_N} \|f^\circ - \ell_n\|_{L^2(\mathcal{P}_\mathcal{X})} \gtrsim N^{-\alpha/d + (1/p - 1/2)_+}.$$

**Remark 4.6.** The $N^2 \log N$ term in the pretraining generalization gap is due to the covering bound of the attention matrix $\Gamma$, while the entropy of the DNN class is only $N \log N$. Hence the task diversity requirement may be lessened to the latter by considering low-rank structure or approximation of attention heads (Bhojanapalli et al., 2020; Chen et al., 2021).

### 4.3 Avoiding the Curse of Dimensionality

The above rate inevitably suffers from the curse of dimensionality as $d$ appears in the exponent of the optimal rate. We also consider the *anisotropic Besov space* (Nikol'skii, 1975), a generalization allowing for different degrees of smoothness $(\alpha_1, \cdots, \alpha_d)$ in each coordinate. Then the optimal rate is nearly dimension-free in the sense that the rate only depends on $d$ through the quantity $\widetilde{\alpha} := (\sum_i \alpha_i^{-1})^{-1}$, and becomes independent of dimension if only a few directions are important i.e. have small $\alpha_i$. Rigorous definitions, statements and proofs are provided in Appendix C.2.

Extending Theorem 4.5, we show that ICL again attains near-optimal estimation error in the anisotropic Besov space, circumventing the curse of dimensionality and theoretically establishing the efficiacy of in-context learning in high-dimensional settings.

**Theorem 4.7** (informal version of Theorem C.7). *For the anisotropic Besov space of smoothness $(\alpha_1, \cdots, \alpha_d)$, assume variance decay (4) with $s = \widetilde{\alpha} > 1/p$. If $T \gtrsim nN$ and $N \asymp n^{\frac{1}{2\widetilde{\alpha}+1}}$, in-context learning achieves the minimax optimal rate $n^{-\frac{2\widetilde{\alpha}}{2\widetilde{\alpha}+1}}$ up to a log factor.*

## 4.4 Learning a Coarser Basis

Thus far, we have demonstrated the importance of sufficient pretraining to achieve optimal risk; as another application of our framework, we illustrate how pretraining can actively *mitigate* the complexity of in-context learning. Consider the case where $(\psi_j^\circ)_{j=1}^\infty$ is no longer the B-spline basis of $B_{p,q}^\alpha(\mathcal{X})$ but instead is chosen from some wider function space, say the unit ball of $B_{p,q}^\tau(\mathcal{X})$ for a smaller smoothness $\tau < \alpha$. Without knowledge of the basis, the sample complexity of learning any regression function $F_\beta^\circ$ is a priori lower bounded by the minimax rate $n^{-\frac{2\tau}{2\tau+d}}$ by Proposition 4.2. For ICL, this difficulty manifests as an increase in the metric entropy of the class $\mathcal{F}_N$ which must be powerful enough to approximate $\psi_{1:N}^\circ$ (Corollary C.12), giving rise to the modified risk bound:

**Corollary 4.8** (ICL for coarser basis). *Suppose $\alpha > \tau > d/p$, the basis $(\psi_j^\circ)_{j=1}^\infty \subset \mathbb{U}(B_{p,q}^\tau(\mathcal{X}))$ and Assumptions 1, 2 hold with $r = 1/2, s = \alpha/d$. Then if $n \gtrsim N \log N$,*

$$\bar{R}(\widehat{\Theta}) \lesssim N^{-\frac{2\alpha}{d}} + \frac{N \log N}{n} + \frac{N^{1+\frac{\alpha}{\tau}+\frac{d}{\tau}} \log^3 N}{T}.$$

*Hence if $T \gtrsim n^{1+\frac{d}{2\alpha+d}\frac{\alpha+d}{\tau}}$ and $N \asymp n^{\frac{d}{2\alpha+d}}$, the risk converges as $n^{-\frac{2\alpha}{2\alpha+d}} \log n$.*

The pretraining generalization gap is now dominated by the higher complexity $N^{1+\frac{\alpha}{\tau}+\frac{d}{\tau}} \log^3 N$ of the DNN class and strictly worse compared to $N^2 \log N$ for Theorem 4.5. The required number of tasks also suffers and the exponent is no longer $\frac{2\alpha+2d}{2\alpha+d} \in (1, 2)$ but scales as $O(d)$. Nevertheless, observe that the burden of complexity is entirely carried by $T$; with sufficient pretraining, the third term can be made arbitrarily small and the ICL risk again attains $n^{-\frac{2\alpha}{2\alpha+d}}$. Hence ICL improves upon the a priori lower bound $n^{-\frac{2\tau}{2\tau+d}}$ at inference time by encoding information on the coarser basis during pretraining. We remark that the result is also readily adapted to the anisotropic setting.

## 4.5 Sequential Input and Transformers

We now consider a more complex setting where the inputs $x \in [0,1]^{d \times \infty}$ are bidirectional *sequences* of tokens (e.g. entire documents) and $\phi$ is itself a transformer network.[2] In this infinite-dimensional setting, transformers can still circumvent the curse of dimensionality and in fact achieve near-optimal sample complexity due to their parameter sharing and feature extraction capabilities (Takakura and Suzuki, 2023). Our goal in this section is to extend this guarantee to ICL of trained transformers.

For sequential data, it is natural to suppose the smoothness w.r.t. each coordinate can vary depending on the input. For example, the position of important tokens in a sentence will change if irrelevant strings are inserted. To this end, we adopt the *piecewise $\gamma$-smooth function class* introduced by Takakura and Suzuki (2023), which allows for arbitrary bounded permutations among input tokens; see Appendix D.1 for definitions. Also borrowing from their setup, we consider multi-head sliding window self-attention layers with window size $U$, embedding dimension $D$, number of heads $H$ with key, query, value matrices $K^{(h)}, Q^{(h)} \in \mathbb{R}^{D \times d}$, $V^{(h)} \in \mathbb{R}^{d \times d}$ and norm bound $M$ defined as[3]

$$\mathcal{F}_{\text{Attn}}(U, D, H, M) = \left\{ g : \mathbb{R}^{d \times \infty} \to \mathbb{R}^{d \times \infty} \;\middle|\; \max_{1 \le h \le H} \|K^{(h)}\|_\infty \vee \|Q^{(h)}\|_\infty \vee \|V^{(h)}\|_\infty \le M, \right.$$

$$\left. g(x)_i = x_i + \sum_{h=1}^H V^{(h)} x_{i-U:i+U} \text{Softmax}\left( (K^{(h)} x_{i-U:i+U})^\top Q^{(h)} x_i \right) \right\}.$$

We also consider a linear embedding layer $\text{Enc}(x) = Ex + P$, $E \in \mathbb{R}^{D \times d}$ with absolute positional encoding $P \in \mathbb{R}^D$ of bounded norm. Then the class of depth $J$ transformers is defined as

$$\mathcal{F}_{\text{TF}}(J, U, D, H, L, W, S, M) := \left\{ f_J \circ g_J \circ \cdots \circ f_1 \circ g_1 \circ \text{Enc} \mid \|E\| \le M, \right.$$

$$\left. f_i \in \mathcal{F}_{\text{DNN}}(L, W, S, M), \; g_i \in \mathcal{F}_{\text{Attn}}(U, D, H, M) \right\}.$$

Our result, proved in Appendix D.2, reads:

---

[2] We clarify that this is *not* equivalent to a multi-layer transformer setting where $\phi$ is the rest of the transformer. Instead, $\phi$ operates on individual tokens $x_i$ separately, which may now themselves be sequences of unbounded dimension. The extracted per-token features are cross-referenced only at the final attention layer $f_\Theta$.

[3] Here the $i$th column and $(j, i)$th component of $x \in \mathbb{R}^{d \times \infty}$ for $i \in \mathbb{Z}, j \in [d]$ are denoted by $x_i$ and $x_{ij}$, respectively. These are not to be confused with sample indexes (1) as those will not be used in this section.

**Theorem 4.9** (informal version of Theorem D.1). *Suppose $\mathcal{F}^\circ$ consists of functions on $[0,1]^{d \times \infty}$ of bounded piecewise $\gamma$-smooth and $L^\infty$-norm with smoothness $\alpha \in \mathbb{R}_{>0}^{d \times \infty}$, and let $\gamma$ be mixed or anisotropic smoothness with $\alpha^\dagger = \max_{i,j} \alpha_{ij}$ or $(\sum_{i,j} \alpha_{ij}^{-1})^{-1}$, respectively. Under suitable regularity and decay assumptions, by taking $\mathcal{F}_N$ to be a class of clipped transformers it holds that*

$$\bar{R}(\widehat{\Theta}) \lesssim N^{-2\alpha^\dagger} + \frac{N \log N}{n} + \frac{N^{2 \vee (1 + 1/\alpha^\dagger)} \operatorname{polylog}(N)}{T}.$$

*Hence if $T \gtrsim n N^{1 \vee 1/\alpha^\dagger}$ and $N \asymp n^{\frac{1}{2\alpha^\dagger + 1}}$, ICL achieves the rate $n^{-\frac{2\alpha^\dagger}{2\alpha^\dagger + 1}} \operatorname{polylog}(n)$.*

This matches the optimal rate in finite dimensions independently of the (possibly infinite) length of the input or context window. The dynamical feature extraction ability of attention layers in the $\mathcal{F}_{\mathrm{TF}}$ class is essential in dealing with input-dependent smoothness, further justifying the efficacy of ICL of sequential data.

# 5  Minimax Lower Bounds

In this section, we provide lower bounds for the minimax rate in both $n, T$ by extending the theory of Yang and Barron (1999), which can be leveraged to yield results stronger than optimality in merely $n$. The bound is purely information-theoretic and hence applies to not just ICL but any meta-learning scheme for the regression problem of Section 2.1 from the data $\mathcal{D}_{n,T} = \{(\mathbf{X}^{(t)}, \boldsymbol{y}^{(t)})\}_{t=1}^{T+1}$, where the index $T + 1$ corresponds to the test task.

For this section we assume that the noise (1) is i.i.d. Gaussian, $\xi_k \sim \mathcal{N}(0, \sigma^2)$, instead of bounded; while the exact shape of the noise distribution is not important, having restricted support may convey additional information and affect the minimax rate. We also suppose for simplicity that the support of $\mathcal{P}_\beta$ is included in $\mathcal{B} := \{\beta \in \mathbb{R}^\infty \mid |\beta_j|^2 \lesssim j^{-2s-1}(\log j)^{-2}, \ j \in \mathbb{N}\}$ and that the aggregated coefficients $\bar{\beta}_j$ for $\underline{N} \leq j \leq \bar{N}$ satisfy $\mathbb{E}_\beta[\bar{\beta}_j^2] \leq \sigma_\beta^2$ for some $\sigma_\beta$ dependent on $N$. The proof of the following statement is given in Appendix F.1.

**Proposition 5.1.** *For $\varepsilon_{n,1}, \varepsilon_{n,2}, \delta_n > 0$, let $Q_1$ and $Q_2$ be the $\varepsilon_{n,1}$- and $\varepsilon_{n,2}$-covering numbers of $\mathcal{F}_N$ and $\mathcal{B}$ respectively, and $M$ be the $\delta_n$-packing number of $\mathcal{F}^\circ$. Suppose that the following conditions are satisfied:*

$$\frac{1}{2\sigma^2}\left(n(T+1)\sigma_\beta^2 \varepsilon_{n,1}^2 + C_2 n \varepsilon_{n,2}^2\right) \leq \log Q_1 + \log Q_2 \leq \tfrac{1}{8}\log M, \quad 4\log 2 \leq \log M. \quad (7)$$

*Then the minimax rate is lower bounded as*

$$\inf_{\widehat{f}: \mathcal{D}_{n,T} \to \mathbb{R}} \sup_{f^\circ \in \mathcal{F}^\circ} \mathbb{E}_{\mathcal{D}_{n,T}}[\|\widehat{f} - f^\circ\|_{L^2(\mathcal{P}_\mathcal{X})}^2] \geq \tfrac{1}{4}\delta_n^2.$$

Finally, Proposition 5.1 is applied to obtain concrete lower bounds for the settings studied in Section 4 throughout Appendices F.2-F.4.

**Corollary 5.2** (minimax lower bound). *The minimax rates in the previous regression settings are lower bounded as follows.*

(i) *Besov space (Section 4.2):* $\inf_{\widehat{f}} \sup_{f^\circ} \mathbb{E}_{\mathcal{D}_{n,T}}[\|\widehat{f} - f^\circ\|^2] \gtrsim n^{-\frac{2\alpha}{2\alpha+d}}$,

(ii) *Coarser basis (Section 4.4):* $\inf_{\widehat{f}} \sup_{f^\circ} \mathbb{E}_{\mathcal{D}_{n,T}}[\|\widehat{f} - f^\circ\|^2] \gtrsim n^{-\frac{2\alpha}{2\alpha+d}} + (nT)^{-\frac{2\tau}{2\tau+d}}$,

(iii) *Sequential input (Section 4.5):* $\inf_{\widehat{f}} \sup_{f^\circ} \mathbb{E}_{\mathcal{D}_{n,T}}[\|\widehat{f} - f^\circ\|^2] \gtrsim n^{-\frac{2\alpha^\dagger}{2\alpha^\dagger+1}}$.

These results match the upper bounds for (i), (iii) and show that **ICL is provably jointly optimal** in $n, T$ in the 'large $T$' regime. Moreover, we can check for the coarser basis setting that insufficient pretraining $T = O(1)$ indeed leads to the worse complexity $n^{-\frac{2\tau}{2\tau+d}}$, while the faster rate $n^{-\frac{2\alpha}{2\alpha+d}}$ is retrieved when $T \gtrsim n^{\frac{(\alpha-\tau)d}{(2\alpha+d)\tau}}$. This aligns with the discussion in Section 4.4, showing that ICL is **provably suboptimal** in the 'small $T$' regime.

**Remark 5.3.** The obtained upper and lower bounds in the coarser basis setting are not tight as $T$ varies, hence it remains to be shown whether there exists a meta-learning algorithm that attains the lower bound (ii). The task diversity threshold for optimal learning suggested by the bounds are also different ($n^{1+\frac{d(\alpha+d)}{(2\alpha+d)}}$ v.s. $n^{\frac{(\alpha-\tau)d}{(2\alpha+d)\tau}}$); it would be interesting for future work to resolve this gap.

## 6  Conclusion

In this paper, we performed a learning-theoretic analysis of ICL of a transformer consisting of a DNN and a linear attention layer pretrained on nonparametric regression tasks. We developed a general framework for bounding the in-context estimation error of the empirical risk minimizer in terms of both the number of tasks and samples, and proved that ICL can achieve nearly minimax optimal rates in the Besov space, anisotropic Besov space and $\gamma$-smooth class. We also demonstrated that ICL can improve upon the a priori optimal rate by learning informative representations during pretraining. We supplemented our analyses with corresponding minimax lower bounds jointly in $n, T$ and also performed numerical experiments validating our findings. Our work opens up interesting approaches of adapting classical learning theory to study emergent phenomena of foundation models.

**Limitations.**  Our transformer model is limited to a single layer of linear self-attention and does not consider more complex in-context learning behavior which may arise in transformers with multiple attention layers. Moreover, the obtained upper and lower bounds are not tight in certain regimes, suggesting future research directions for meta-learning.

## Acknowledgments

JK was partially supported by JST CREST (JPMJCR2015). TS was partially supported by JSPS KAKENHI (24K02905, 20H00576) and JST CREST (JPMJCR2115). We would like to express our gratitude to Masaaki Imaizumi for valuable and insightful discussions on the topic in relation to his work in progress (Imaizumi, 2024).

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

## Table of Contents

# Appendix

## A    Proof of Proposition 3.2

### A.1    Decomposing Approximation Error

Recall that

$$\Theta^* = (\Gamma^*, \phi^*) = \left( \left( \Sigma_{\Psi, N} + \frac{1}{n} \Sigma_{\bar{\beta}, N}^{-1} \right)^{-1}, \phi_{N:\bar{N}}^* \right).$$

We introduce some additional notation in the following fashion. For brevity, we write $\bar{N} : \infty$ instead of $(\bar{N}+1) : \infty$ as an exception.

$$\Phi^* = (\phi_{N:\bar{N}}^*(x_1), \cdots, \phi_{N:\bar{N}}^*(x_n)) \in \mathbb{R}^{N \times n}, \quad \xi = (\xi_1, \ldots, \xi_n)^\top \in \mathbb{R}^n,$$

$$\Psi^\circ = (\psi^\circ(x_1), \ldots, \psi^\circ(x_n)) \in \mathbb{R}^{\infty \times n}, \quad \Psi_{N:\bar{N}}^\circ = (\psi_{N:\bar{N}}^\circ(x_1), \cdots, \psi_{N:\bar{N}}^\circ(x_n)) \in \mathbb{R}^{N \times n},$$

$$\Psi_{N:\infty}^\circ = (\psi_{N:\infty}^\circ(x_1), \cdots, \psi_{N:\infty}^\circ(x_n)) \in \mathbb{R}^{\infty \times n}.$$

Since clipping $\check{f}_\Theta$ does not make its difference with $F_\beta^\circ \in [-B, B]$ larger, it holds that

$$
\begin{aligned}
R(\Theta^*) &= \mathbb{E}\left[ \left( F_\beta^\circ(\tilde{x}) - f_{\Theta^*}(\mathbf{X}, \boldsymbol{y}, \tilde{x}) \right)^2 \right] \\
&\leq \mathbb{E}\left[ \left( F_\beta^\circ(\tilde{x}) - \check{f}_{\Theta^*}(\mathbf{X}, \boldsymbol{y}, \tilde{x}) \right)^2 \right] \\
&\leq 2\mathbb{E}\left[ \left( F_\beta^\circ(\tilde{x}) - F_{\beta, \bar{N}}^\circ(\tilde{x}) \right)^2 \right] + 2\mathbb{E}\left[ \left( F_{\beta, \bar{N}}^\circ(\tilde{x}) - \check{f}_{\Theta^*}(\mathbf{X}, \boldsymbol{y}, \tilde{x}) \right)^2 \right] \\
&\lesssim N^{-2s} + \mathbb{E}\left[ \left( F_{\beta, \bar{N}}^\circ(\tilde{x}) - \phi^*(\tilde{x})^\top \frac{\Gamma^* \Phi^* \boldsymbol{y}}{n} \right)^2 \right]
\end{aligned}
$$

due to Assumption 2 and $\bar{N} \asymp N$. Expanding the attention output as

$$
\begin{aligned}
\phi^*(\tilde{x})^\top \frac{\Gamma^* \Phi^* \boldsymbol{y}}{n} &= (\phi^*(\tilde{x}) - \psi_{N:\bar{N}}^\circ(\tilde{x}) + \psi_{N:\bar{N}}^\circ(\tilde{x}))^\top \Gamma^* \frac{(\Phi^* - \Psi_{N:\bar{N}}^\circ + \Psi_{N:\bar{N}}^\circ) \boldsymbol{y}}{n} \\
&= (\phi^*(\tilde{x}) - \psi_{N:\bar{N}}^\circ(\tilde{x}))^\top \Gamma^* \frac{(\Phi^* - \Psi_{N:\bar{N}}^\circ) \boldsymbol{y}}{n} \\
&\quad + (\phi^*(\tilde{x}) - \psi_{N:\bar{N}}^\circ(\tilde{x}))^\top \Gamma^* \frac{\Psi_{N:\bar{N}}^\circ \boldsymbol{y}}{n} \\
&\quad + \psi_{N:\bar{N}}^\circ(\tilde{x})^\top \Gamma^* \frac{(\Phi^* - \Psi_{N:\bar{N}}^\circ) \boldsymbol{y}}{n} \\
&\quad + \psi_{N:\bar{N}}^\circ(\tilde{x})^\top \Gamma^* \frac{\Psi_{N:\bar{N}}^\circ \boldsymbol{y}}{n},
\end{aligned}
$$

and the final term further as

$$
\begin{aligned}
\psi_{N:\bar{N}}^\circ(\tilde{x})^\top \Gamma^* \frac{\Psi_{N:\bar{N}}^\circ \boldsymbol{y}}{n} &= \psi_{N:\bar{N}}^\circ(\tilde{x})^\top \Gamma^* \frac{\Psi_{N:\bar{N}}^\circ (\Psi^{\circ\top} \beta + \xi)}{n} \\
&= \psi_{N:\bar{N}}^\circ(\tilde{x})^\top \Gamma^* \frac{\Psi_{N:\bar{N}}^\circ \Psi_{N:\bar{N}}^{\circ\top} \bar{\beta}_{N:\bar{N}}}{n} + \psi_{1:N}^\circ(\tilde{x})^\top \Gamma^* \frac{\Psi_{N:\bar{N}}^\circ (\Psi_{\bar{N}:\infty}^{\circ\top} \beta_{\bar{N}:\infty} + \xi)}{n},
\end{aligned}
$$

we obtain that

$$
\begin{aligned}
&\mathbb{E}\left[ \left( F_{\beta, \bar{N}}^\circ(\tilde{x}) - \phi^*(\tilde{x})^\top \frac{\Gamma^* \Phi^* \boldsymbol{y}}{n} \right)^2 \right] \\
&\lesssim \mathbb{E}\left[ \left( F_{\beta, \bar{N}}^\circ(\tilde{x}) - \psi_{N:\bar{N}}^\circ(\tilde{x})^\top \Gamma^* \frac{\Psi_{N:\bar{N}}^\circ \Psi_{N:\bar{N}}^{\circ\top} \bar{\beta}_{N:\bar{N}}}{n} \right)^2 \right]
\end{aligned}
\tag{8}
$$

$$+ \mathbb{E}\left[\left(\psi_{\underline{N}:\bar{N}}^{\circ}(\tilde{x})^{\top}\Gamma^{*}\frac{\Psi_{\underline{N}:\bar{N}}^{\circ}(\Psi_{\bar{N}:\infty}^{\circ\top}\beta_{\bar{N}:\infty}+\xi)}{n}\right)^{2}\right] \tag{9}$$

$$+ \mathbb{E}\left[\left((\phi^{*}(\tilde{x})-\psi_{\underline{N}:\bar{N}}^{\circ}(\tilde{x}))^{\top}\Gamma^{*}\frac{(\Phi^{*}-\Psi_{\underline{N}:\bar{N}}^{\circ})\boldsymbol{y}}{n}\right)^{2}\right] \tag{10}$$

$$+ \mathbb{E}\left[\left((\phi^{*}(\tilde{x})-\psi_{\underline{N}:\bar{N}}^{\circ}(\tilde{x}))^{\top}\Gamma^{*}\frac{\Psi_{\underline{N}:\bar{N}}^{\circ}\boldsymbol{y}}{n}\right)^{2}\right] \tag{11}$$

$$+ \mathbb{E}\left[\left(\psi_{\underline{N}:\bar{N}}^{\circ}(\tilde{x})^{\top}\Gamma^{*}\frac{(\Phi^{*}-\Psi_{\underline{N}:\bar{N}}^{\circ})\boldsymbol{y}}{n}\right)^{2}\right]. \tag{12}$$

We proceed to control each term separately, from which the statement of Proposition 3.2 will follow.

## A.2 Bounding Term (8)

Since $F_{\beta,\bar{N}}^{\circ}(\tilde{x})=\psi_{1:\bar{N}}^{\circ}(\tilde{x})^{\top}\beta_{1:\bar{N}}=\psi_{\underline{N}:\bar{N}}^{\circ}(\tilde{x})^{\top}\bar{\beta}_{\underline{N}:\bar{N}}$, we can introduce a $(\Gamma^{*})^{-1}$ factor to bound

$$\mathbb{E}\left[\left(F_{\beta,\bar{N}}^{\circ}(\tilde{x})-\psi_{\underline{N}:\bar{N}}^{\circ}(\tilde{x})^{\top}\Gamma^{*}\frac{\Psi_{\underline{N}:\bar{N}}^{\circ}\Psi_{\underline{N}:\bar{N}}^{\circ\top}\bar{\beta}_{\underline{N}:\bar{N}}}{n}\right)^{2}\right]$$

$$= \mathbb{E}\left[\left(F_{\beta,\bar{N}}^{\circ}(\tilde{x})-\psi_{\underline{N}:\bar{N}}^{\circ}(\tilde{x})^{\top}\Gamma^{*}\left(\frac{\Psi_{\underline{N}:\bar{N}}^{\circ}\Psi_{\underline{N}:\bar{N}}^{\circ\top}}{n}-(\Gamma^{*})^{-1}+(\Gamma^{*})^{-1}\right)\bar{\beta}_{\underline{N}:\bar{N}}\right)^{2}\right]$$

$$= \mathbb{E}\left[\left(\psi_{\underline{N}:\bar{N}}^{\circ}(\tilde{x})^{\top}\Gamma^{*}\left(\frac{\Psi_{\underline{N}:\bar{N}}^{\circ}\Psi_{\underline{N}:\bar{N}}^{\circ\top}}{n}-\Sigma_{\Psi,N}-\frac{1}{n}\Sigma_{\bar{\beta},N}^{-1}\right)\bar{\beta}_{\underline{N}:\bar{N}}\right)^{2}\right]$$

$$\leq 2\mathbb{E}\left[\left(\psi_{\underline{N}:\bar{N}}^{\circ}(\tilde{x})^{\top}\Gamma^{*}\left(\frac{\Psi_{\underline{N}:\bar{N}}^{\circ}\Psi_{\underline{N}:\bar{N}}^{\circ\top}}{n}-\Sigma_{\Psi,N}\right)\bar{\beta}_{\underline{N}:\bar{N}}\right)^{2}\right] \tag{13}$$

$$+ 2\mathbb{E}\left[\left(\psi_{\underline{N}:\bar{N}}^{\circ}(\tilde{x})^{\top}\Gamma^{*}\frac{\Sigma_{\bar{\beta},N}^{-1}}{n}\bar{\beta}_{\underline{N}:\bar{N}}\right)^{2}\right]. \tag{14}$$

Denote the operator norm (with respect to $L^{2}$ norm of vectors) by $\|\cdot\|_{\mathrm{op}}$. For (13), noting that $\Sigma_{\Psi,N}$ is positive definite,

$$\mathbb{E}\left[\left(\psi_{\underline{N}:\bar{N}}^{\circ}(\tilde{x})^{\top}\Gamma^{*}\left(\frac{\Psi_{\underline{N}:\bar{N}}^{\circ}\Psi_{\underline{N}:\bar{N}}^{\circ\top}}{n}-\Sigma_{\Psi,N}\right)\bar{\beta}_{\underline{N}:\bar{N}}\right)^{2}\right]$$

$$= \mathbb{E}\left[\left(\psi_{\underline{N}:\bar{N}}^{\circ}(\tilde{x})^{\top}\Gamma^{*}\Sigma_{\Psi,N}^{1/2}\left(\Sigma_{\Psi,N}^{-1/2}\frac{\Psi_{\underline{N}:\bar{N}}^{\circ}\Psi_{\underline{N}:\bar{N}}^{\circ\top}}{n}\Sigma_{\Psi,N}^{-1/2}-\mathbf{I}_{N}\right)\Sigma_{\Psi,N}^{1/2}\bar{\beta}_{\underline{N}:\bar{N}}\right)^{2}\right]$$

$$= \mathbb{E}_{\mathbf{X},\beta}\left[\bar{\beta}_{\underline{N}:\bar{N}}^{\top}\Sigma_{\Psi,N}^{1/2}\left(\Sigma_{\Psi,N}^{-1/2}\frac{\Psi_{\underline{N}:\bar{N}}^{\circ}\Psi_{\underline{N}:\bar{N}}^{\circ\top}}{n}\Sigma_{\Psi,N}^{-1/2}-\mathbf{I}_{N}\right)\Sigma_{\Psi,N}^{1/2}\Gamma^{*}\mathbb{E}_{\tilde{x}}\left[\psi_{\underline{N}:\bar{N}}^{\circ}(\tilde{x})\psi_{\underline{N}:\bar{N}}^{\circ}(\tilde{x})^{\top}\right]\right.$$

$$\left. \times \Gamma^{*}\Sigma_{\Psi,N}^{1/2}\left(\Sigma_{\Psi,N}^{-1/2}\frac{\Psi_{\underline{N}:\bar{N}}^{\circ}\Psi_{\underline{N}:\bar{N}}^{\circ\top}}{n}\Sigma_{\Psi,N}^{-1/2}-\mathbf{I}_{N}\right)\Sigma_{\Psi,N}^{1/2}\bar{\beta}_{\underline{N}:\bar{N}}\right]$$

$$= \mathbb{E}_{\mathbf{X}}\left[\mathrm{Tr}\left[\left(\Sigma_{\Psi,N}^{-1/2}\frac{\Psi_{\underline{N}:\bar{N}}^{\circ}\Psi_{\underline{N}:\bar{N}}^{\circ\top}}{n}\Sigma_{\Psi,N}^{-1/2}-\mathbf{I}_{N}\right)\Sigma_{\Psi,N}^{1/2}\mathbb{E}_{\beta}\left[\bar{\beta}_{\underline{N}:\bar{N}}\bar{\beta}_{\underline{N}:\bar{N}}^{\top}\right]\Sigma_{\Psi,N}^{1/2}\right.\right.$$

$$\left.\left. \times \left(\Sigma_{\Psi,N}^{-1/2}\frac{\Psi_{\underline{N}:\bar{N}}^{\circ}\Psi_{\underline{N}:\bar{N}}^{\circ\top}}{n}\Sigma_{\Psi,N}^{-1/2}-\mathbf{I}_{N}\right)\Sigma_{\Psi,N}^{1/2}\Gamma^{*}\Sigma_{\Psi,N}\Gamma^{*}\Sigma_{\Psi,N}^{1/2}\right]\right]$$

$$\lesssim \mathbb{E}_{\mathbf{X}} \left[ \mathrm{Tr} \left[ \left( \Sigma_{\Psi,N}^{-1/2} \frac{\Psi_{\underline{N}:\bar{N}}^{\circ} \Psi_{\underline{N}:\bar{N}}^{\circ\top}}{n} \Sigma_{\Psi,N}^{-1/2} - \mathbf{I}_N \right)^2 \Sigma_{\Psi,N}^{1/2} \Sigma_{\bar{\beta},N} \Sigma_{\Psi,N}^{1/2} \right] \right]$$

due to the independence of $\mathbf{X}, \tilde{x}$ and $\beta$. For the last inequality, we have used the fact that both the $\Sigma_{\Psi,N}^{1/2} \Gamma^* \Sigma_{\Psi,N} \Gamma^* \Sigma_{\Psi,N}^{1/2}$ term and the matrix multiplied to it are positive semi-definite, and the former is bounded above as $\precsim \mathbf{I}_N$ by Assumption 1.

Furthermore, we utilize the following result proved in Appendix A.5:

**Lemma A.1.** $\mathbb{E}_{\mathbf{X}} \left[ \left\| \Sigma_{\Psi,N}^{-1/2} \frac{\Psi_{\underline{N}:\bar{N}}^{\circ} \Psi_{\underline{N}:\bar{N}}^{\circ\top}}{n} \Sigma_{\Psi,N}^{-1/2} - \mathbf{I}_N \right\|_{\mathrm{op}}^2 \right] \lesssim \frac{N^{2r}}{n} \log N + \frac{N^{4r}}{n^2} \log^2 N.$

It follows that (13) is bounded as

$$\mathbb{E} \left[ \left( \psi_{\underline{N}:\bar{N}}^{\circ}(\tilde{x})^\top \Gamma^* \left( \frac{\Psi_{\underline{N}:\bar{N}}^{\circ} \Psi_{\underline{N}:\bar{N}}^{\circ\top}}{n} - \Sigma_{\Psi,N} \right) \bar{\beta}_{\underline{N}:\bar{N}} \right)^2 \right]$$

$$\lesssim \mathbb{E}_{\mathbf{X}} \left[ \left\| \Sigma_{\Psi,N}^{-1/2} \frac{\Psi_{\underline{N}:\bar{N}}^{\circ} \Psi_{\underline{N}:\bar{N}}^{\circ\top}}{n} \Sigma_{\Psi,N}^{-1/2} - \mathbf{I}_N \right\|_{\mathrm{op}}^2 \right] \mathrm{Tr} \left[ \Sigma_{\Psi,N}^{1/2} \Sigma_{\bar{\beta},N} \Sigma_{\Psi,N}^{1/2} \right]$$

$$\lesssim \left( \frac{N^{2r}}{n} \log N + \frac{N^{4r}}{n^2} \log^2 N \right) \| \Sigma_{\Psi,N} \|_{\mathrm{op}} \mathrm{Tr}(\Sigma_{\bar{\beta},N})$$

$$\lesssim \frac{N^{2r}}{n} \log N + \frac{N^{4r}}{n^2} \log^2 N$$

since $\mathrm{Tr}(\Sigma_{\bar{\beta},N})$ is bounded by Assumption 2.

Moreover for (14), we compute

$$\mathbb{E} \left[ \left( \psi_{\underline{N}:\bar{N}}^{\circ}(\tilde{x})^\top \Gamma^* \frac{\Sigma_{\bar{\beta},N}^{-1}}{n} \bar{\beta}_{\underline{N}:\bar{N}} \right)^2 \right]$$

$$= \mathbb{E} \left[ \bar{\beta}_{\underline{N}:\bar{N}}^\top \frac{\Sigma_{\bar{\beta},N}^{-1}}{n} \Gamma^* \psi_{\underline{N}:\bar{N}}^{\circ}(\tilde{x}) \psi_{\underline{N}:\bar{N}}^{\circ}(\tilde{x})^\top \Gamma^* \frac{\Sigma_{\bar{\beta},N}^{-1}}{n} \bar{\beta}_{\underline{N}:\bar{N}} \right]$$

$$= \mathbb{E} \left[ \mathrm{Tr} \left[ \frac{\Sigma_{\bar{\beta},N}^{-1}}{n} \Gamma^* \psi_{\underline{N}:\bar{N}}^{\circ}(\tilde{x}) \psi_{\underline{N}:\bar{N}}^{\circ}(\tilde{x})^\top \Gamma^* \frac{\Sigma_{\bar{\beta},N}^{-1}}{n} \bar{\beta}_{\underline{N}:\bar{N}} \bar{\beta}_{\underline{N}:\bar{N}}^\top \right] \right]$$

$$= \mathrm{Tr} \left[ \frac{\Sigma_{\bar{\beta},N}^{-1}}{n} \Gamma^* \mathbb{E}_{\tilde{x}} \left[ \psi_{\underline{N}:\bar{N}}^{\circ}(\tilde{x}) \psi_{\underline{N}:\bar{N}}^{\circ}(\tilde{x})^\top \right] \Gamma^* \frac{\Sigma_{\bar{\beta},N}^{-1}}{n} \mathbb{E}_{\beta} \left[ \bar{\beta}_{\underline{N}:\bar{N}} \bar{\beta}_{\underline{N}:\bar{N}}^\top \right] \right]$$

$$= \mathrm{Tr} \left[ \frac{\Sigma_{\bar{\beta},N}^{-1}}{n} \underbrace{\Gamma^* \Sigma_{\Psi,N} \Gamma^*}_{\text{positive definite}} \frac{\Sigma_{\bar{\beta},N}^{-1}}{n} \Sigma_{\bar{\beta},N} \right]$$

$$\leq \frac{1}{n} \mathrm{Tr} \left[ \left( \Sigma_{\Psi,N} + \frac{1}{n} \Sigma_{\bar{\beta},N}^{-1} \right) \Gamma^* \Sigma_{\Psi,N} \Gamma^* \right]$$

$$= \frac{1}{n} \mathrm{Tr} \left[ \Sigma_{\Psi,N} \left( \Sigma_{\Psi,N} + \frac{1}{n} \Sigma_{\bar{\beta},N}^{-1} \right)^{-1} \right] \leq \frac{N}{n}.$$

### A.3 Bounding Term (9)

Since the sequence of covariates $x_1, \cdots, x_n$ and noise $\xi_1, \cdots, \xi_n$ are each i.i.d.,

$$\mathbb{E} \left[ \left( \psi_{\underline{N}:\bar{N}}^{\circ}(\tilde{x})^\top \Gamma^* \frac{\Psi_{\underline{N}:\bar{N}}^{\circ} (\Psi_{\bar{N}:\infty}^{\circ\top} \beta_{\bar{N}:\infty} + \xi)}{n} \right)^2 \right]$$

$$= \mathbb{E}\left[\left(\sum_{i=1}^{n} \psi^{\circ}_{\underline{N}:\bar{N}}(\tilde{x})^{\top}\Gamma^{*}\frac{\psi^{\circ}_{\underline{N}:\bar{N}}(x_i)(\beta^{\top}_{\bar{N}:\infty}\psi^{\circ}_{\bar{N}:\infty}(x_i)+\xi_i)}{n}\right)^{2}\right]$$

$$= \frac{1}{n^2}\mathbb{E}\left[\left(\sum_{i=1}^{n}\psi^{\circ}_{\underline{N}:\bar{N}}(\tilde{x})^{\top}\Gamma^{*}\psi^{\circ}_{\underline{N}:\bar{N}}(x_i)\beta^{\top}_{\bar{N}:\infty}\psi^{\circ}_{\bar{N}:\infty}(x_i)+\sum_{i=1}^{n}\psi^{\circ}_{\underline{N}:\bar{N}}(\tilde{x})^{\top}\Gamma^{*}\psi^{\circ}_{\underline{N}:\bar{N}}(x_i)\xi_i\right)^{2}\right]$$

$$\leq \frac{1}{n}\mathbb{E}\left[\left(\psi^{\circ}_{\underline{N}:\bar{N}}(\tilde{x})^{\top}\Gamma^{*}\psi^{\circ}_{\underline{N}:\bar{N}}(x)\beta^{\top}_{\bar{N}:\infty}\psi^{\circ}_{\bar{N}:\infty}(x)\right)^{2}\right] \tag{15}$$

$$+ \frac{n-1}{n}\mathbb{E}\left[\psi^{\circ}_{\underline{N}:\bar{N}}(\tilde{x})^{\top}\Gamma^{*}\psi^{\circ}_{\underline{N}:\bar{N}}(x)\beta^{\top}_{\bar{N}:\infty}\psi^{\circ}_{\bar{N}:\infty}(x)\psi^{\circ}_{\underline{N}:\bar{N}}(\tilde{x})^{\top}\Gamma^{*}\psi^{\circ}_{\underline{N}:\bar{N}}(x')\beta^{\top}_{\bar{N}:\infty}\psi^{\circ}_{\bar{N}:\infty}(x')\right] \tag{16}$$

$$+ \frac{\sigma^2}{n}\mathbb{E}\left[\left(\psi^{\circ}_{\underline{N}:\bar{N}}(\tilde{x})^{\top}\Gamma^{*}\psi^{\circ}_{\underline{N}:\bar{N}}(x)\right)^{2}\right] \tag{17}$$

for independent samples $x, x', \tilde{x} \sim \mathcal{P}_{\mathcal{X}}$. We now bound the three terms separately below.

For (15), we have that

$$\frac{1}{n}\mathbb{E}\left[\left(\psi^{\circ}_{\underline{N}:\bar{N}}(\tilde{x})^{\top}\Gamma^{*}\psi^{\circ}_{\underline{N}:\bar{N}}(x)\beta^{\top}_{\bar{N}:\infty}\psi^{\circ}_{\bar{N}:\infty}(x)\right)^{2}\right]$$

$$= \frac{1}{n}\mathbb{E}\left[\psi^{\circ}_{\underline{N}:\bar{N}}(x)^{\top}\Gamma^{*}\psi^{\circ}_{\underline{N}:\bar{N}}(\tilde{x})\psi^{\circ}_{\underline{N}:\bar{N}}(\tilde{x})^{\top}\Gamma^{*}\psi^{\circ}_{\underline{N}:\bar{N}}(x)\psi^{\circ}_{\bar{N}:\infty}(x)^{\top}\beta_{\bar{N}:\infty}\beta^{\top}_{\bar{N}:\infty}\psi^{\circ}_{\bar{N}:\infty}(x)\right]$$

$$= \frac{1}{n}\mathbb{E}_x\left[\psi^{\circ}_{\underline{N}:\bar{N}}(x)^{\top}\Gamma^{*}\Sigma_{\Psi,N}\Gamma^{*}\psi^{\circ}_{\underline{N}:\bar{N}}(x)\psi^{\circ}_{\bar{N}:\infty}(x)^{\top}\mathbb{E}_{\beta}\left[\beta_{\bar{N}:\infty}\beta^{\top}_{\bar{N}:\infty}\right]\psi^{\circ}_{\bar{N}:\infty}(x)\right]$$

$$\lesssim \frac{1}{n}\mathbb{E}_x\left[\|\psi^{\circ}_{\underline{N}:\bar{N}}(x)\|^2\psi^{\circ}_{\bar{N}:\infty}(x)^{\top}\mathbb{E}_{\beta}\left[\beta_{\bar{N}:\infty}\beta^{\top}_{\bar{N}:\infty}\right]\psi^{\circ}_{\bar{N}:\infty}(x)\right]$$

$$\lesssim \frac{1}{n}\sup_{x\in\mathrm{supp}\,\mathcal{P}_{\mathcal{X}}}\|\psi^{\circ}_{\underline{N}:\bar{N}}(x)\|^2 \cdot \lim_{M\to\infty}\mathrm{Tr}\left(\mathbb{E}_{\beta}\left[\beta_{\bar{N}:M}\beta^{\top}_{\bar{N}:M}\right]\mathbb{E}_x\left[\psi^{\circ}_{\bar{N}:M}(x)\psi^{\circ}_{\bar{N}:M}(x)^{\top}\right]\right)$$

$$\lesssim \frac{1}{n}\sup_{x\in\mathrm{supp}\,\mathcal{P}_{\mathcal{X}}}\|\psi^{\circ}_{\underline{N}:\bar{N}}(x)\|^2 \cdot \lim_{M\to\infty}\mathrm{Tr}\left(\mathbb{E}_{\beta}\left[\beta_{\bar{N}:M}\beta^{\top}_{\bar{N}:M}\right]\right)$$

since $\Sigma_{\Psi,M} \preceq C_2\mathbf{I}_N$ as $M \to \infty$ by Assumption 2. As $\sup_x\|\psi^{\circ}_{\underline{N}:\bar{N}}(x)\|^2 \lesssim N^{-2r}$ by (2) and the diagonal of $\mathbb{E}_{\beta}[\beta_{\bar{N}:M}\beta^{\top}_{\bar{N}:M}]$ decays faster than $\bar{N}^{-2s}M^{-1}(\log M)^{-2}$ when $M > \bar{N}$ due to (4), it follows that

$$\frac{1}{n}\sup_{x\in\mathrm{supp}\,\mathcal{P}_{\mathcal{X}}}\|\psi^{\circ}_{\underline{N}:\bar{N}}(x)\|^2 \cdot \lim_{M\to\infty}\mathrm{Tr}\left(\mathbb{E}_{\beta}\left[\beta_{\bar{N}:M}\beta^{\top}_{\bar{N}:M}\right]\right) \lesssim \frac{1}{n}N^{2r}N^{-2s}.$$

Similarly, for (16), we have that

$$\frac{n-1}{n}\mathbb{E}\left[\psi^{\circ}_{\underline{N}:\bar{N}}(\tilde{x})^{\top}\Gamma^{*}\psi^{\circ}_{\underline{N}:\bar{N}}(x)\beta^{\top}_{\bar{N}:\infty}\psi^{\circ}_{\bar{N}:\infty}(x)\psi^{\circ}_{\underline{N}:\bar{N}}(\tilde{x})^{\top}\Gamma^{*}\psi^{\circ}_{\underline{N}:\bar{N}}(x')\beta^{\top}_{\bar{N}:\infty}\psi^{\circ}_{\bar{N}:\infty}(x')\right]$$

$$\leq \mathbb{E}\left[\left(\psi^{\circ}_{\underline{N}:\bar{N}}(\tilde{x})^{\top}\Gamma^{*}\psi^{\circ}_{\underline{N}:\bar{N}}(x)\beta^{\top}_{\bar{N}:\infty}\psi^{\circ}_{\bar{N}:\infty}(x)\right)^{\top}\left(\psi^{\circ}_{\underline{N}:\bar{N}}(\tilde{x})^{\top}\Gamma^{*}\psi^{\circ}_{\underline{N}:\bar{N}}(x')\beta^{\top}_{\bar{N}:\infty}\psi^{\circ}_{\bar{N}:\infty}(x')\right)\right]$$

$$= \mathbb{E}_{x,x',\beta}\left[\psi^{\circ}_{\bar{N}:\infty}(x)^{\top}\beta_{\bar{N}:\infty}\psi^{\circ}_{\underline{N}:\bar{N}}(x)^{\top}\Gamma^{*}\Sigma_{\Psi,N}\Gamma^{*}\psi^{\circ}_{\underline{N}:\bar{N}}(x')\beta^{\top}_{\bar{N}:\infty}\psi^{\circ}_{\bar{N}:\infty}(x')\right]$$

$$= \mathbb{E}_{\beta}\left[\mathbb{E}_x\left[\psi^{\circ}_{\underline{N}:\bar{N}}(x)\beta^{\top}_{\bar{N}:\infty}\psi^{\circ}_{\bar{N}:\infty}(x)\right]^{\top}\Gamma^{*}\Sigma_{\Psi,N}\Gamma^{*}\mathbb{E}_x\left[\psi^{\circ}_{\underline{N}:\bar{N}}(x)\beta^{\top}_{\bar{N}:\infty}\psi^{\circ}_{\bar{N}:\infty}(x)\right]\right]$$

$$\lesssim \mathbb{E}_{\beta}\left[\left\|\mathbb{E}_x\left[\psi^{\circ}_{\underline{N}:\bar{N}}(x)\beta^{\top}_{\bar{N}:\infty}\psi^{\circ}_{\bar{N}:\infty}(x)\right]\right\|^2\right]$$

$$\lesssim \mathbb{E}_{\beta}\left[\psi^{\circ}_{\bar{N}:\infty}(x)^{\top}\beta_{\bar{N}:\infty}\beta^{\top}_{\bar{N}:\infty}\psi^{\circ}_{\bar{N}:\infty}(x)\right]$$

$$\lesssim N^{-2s}.$$

And for (17), we have that

$$\frac{\sigma^2}{n}\mathbb{E}\left[\left(\psi^{\circ}_{\underline{N}:\bar{N}}(\tilde{x})^{\top}\Gamma^{*}\psi^{\circ}_{\underline{N}:\bar{N}}(x)\right)^{2}\right]$$

$$
= \frac{\sigma^2}{n} \mathrm{Tr} \left[ \Gamma^* \mathbb{E}_{\tilde{x}} \left[ \psi^{\circ}_{\underline{N}:\bar{N}}(\tilde{x}) \psi^{\circ}_{\underline{N}:\bar{N}}(\tilde{x})^{\top} \right] \Gamma^* \mathbb{E}_x \left[ \psi^{\circ}_{\underline{N}:\bar{N}}(x) \psi^{\circ}_{\underline{N}:\bar{N}}(x)^{\top} \right] \right]
$$

$$
= \frac{\sigma^2}{n} \mathrm{Tr} \big[ \underbrace{\Gamma^* \Sigma_{\Psi,N} \Gamma^*}_{\text{positive definite}} \Sigma_{\Psi,N} \big]
$$

$$
\leq \frac{\sigma^2}{n} \mathrm{Tr} \left[ \Gamma^* \Sigma_{\Psi,N} \Gamma^* \left( \Sigma_{\Psi,N} + \frac{1}{n} \Sigma_{\bar{\beta},N}^{-1} \right) \right]
$$

$$
= \frac{\sigma^2}{n} \mathrm{Tr} \left[ \Gamma^* \Sigma_{\Psi,N} \right] \leq \frac{\sigma^2 N}{n}.
$$

Therefore, we obtain the following bound:

$$
\mathbb{E} \left[ \left( \psi^{\circ}_{\underline{N}:\bar{N}}(\tilde{x})^{\top} \Gamma^* \frac{\Psi^{\circ}_{\underline{N}:\bar{N}} (\Psi^{\circ\top}_{\bar{N}:\infty} \beta_{\bar{N}:\infty} + \xi)}{n} \right)^2 \right] \lesssim \frac{N^{2r} N^{-2s}}{n} + N^{-2s} + \frac{\sigma^2 N}{n}.
$$

### A.4  Bounding Terms (10)-(12)

For (10), we use the Cauchy-Schwarz inequality and Assumption 3 to bound

$$
\mathbb{E} \left[ \left( (\phi^*(\tilde{x}) - \psi^{\circ}_{\underline{N}:\bar{N}}(\tilde{x}))^{\top} \Gamma^* \frac{(\Phi^* - \Psi^{\circ}_{\underline{N}:\bar{N}}) \boldsymbol{y}}{n} \right)^2 \right]
$$

$$
\leq \frac{1}{n} \sum_{i=1}^{n} \mathbb{E} \left[ \left( (\phi^*(\tilde{x}) - \psi^{\circ}_{\underline{N}:\bar{N}}(\tilde{x}))^{\top} \Gamma^* (\phi^*(x_i) - \psi^{\circ}_{\underline{N}:\bar{N}}(x_i)) y_i \right)^2 \right]
$$

$$
\leq \frac{1}{n} \sum_{i=1}^{n} \mathbb{E} \left[ \| \phi^*(\tilde{x}) - \psi^{\circ}_{\underline{N}:\bar{N}}(\tilde{x}) \|^2 \| \Gamma^* (\phi^*(x_i) - \psi^{\circ}_{\underline{N}:\bar{N}}(x_i)) \|^2 y_i^2 \right]
$$

$$
\leq \| \Gamma^* \|_{\mathrm{op}}^2 \left( \sup_{x \in \mathrm{supp} \, \mathcal{P}_{\mathcal{X}}} \| \phi^*(x) - \psi^{\circ}_{\underline{N}:\bar{N}}(x) \|^2 \right)^2 \mathbb{E} \left[ y^2 \right]
$$

$$
\leq \| \Gamma^* \|_{\mathrm{op}}^2 \left( N \max_{\underline{N} \leq j \leq \bar{N}} \| \phi^*_j - \psi^{\circ}_j \|_{L^\infty(\mathcal{P}_{\mathcal{X}})}^2 \right)^2 \mathbb{E} \left[ y^2 \right]
$$

$$
\lesssim N^2 \delta_N^4 (B^2 + \sigma^2).
$$

Similarly for (11) and (12), we have

$$
\mathbb{E} \left[ \left( (\phi^*(\tilde{x}) - \psi^{\circ}_{\underline{N}:\bar{N}}(\tilde{x}))^{\top} \Gamma^* \frac{\Psi^{\circ}_{\underline{N}:\bar{N}} \boldsymbol{y}}{n} \right)^2 \right]
$$

$$
\leq \frac{1}{n} \sum_{i=1}^{n} \mathbb{E} \left[ \left( (\phi^*(\tilde{x}) - \psi^{\circ}_{\underline{N}:\bar{N}}(\tilde{x}))^{\top} \Gamma^* \psi^{\circ}_{\underline{N}:\bar{N}}(x_i) y_i \right)^2 \right]
$$

$$
\leq \frac{1}{n} \sum_{i=1}^{n} \mathbb{E} \left[ \| \phi^*(\tilde{x}) - \psi^{\circ}_{\underline{N}:\bar{N}}(\tilde{x}) \|^2 \| \Gamma^* \psi^{\circ}_{\underline{N}:\bar{N}}(x_i) \|^2 y_i^2 \right]
$$

$$
\leq \sup_{x \in \mathrm{supp} \, \mathcal{P}_{\mathcal{X}}} \| \phi^*(x) - \psi^{\circ}_{\underline{N}:\bar{N}}(x) \|^2 \| \Gamma^* \|_{\mathrm{op}}^2 \sup_{x \in \mathrm{supp} \, \mathcal{P}_{\mathcal{X}}} \| \psi^{\circ}_{\underline{N}:\bar{N}}(x) \|^2 \mathbb{E} \left[ y^2 \right]
$$

$$
\lesssim N \delta_N^2 \cdot N^{2r} (B^2 + \sigma^2) = N^{2r+1} \delta_N^2 (B^2 + \sigma^2)
$$

and

$$
\mathbb{E} \left[ \left( \psi^{\circ}_{\underline{N}:\bar{N}}(\tilde{x})^{\top} \Gamma^* \frac{(\Phi^* - \Psi^{\circ}_{\underline{N}:\bar{N}}) \boldsymbol{y}}{n} \right)^2 \right]
$$

$$
\leq \frac{1}{n} \sum_{i=1}^{n} \mathbb{E} \left[ \left( \psi^{\circ}_{\underline{N}:\bar{N}}(\tilde{x})^{\top} \Gamma^* (\phi^*(x_i) - \psi^{\circ}_{\underline{N}:\bar{N}}(x_i)) y_i \right)^2 \right]
$$

$$\leq \frac{1}{n}\sum_{i=1}^{n}\mathbb{E}\left[\|\psi_{\underline{N}:\bar{N}}^{\circ}(\tilde{x})\|^2\|\Gamma^*(\phi^*(x_i)-\psi_{\underline{N}:\bar{N}}^{\circ}(x_i))\|^2 y_i^2\right]$$

$$\leq \sup_{x\in\operatorname{supp}\mathcal{P}_{\mathcal{X}}}\|\psi_{\underline{N}:\bar{N}}^{\circ}(x)\|^2\|\Gamma^*\|_{\operatorname{op}}^2 \sup_{x\in\operatorname{supp}\mathcal{P}_{\mathcal{X}}}\|(\phi^*(x)-\psi_{\underline{N}:\bar{N}}^{\circ}(x))\|^2\mathbb{E}\left[y^2\right]$$

$$\lesssim N^{2r+1}\delta_N^2(B^2+\sigma^2).$$

This concludes the proof of Proposition 3.2.

## A.5 Proof of Lemma A.1

We will make use of the following concentration bound and its corollary, proved at the end of this subsection.

**Theorem A.2** (matrix Bernstein inequality). *Let* $\mathbf{S}_1,\cdots,\mathbf{S}_n\in\mathbb{R}^{N\times N}$ *be independent random matrices such that* $\mathbb{E}[\mathbf{S}_i]=0$ *and* $\|\mathbf{S}_i\|_{\operatorname{op}}\leq L$ *almost surely for all* $i$. *Define* $\mathbf{Z}=\sum_{i=1}^{n}\mathbf{S}_n$ *and the matrix variance statistic* $v(\mathbf{Z})$ *as*

$$v(\mathbf{Z})=\max\left\{\|\mathbb{E}[\mathbf{Z}\mathbf{Z}^{\top}]\|_{\operatorname{op}},\|\mathbb{E}[\mathbf{Z}^{\top}\mathbf{Z}]\|_{\operatorname{op}}\right\}=\max\left\{\left\|\sum_{i=1}^{n}\mathbb{E}[\mathbf{S}_i\mathbf{S}_i^{\top}]\right\|_{\operatorname{op}},\left\|\sum_{i=1}^{n}\mathbb{E}[\mathbf{S}_i^{\top}\mathbf{S}_i]\right\|_{\operatorname{op}}\right\}.$$

*Then it holds for all* $t>0$ *that*

$$\Pr\left(\|\mathbf{Z}\|_{\operatorname{op}}\geq t\right)\leq 2N\exp\left(-\frac{t^2}{2v(\mathbf{Z})+\frac{2}{3}Lt}\right).$$

*Proof.* See Tropp (2015), Theorem 6.1.1. □

**Corollary A.3.** *For matrices* $\mathbf{S}_1,\cdots,\mathbf{S}_n$ *satisfying the conditions of Theorem A.2, it holds that*

$$\mathbb{E}\left[\frac{1}{n^2}\|\mathbf{Z}\|_{\operatorname{op}}^2\right]\leq\frac{4v(\mathbf{Z})}{n^2}(1+\log 2N)+\frac{16L^2}{9n^2}(2+2\log 2N+(\log 2N)^2).$$

We will apply Corollary A.3 to the matrices

$$\mathbf{S}_i:=\Sigma_{\Psi,N}^{-1/2}\psi_{\underline{N}:\bar{N}}^{\circ}(x_i)\psi_{\underline{N}:\bar{N}}^{\circ}(x_i)^{\top}\Sigma_{\Psi,N}^{-1/2}-\mathbf{I}_N.$$

It is straightforward to verify that

$$\mathbb{E}[\mathbf{S}_i]=\Sigma_{\Psi,N}^{-1/2}\Sigma_{\Psi,N}\Sigma_{\Psi,N}^{-1/2}-\mathbf{I}_N=0$$

and

$$\|\mathbf{S}_i\|_{\operatorname{op}}\lesssim\|\psi_{\underline{N}:\bar{N}}^{\circ}(x_i)\psi_{\underline{N}:\bar{N}}^{\circ}(x_i)^{\top}\|_{\operatorname{op}}+1=\sum_{j=\underline{N}}^{\bar{N}}\psi_j^{\circ}(x_i)^2+1\lesssim N^{2r}$$

almost surely by Assumption 1.

Next, we evaluate the matrix variance statistic. Since each $\mathbf{S}_i$ is symmetric,

$$v(\mathbf{Z})=\left\|\sum_{i=1}^{n}\mathbb{E}[\mathbf{S}_i\mathbf{S}_i^{\top}]\right\|_{\operatorname{op}}$$

$$=\left\|\sum_{i=1}^{n}\left(\mathbb{E}_{\mathbf{X}}\left[\Sigma_{\Psi,N}^{-1/2}\psi_{\underline{N}:\bar{N}}^{\circ}(x_i)\psi_{\underline{N}:\bar{N}}^{\circ}(x_i)^{\top}\Sigma_{\Psi,N}^{-1}\psi_{\underline{N}:\bar{N}}^{\circ}(x_i)\psi_{\underline{N}:\bar{N}}^{\circ}(x_i)^{\top}\Sigma_{\Psi,N}^{-1/2}\right]\right.\right.$$

$$\left.\left.-2\mathbb{E}_{\mathbf{X}}\left[\Sigma_{\Psi,N}^{-1/2}\psi_{\underline{N}:\bar{N}}^{\circ}(x_i)\psi_{\underline{N}:\bar{N}}^{\circ}(x_i)^{\top}\Sigma_{\Psi,N}^{-1/2}\right]+\mathbf{I}_N\right)\right\|_{\operatorname{op}}$$

$$=\left\|\sum_{i=1}^{n}\mathbb{E}_{\mathbf{X}}\left[\Sigma_{\Psi,N}^{-1/2}\psi_{\underline{N}:\bar{N}}^{\circ}(x_i)\psi_{\underline{N}:\bar{N}}^{\circ}(x_i)^{\top}\Sigma_{\Psi,N}^{-1}\psi_{\underline{N}:\bar{N}}^{\circ}(x_i)\psi_{\underline{N}:\bar{N}}^{\circ}(x_i)^{\top}\Sigma_{\Psi,N}^{-1/2}\right]-n\mathbf{I}_N\right\|_{\operatorname{op}}$$

$$\leq n+n\left\|\mathbb{E}_x\left[\Sigma_{\Psi,N}^{-1/2}\psi_{\underline{N}:\bar{N}}^{\circ}(x)\psi_{\underline{N}:\bar{N}}^{\circ}(x)^{\top}\Sigma_{\Psi,N}^{-1}\psi_{\underline{N}:\bar{N}}^{\circ}(x)\psi_{\underline{N}:\bar{N}}^{\circ}(x)^{\top}\Sigma_{\Psi,N}^{-1/2}\right]\right\|_{\operatorname{op}}$$

for a single sample $x \sim \mathcal{P}_{\mathcal{X}}$. The second term is further bounded as

$$\left\| \mathbb{E}_x \left[ \Sigma_{\Psi,N}^{-1/2} \psi_{\underline{N}:\bar{N}}^{\circ}(x) \psi_{\underline{N}:\bar{N}}^{\circ}(x)^{\top} \Sigma_{\Psi,N}^{-1} \psi_{\underline{N}:\bar{N}}^{\circ}(x) \psi_{\underline{N}:\bar{N}}^{\circ}(x)^{\top} \Sigma_{\Psi,N}^{-1/2} \right] \right\|_{\mathrm{op}}$$

$$\leq \left\| \mathbb{E}_x \left[ \Sigma_{\Psi,N}^{-1/2} \psi_{\underline{N}:\bar{N}}^{\circ}(x) \psi_{\underline{N}:\bar{N}}^{\circ}(x)^{\top} \Sigma_{\Psi,N}^{-1/2} \right] \right\|_{\mathrm{op}} \cdot \left\| \psi_{\underline{N}:\bar{N}}^{\circ\top} \Sigma_{\Psi,N}^{-1} \psi_{\underline{N}:\bar{N}}^{\circ} \right\|_{L^{\infty}(\mathcal{P}_{\mathcal{X}})}$$

$$\lesssim \| \mathbf{I}_N \|_{\mathrm{op}} \cdot \left\| \sum_{j=\underline{N}}^{\bar{N}} (\psi_j^{\circ})^2 \right\|_{L^{\infty}(\mathcal{P}_{\mathcal{X}})}$$

$$\lesssim N^{2r},$$

again by Assumption 1.

Hence we may apply Corollary A.3 with $v(\mathbf{Z}) \lesssim n N^{2r}$, $L \lesssim N^{2r}$ to conclude:

$$\mathbb{E}_{\mathbf{X}} \left[ \left\| \Sigma_{\Psi,N}^{-1/2} \frac{\Psi_{\underline{N}:\bar{N}}^{\circ} \Psi_{\underline{N}:\bar{N}}^{\circ\top}}{n} \Sigma_{\Psi,N}^{-1/2} - \mathbf{I}_N \right\|_{\mathrm{op}}^2 \right] = \mathbb{E} \left[ \left\| \frac{1}{n} \sum_{i=1}^n \mathbf{S}_i \right\|_{\mathrm{op}}^2 \right] \lesssim \frac{N^{2r}}{n} \log N + \frac{N^{4r}}{n^2} \log^2 N.$$

**Proof of Corollary A.3.** From Theorem A.2 and with the change of variables $\lambda = t^2/n^2$, we have

$$\Pr \left( \frac{1}{n^2} \| \mathbf{Z} \|_{\mathrm{op}}^2 \geq \lambda \right) \leq 2N \exp \left( - \frac{n^2 \lambda}{2v(\mathbf{Z}) + \frac{2}{3} Ln\sqrt{\lambda}} \right).$$

Since the probability is also bounded above by 1, it follows that

$$\Pr \left( \frac{1}{n^2} \| \mathbf{Z} \|_{\mathrm{op}}^2 \geq \lambda \right) \leq 1 \wedge 2N \exp \left( - \frac{n^2 \lambda}{2(2v(\mathbf{Z}) \vee \frac{2}{3} Ln\sqrt{\lambda})} \right)$$

$$\leq 1 \wedge 2N \exp \left( - \frac{n^2 \lambda}{4v(\mathbf{Z})} \right) + 1 \wedge 2N \exp \left( - \frac{3n\sqrt{\lambda}}{4L} \right),$$

and the expectation can be controlled as

$$\mathbb{E} \left[ \frac{1}{n^2} \| \mathbf{Z} \|_{\mathrm{op}}^2 \right] = \int_0^{\infty} \Pr \left( \frac{1}{n^2} \| \mathbf{Z} \|_{\mathrm{op}}^2 \geq \lambda \right) \mathrm{d}\lambda$$

$$\leq \int_0^{\infty} 1 \wedge 2N \exp \left( - \frac{n^2 \lambda}{4v(\mathbf{Z})} \right) \mathrm{d}\lambda + \int_0^{\infty} 1 \wedge 2N \exp \left( - \frac{3n\sqrt{\lambda}}{4L} \right) \mathrm{d}\lambda.$$

For the first integral, we truncate at $\lambda_1 := \frac{4v(\mathbf{Z})}{n^2} \log 2N$ so that

$$\int_0^{\infty} 1 \wedge 2N \exp \left( - \frac{n^2 \lambda}{4v(\mathbf{Z})} \right) \mathrm{d}\lambda = \lambda_1 + \int_{\lambda_1}^{\infty} 2N \exp \left( - \frac{n^2 \lambda}{4v(\mathbf{Z})} \right) \mathrm{d}\lambda$$

$$= \frac{4v(\mathbf{Z})}{n^2} \log 2N + \frac{4v(\mathbf{Z})}{n^2}.$$

For the second integral, we truncate at $\lambda_2 := \left( \frac{4L}{3n} \log 2N \right)^2$ so that

$$\int_0^{\infty} 1 \wedge 2N \exp \left( - \frac{3n\sqrt{\lambda}}{4L} \right) \mathrm{d}\lambda$$

$$= \lambda_2 + \int_{\lambda_2}^{\infty} 2N \exp \left( - \frac{3n\sqrt{\lambda}}{4L} \right) \mathrm{d}\lambda$$

$$= \lambda_2 - \frac{16LN}{3n} \left( \sqrt{\lambda} + \frac{4L}{3n} \right) \exp \left( - \frac{3n\sqrt{\lambda}}{4L} \right) \Big|_{\lambda=\lambda_2}^{\infty}$$

$$= \left( \frac{4L}{3n} \log 2N \right)^2 + \frac{8L}{3n} \left( \frac{4L}{3n} \log 2N + \frac{4L}{3n} \right).$$

Adding up, we conclude that

$$\mathbb{E} \left[ \frac{1}{n^2} \| \mathbf{Z} \|_{\mathrm{op}}^2 \right] \leq \frac{4v(\mathbf{Z})}{n^2} (1 + \log 2N) + \frac{16L^2}{9n^2} (2 + 2\log 2N + (\log 2N)^2)$$

as was to be shown. $\qquad \square$

# B Proofs on Metric Entropy

## B.1 Modified Proof of Theorem 3.1

For a full proof of the original statement, we refer the reader to Section B.1 of Hayakawa and Suzuki (2020), which corrects some technical flaws in the original proof. Here we only outline the necessary modification to incorporate the bounded noise setting.

Denote an $\epsilon$-cover of the model space by $f_1, \cdots, f_M$. The only step which relies on the normality of noise $\xi_{1:n}$ is a concentration result for the random variables

$$\varepsilon_j := \frac{\sum_{i=1}^n \xi_i(f_j(x_i) - f^\circ(x_i))}{\left[(\sum_{i=1}^n (f_j(x_i) - f^\circ(x_i))^2\right]^{1/2}}, \quad 1 \le j \le M,$$

where it is shown via the normality of $\varepsilon_j$ that

$$\mathbb{E}\left[\max_{1 \le j \le M} \varepsilon_j^2\right] \le 4\sigma^2 (\log M + 1).$$

We will instead rely on Hoeffding's inequality. By writing $\varepsilon_j = w_j^\top \xi_{1:n} = \sum_{i=1}^n w_{j,i}\xi_i$ where

$$w_{j,i} = \frac{f_j(x_i) - f^\circ(x_i)}{\left[(\sum_{i=1}^n (f_j(x_i) - f^\circ(x_i))^2\right]^{1/2}},$$

since $|w_{j,i}\xi_i| \le \sigma |w_{j,i}|$ a.s. it follows that

$$\Pr(|\varepsilon_j| \ge u) \le 2\exp\left(-\frac{2u^2}{\sum_{i=1}^n (2\sigma|w_{j,i}|)^2}\right) = 2\exp\left(-\frac{u^2}{2\sigma^2}\right).$$

for all $u > 0$. Then the squared-exponential moment of each $\varepsilon_j$ is bounded as

$$\begin{aligned}
\mathbb{E}\left[\exp(t\varepsilon_j^2)\right] &= 1 + \int_1^\infty \Pr\left(\exp(t\varepsilon_j^2) \ge \lambda\right) d\lambda \\
&\le 1 + \int_1^\infty 2\exp\left(-\frac{\log \lambda}{2\sigma^2 t}\right) d\lambda \\
&\le 1 + 2\int_1^\infty \lambda^{-\frac{1}{2\sigma^2 t}} d\lambda \le 3
\end{aligned}$$

by setting $t = \frac{1}{4\sigma^2}$. Hence via Jensen's inequality we obtain

$$\exp\left(t\mathbb{E}\left[\max_{1 \le j \le M} \varepsilon_j^2\right]\right) \le \mathbb{E}\left[\max_{1 \le j \le M} \exp(t\varepsilon_j^2)\right] \le \sum_{j=1}^M \mathbb{E}\left[\exp(t\varepsilon_j^2)\right] \le 3M$$

and thus

$$\mathbb{E}\left[\max_{1 \le j \le M} \varepsilon_j^2\right] \le 4\sigma^2 \log 3M,$$

retrieving the original result up to a constant. $\qquad\square$

## B.2 Proof of Lemma 3.3

Let us take two functions $f_{\Theta_1}, f_{\Theta_2} \in \mathcal{T}_N$ for $\Theta_i = (\Gamma_i, \phi_i)$, $i = 1, 2$ separated as

$$\|\Gamma_1 - \Gamma_2\|_{\mathrm{op}} \le \delta_1, \quad \max_{1 \le j \le N} \|\phi_{1,j} - \phi_{2,j}\|_{L^\infty(\mathcal{P}_\mathcal{X})} \le \delta_2.$$

Then it holds that

$$\begin{aligned}
&|f_{\Theta_1}(\mathbf{X}, \boldsymbol{y}, \tilde{x}) - f_{\Theta_2}(\mathbf{X}, \boldsymbol{y}, \tilde{x})| \\
&\le |\check{f}_{\Theta_1}(\mathbf{X}, \boldsymbol{y}, \tilde{x}) - \check{f}_{\Theta_2}(\mathbf{X}, \boldsymbol{y}, \tilde{x})| \\
&= \left|\left\langle \frac{\Gamma_1 \phi_1(\mathbf{X})\boldsymbol{y}}{n}, \phi_1(\tilde{x})\right\rangle - \left\langle \frac{\Gamma_2 \phi_2(\mathbf{X})\boldsymbol{y}}{n}, \phi_2(\tilde{x})\right\rangle\right|
\end{aligned}$$

$$
= \frac{1}{n} \Big| \phi_1(\tilde{x})^\top \Gamma_1 \phi_1(\mathbf{X})\boldsymbol{y} - \phi_2(\tilde{x})^\top \Gamma_1 \phi_1(\mathbf{X})\boldsymbol{y} + \phi_2(\tilde{x})^\top \Gamma_1 \phi_1(\mathbf{X})\boldsymbol{y}
$$
$$
\qquad - \phi_2(\tilde{x})^\top \Gamma_2 \phi_1(\mathbf{X})\boldsymbol{y} + \phi_2(\tilde{x})^\top \Gamma_2 \phi_1(\mathbf{X})\boldsymbol{y} - \phi_2(\tilde{x})^\top \Gamma_2 \phi_2(\mathbf{X})\boldsymbol{y} \Big|
$$
$$
\leq \frac{1}{n} \|\phi_1(\tilde{x}) - \phi_2(\tilde{x})\| \|\Gamma_1\|_{\mathrm{op}} \|\phi_1(\mathbf{X})\boldsymbol{y}\| + \frac{1}{n} \|\phi_2(\tilde{x})\| \|\Gamma_1 - \Gamma_2\|_{\mathrm{op}} \|\phi_1(\mathbf{X})\boldsymbol{y}\|
$$
$$
\qquad + \frac{1}{n} \|\phi_2(\tilde{x})\| \|\Gamma_2\|_{\mathrm{op}} \|(\phi_1(\mathbf{X}) - \phi_2(\mathbf{X}))\boldsymbol{y}\|
$$
$$
\leq \left( \frac{\sqrt{N}\delta_2 C_2}{n} + \frac{B_N' \delta_1}{n} \right) \sum_{i=1}^n \|\phi_1(x_i)\| |y_i| + \frac{B_N' C_2}{n} \sum_{i=1}^n \|\phi_1(x_i) - \phi_2(x_i)\| |y_i|
$$
$$
\leq (\sqrt{N}\delta_2 C_2 + B_N' \delta_1) B_N'(B + \sigma) + B_N' C_2 \sqrt{N}\delta_2 (B + \sigma)
$$
$$
= B_N'^2 (B + \sigma)\delta_1 + 2 B_N'(B + \sigma)C_2 \sqrt{N}\delta_2.
$$

Hence to construct an $\epsilon$-cover $G_\mathcal{T}$ of $\mathcal{T}_N$, it suffices to exhibit a $\delta_1$-cover $G_\mathcal{S}$ of $\mathcal{S}_N$ and a $\delta_2$-cover $G_\mathcal{F}$ of $\mathcal{F}_N$ for

$$
\delta_1 = \frac{\epsilon}{2 B_N'^2 (B + \sigma)}, \quad \delta_2 = \frac{\epsilon}{4 B_N'(B + \sigma)C_2\sqrt{N}}
$$

and set $G_\mathcal{T} = G_\mathcal{S} \times G_\mathcal{F}$. For the metric entropy, this implies

$$
\mathcal{V}(\mathcal{T}_N, \|\cdot\|_{L^\infty}, \epsilon) \leq \mathcal{V}(\mathcal{S}_N, \|\cdot\|_{\mathrm{op}}, \delta_1) + \mathcal{V}(\mathcal{F}_N, \|\cdot\|_{L^\infty}, \delta_2).
$$

We further bound the metric entropy of $\mathcal{S}_N$ with the following result, proved in Appendix B.3.

**Lemma B.1.** *For $\delta \leq \frac{1}{2}$ it holds that $\mathcal{V}(\mathcal{S}_N, \|\cdot\|_{\mathrm{op}}, \delta) \lesssim N^2 \log \frac{1}{\delta}$.*

Substituting into the above, we conclude that

$$
\mathcal{V}(\mathcal{T}_N, \|\cdot\|_{L^\infty}, \epsilon) \lesssim N^2 \log \frac{B_N'^2}{\epsilon} + \mathcal{V}\left( \mathcal{F}_N, \|\cdot\|_{L^\infty}, \frac{\epsilon}{4 B_N'(B + \sigma)C_2\sqrt{N}} \right).
$$

The choice of $\delta_2$ is not important as long as the metric entropy of $\mathcal{F}_N$ is at most polynomial in $\delta_2$. $\qquad \square$

## B.3 Proof of Lemma B.1

Let $\Gamma_1, \Gamma_2 \in \mathcal{S}_N$ and consider their diagonalizations

$$
\Gamma_i = \mathbf{U}_i \Lambda_i \mathbf{U}_i^\top, \quad \Lambda_i = \mathrm{diag}(\lambda_{i,1}, \cdots, \lambda_{i,N}), \quad i = 0, 1,
$$

where $\mathbf{U}_i \in \mathcal{O}_N$, the orthogonal group in dimension $N$, and $0 \leq \lambda_{i,j} \leq C_3$ for each $1 \leq j \leq N$. Assuming

$$
\|\mathbf{U}_1 - \mathbf{U}_2\|_{\mathrm{op}} \leq \frac{\delta}{4C_3}, \quad |\lambda_{1,j} - \lambda_{2,j}| \leq \frac{\delta}{2} \quad \forall j \leq N,
$$

it follows that

$$
\begin{aligned}
\|\Gamma_1 &- \Gamma_2\|_{\mathrm{op}} \\
&= \|\mathbf{U}_1 \Lambda_1 \mathbf{U}_1^\top - \mathbf{U}_1 \Lambda_1 \mathbf{U}_2^\top + \mathbf{U}_1 \Lambda_1 \mathbf{U}_2^\top - \mathbf{U}_2 \Lambda_1 \mathbf{U}_2^\top + \mathbf{U}_2 \Lambda_1 \mathbf{U}_2^\top - \mathbf{U}_2 \Lambda_2 \mathbf{U}_2^\top \|_{\mathrm{op}} \\
&\leq 2\|\Lambda_1\|_{L^\infty} \|\mathbf{U}_1 - \mathbf{U}_2\|_{\mathrm{op}} + \|\Lambda_1 - \Lambda_2\|_{L^\infty} \\
&\leq 2C_3 \cdot \frac{\delta}{4C_3} + \frac{\delta}{2} = \delta.
\end{aligned}
$$

Moreover, the covering number of $\mathcal{O}_N$ in operator norm is given by the following result.

**Theorem B.2** (Szarek (1981), Proposition 6). *There exist universal constants $c_1, c_2 > 0$ such that for all $N \in \mathbb{N}$ and $\delta \in (0, 2]$,*

$$
\left( \frac{c_1}{\delta} \right)^{\frac{N(N-1)}{2}} \leq \mathcal{N}(\mathcal{O}_N, \|\cdot\|_{\mathrm{op}}, \delta) \leq \left( \frac{c_2}{\delta} \right)^{\frac{N(N-1)}{2}}.
$$

Hence we obtain that

$$\mathcal{V}(\mathcal{S}_N, \|\cdot\|_{\mathrm{op}}, \delta) \leq \mathcal{V}\left(\mathcal{O}_N, \|\cdot\|_{\mathrm{op}}, \frac{\delta}{4C_3}\right) + \mathcal{V}\left([0, C_3]^N, \|\cdot\|_{L^\infty}, \frac{\delta}{2}\right)$$

$$\leq \frac{N(N-1)}{2} \log \frac{c_2}{\delta} + N \log \frac{2C_3}{\delta}$$

$$\lesssim N^2 \log \frac{1}{\delta}.$$

Finally, we remark that if elements of the domain $\mathcal{S}_N$ are not constrained to be symmetric, we can alternatively consider the singular value decomposition and separately bound entropy of the two rotation components, giving the same result up to constants. $\qquad\square$

## C  Details on Besov-type Spaces

### C.1  Besov Space

#### C.1.1  Verification of Assumptions

We first give some background on wavelet decomposition. The decay rate $s = \alpha/d$ is intrinsic to the Besov space as shown by the following result, which allows us to translate between functions $f \in B_{p,q}^\alpha(\mathcal{X})$ and their B-spline coefficient sequences.

**Lemma C.1** (DeVore and Popov (1988), Corollary 5.3)**.** *If $\alpha > d/p$ and $m > \alpha + 1 - 1/p$, a function $f \in L^p(\mathcal{X})$ is in $B_{p,q}^\alpha(\mathcal{X})$ if and only if $f$ can be represented as*

$$f = \sum_{k=0}^\infty \sum_{\ell \in I_k^d} \tilde{\beta}_{k,\ell} \omega_{k,\ell}^d$$

*such that the coefficients satisfy*

$$\|\tilde{\beta}\|_{b_{p,q}^\alpha} := \left[ \sum_{k=0}^\infty \left[ 2^{k(\alpha - d/p)} \left( \sum_{\ell \in I_k^d} |\tilde{\beta}_{k,\ell}|^p \right)^{1/p} \right]^q \right]^{1/q} < \infty,$$

*with appropriate modifications if $p = \infty$ or $q = \infty$. Moreover, the two norms $\|\tilde{\beta}\|_{b_{p,q}^\alpha}$ and $\|f\|_{B_{p,q}^\alpha}$ are equivalent.*

In particular, this implies that for any $f \in \mathbb{U}(B_{p,q}^\alpha(\mathcal{X}))$ the $p$-norm average of $\tilde{\beta}_{k,\ell}$ for $\ell \in I_k^d$ at resolution $k$ is bounded as

$$\left( \frac{1}{|I_k^d|} \sum_{\ell \in I_k^d} |\tilde{\beta}_{k,\ell}|^p \right)^{1/p} \lesssim (2^{-kd})^{1/p} \cdot 2^{k(d/p - \alpha)} \|f\|_{B_{p,q}^\alpha} \leq 2^{-k\alpha},$$

and the coefficients $\beta_{k,\ell} = 2^{-kd/2} \tilde{\beta}_{k,\ell}$ w.r.t. the scaled basis $(2^{kd/2} \omega_{k,\ell}^d)_{k,\ell}$ satisfy

$$\left( \frac{1}{|I_k^d|} \sum_{\ell \in I_k^d} |\beta_{k,\ell}|^p \right)^{1/p} \lesssim (2^{kd})^{-\alpha/d - 1/2}. \tag{18}$$

Thus it is natural in a probabilistic sense to assume $\mathbb{E}[\beta_{k,\ell}^p]^{1/p} \lesssim (2^{kd})^{-\alpha/d - 1/2}$. This will be the case if for instance we sample $(\beta_{k,\ell})_{\ell \in I_k^d}$ uniformly from the $p$-norm ball (18). This matches Assumption 4 up to the logarithmic factor and hence the rate is nearly tight in variance, even though (18) only applies to the average over locations rather than each coefficient explicitly. See also the discussion in Lemma 2 of Suzuki (2019).

**Assumption 1.** We take $m$ to be even for simplicity. The wavelet system $(\omega_{K,\ell}^d)_{\ell \in I_K^d}$ at each resolution $K$ is linearly independent; for any $g \in L^2(\mathcal{X})$ that can be expressed as

$$g = \sum_{\ell \in I_K^d} \beta_{K,\ell} 2^{Kd/2} \omega_{K,\ell}^d$$

we have the quasi-norm equivalence (Dũng, 2011b, 2.15)

$$\|g\|_2 \asymp 2^{-Kd/2} \left( \sum_{\ell \in I_K^d} 2^{Kd} \beta_{K,\ell}^2 \right)^{1/2} = \|(\beta_{K,\ell})_{\ell \in I_K^d}\|_2$$

which implies that the covariance matrix $\mathbb{E}_{x \sim \mathrm{Unif}([0,1]^d)}[\psi_{N:\bar{N}}^\circ(x)\psi_{N:\bar{N}}^\circ(x)^\top]$ is bounded above and below. Since we assume $\mathcal{P}_\mathcal{X}$ has Lebesgue density bounded above and below, it follows that $\Sigma_{\Psi,N}$ is uniformly bounded above and below for all $K \geq 0$.

In contrast, any B-spline at a lower resolution $k < K$ can be exactly expressed as a linear combination of elements of $(\omega_{K,\ell}^d)_{\ell \in I_K^d}$ by repeatedly applying the following relation.

**Lemma C.2** (refinement equation). *For even $m$ and $r = (r_1, \cdots, r_d)^\top$, $\mathbf{1} = (1, \cdots, 1)^\top \in \mathbb{R}^d$ it holds that*

$$\omega_{k,\ell}^d = \sum_{r_1,\cdots,r_d=0}^m 2^{(-m+1)d} \prod_{i=1}^d \binom{m}{r_i} \cdot \omega_{k+1,2\ell+r-\frac{m}{2}\mathbf{1}}^d. \tag{19}$$

*Proof.* The relation for one-dimensional wavelets is given in equation (2.21) of Dũng (2011b),

$$\iota_m(x) = 2^{-m+1} \sum_{r=0}^m \binom{m}{r} \iota_m \left( 2x - r + \frac{m}{2} \right),$$

from which it follows that

$$\begin{aligned}
\omega_{k,\ell}^d(x) &= \prod_{i=1}^d \iota_m(2^k x_i - \ell_i) \\
&= \sum_{r_1,\cdots,r_d=0}^m 2^{(-m+1)d} \prod_{i=1}^d \binom{m}{r_i} \iota_m \left( 2^{k+1} x_i - 2\ell_i - r_i + \frac{m}{2} \right) \\
&= \sum_{r_1,\cdots,r_d=0}^m 2^{(-m+1)d} \prod_{i=1}^d \binom{m}{r_i} \cdot \omega_{k+1,2\ell+r-\frac{m}{2}\mathbf{1}}^d(x)
\end{aligned}$$

as was to be shown. $\qquad\square$

Therefore we select all B-splines at a fixed resolution $K$ to approximate the target tasks,

$$N = |I_K^d| = (m + 1 + 2^K)^d \asymp 2^{Kd}$$

and

$$\underline{N} = \sum_{k=0}^{K-1} |I_k^d| + 1 \asymp N, \quad \bar{N} = \sum_{k=0}^K |I_k^d| \asymp N.$$

It is straightforward to see that $0 \leq \omega_{k,\ell}^d(x) \leq 1$ for all $x \in \mathcal{X}$ and moreover the B-splines (extended to all $\ell \in \mathbb{Z}^d$) form a partition of unity of $\mathbb{R}^d$ at all resolutions:

$$\sum_{\ell \in \mathbb{Z}^d} \omega_{k,\ell}^d(x) \equiv 1, \quad \forall x \in \mathbb{R}^d, \quad \forall k \geq 0.$$

Then for all $x \in \mathcal{X}$ we have the bound

$$\sum_{j=\underline{N}}^{\bar{N}} \psi_j^\circ(x)^2 = \sum_{\ell \in I_K^d} 2^{Kd} \omega_{K,\ell}^d(x)^2 \leq 2^{Kd} \sum_{\ell \in \mathbb{Z}^d} \omega_{K,\ell}^d(x) = 2^{Kd} \lesssim N,$$

and hence (2) holds with $r = 1/2$.

**Assumption 2.** For any $\beta \in \operatorname{supp} \mathcal{P}_\beta$ we have that

$$
\begin{aligned}
\|F_\beta^\circ\|_{L^\infty(\mathcal{P}_\mathcal{X})} &\leq \sum_{k=0}^\infty \left\| \sum_{\ell \in I_k^d} \beta_{k,\ell} \cdot 2^{kd/2} \omega_{k,\ell}^d \right\|_{L^\infty(\mathcal{P}_\mathcal{X})} \\
&\leq \sum_{k=0}^\infty 2^{kd/2} \max_{\ell \in I_k^d} |\beta_{k,\ell}| \cdot \left\| \sum_{\ell \in I_k^d} \omega_{k,\ell}^d \right\|_{L^\infty(\mathcal{X})} \\
&\leq \sum_{k=0}^\infty 2^{kd(1/2+1/p)} \left( \frac{1}{|I_k^d|} \sum_{\ell \in I_k^d} |\beta_{k,\ell}|^p \right)^{1/p} \left\| \sum_{\ell \in I_k^d} \omega_{k,\ell}^d \right\|_{L^\infty(\mathcal{X})} \\
&\lesssim \sum_{k=0}^\infty 2^{kd(1/2+1/p)} \cdot (2^{kd})^{-\alpha/d-1/2} \|F_\beta^\circ\|_{B_{p,q}^\alpha} \\
&\lesssim (1 - 2^{d/p-\alpha})^{-1} =: B.
\end{aligned}
$$

Furthermore, the convergence rate of the truncated approximation $F_{\beta,\bar{N}}^\circ$ is determined by the decay rate of $\beta$ in Lemma C.1 as follows (it does not matter whether we bound $F_{\beta,\bar{N}}^\circ$ or $F_{\beta,N}^\circ$ since $\bar{N} \asymp N$). We consider a truncation of all resolutions lower than $K$ so that $\bar{N}, N \asymp 2^{Kd}$. Then it holds that

$$
\begin{aligned}
\|F_\beta^\circ - F_{\beta,\bar{N}}^\circ\|_{L^2(\mathcal{P}_\mathcal{X})}^2 &= \int_\mathcal{X} \left( \sum_{j=\bar{N}+1}^\infty \beta_j \psi_j^\circ(x) \right)^2 \mathcal{P}_\mathcal{X}(\mathrm{d}x) = \sum_{j,k=\bar{N}+1}^\infty \beta_j \beta_k \mathbb{E}_x[\psi_j^\circ(x)\psi_k^\circ(x)] \\
&\leq \lim_{M\to\infty} C_2 \|\beta_{\bar{N}:M}\|^2 = C_2 \sum_{k=K}^\infty \sum_{\ell \in I_k^d} \beta_{k,\ell}^2 \\
&\lesssim \sum_{k=K}^\infty |I_k^d|^{1-2/p} \left( \sum_{\ell \in I_k^d} |\beta_{k,\ell}|^p \right)^{2/p} \lesssim \sum_{k=K}^\infty 2^{kd} \cdot (2^{kd})^{-2\alpha/d-1} \\
&= \frac{2^{-2\alpha K}}{1 - 2^{-2\alpha}} \asymp N^{-2\alpha/d},
\end{aligned}
$$

where for the last two inequalities we have used the inequality $\|z\|_2 \leq D^{1/2-1/p}\|z\|_p$ for $z \in \mathbb{R}^D$ and $p \geq 2$ in conjunction with (18). Thus our choice of $s = \alpha/d$ is justified. Under this choice, (5) directly implies (4) as

$$
\mathbb{E}_\beta[\beta_{K,\ell}^2] \lesssim 2^{-Kd(2s+1)} K^{-2} \asymp \bar{N}^{-2s-1} (\log \bar{N})^{-2}
$$

holds for the basis elements at each resolution $K$, that is for those numbered between $N$ and $\bar{N}$.

**Remark C.3.** When $1 \leq p < 2$, the truncation up to $\bar{N}$ does not suffice to achieve the $N^{-2\alpha/d}$ approximation rate, and basis elements must be judiciously selected from a wider resolution range. More concretely, a size $N$ subset of all wavelets up to resolution $K' = \lceil K(1 + \nu^{-1}) \rceil$ where $\nu = p\alpha/2d - 1/2 > 0$ must be used (Suzuki and Nitanda, 2021, Lemma 2). Hence the exponent is a factor of $1 + \nu^{-1}$ worse w.r.t. $N' \asymp 2^{K'd}$, leading to the inevitable suboptimal rate.

To show boundedness of $\operatorname{Tr}(\Sigma_{\bar{\beta},N})$, we analyze the composition of the aggregated coefficients $\bar{\beta}$ using the following result.

**Corollary C.4.** *For any $0 \leq k < k'$ there exists constants $\gamma_{k,k',\ell,\ell'} \geq 0$ for $\ell \in I_k^d$, $\ell' \in I_{k'}^d$ such that*

$$
\sum_{\ell \in I_k^d} \beta_{k,\ell} 2^{kd/2} \omega_{k,\ell}^d = \sum_{\ell' \in I_{k'}^d} \bar{\beta}_{k',\ell'} 2^{k'd/2} \omega_{k',\ell'}^d, \quad \bar{\beta}_{k',\ell'} = \sum_{\ell \in I_k^d} \gamma_{k,k',\ell,\ell'} \beta_{k,\ell}
$$

*holds for all $(\beta_{k,\ell})_{\ell \in I_k^d}$. Moreover, it holds that*

$$
\sum_{\ell \in I_k^d} \gamma_{k,k',\ell,\ell'} \leq 2^{(k-k')d/2}, \quad \sum_{\ell' \in I_{k'}^d} \gamma_{k,k',\ell,\ell'} \leq 2^{(k'-k)d/2}.
$$

The statement follows directly from the more general Proposition C.10, stated and proved in Appendix C.3 below, by restricting to wavelets with uniform resolution across dimensions. Using Corollary C.4, we can refine each lower resolution component of $F_\beta^\circ$ to resolution $K$:

$$F_{\beta,\bar{N}}^\circ = \sum_{j=1}^{\bar{N}} \beta_j \psi_j^\circ = \sum_{k=0}^{K} \sum_{\ell \in I_k^d} \beta_{k,\ell} 2^{kd/2} \omega_{k,\ell}^d = \sum_{k=0}^{K} \sum_{\ell' \in I_K^d} \sum_{\ell \in I_k^d} \gamma_{k,K,\ell,\ell'} \beta_{k,\ell} 2^{Kd/2} \omega_{K,\ell'}^d.$$

Thus each aggregated coefficient, indexed here by $\ell \in I_K^d$, can be expressed as

$$\bar{\beta}_{K,\ell} = \sum_{k=0}^{K} \sum_{\ell \in I_k^d} \gamma_{k,K,\ell,\ell'} \beta_{k,\ell}.$$

Hence it follows that

$$\mathbb{E}_\beta[\bar{\beta}_{K,\ell}^2] = \sum_{k=0}^{K} \sum_{\ell \in I_k^d} \gamma_{k,K,\ell,\ell'}^2 \mathbb{E}_\beta[\beta_{k,\ell}^2] \lesssim \sum_{k=0}^{K} \left( \sum_{\ell \in I_k^d} \gamma_{k,K,\ell,\ell'} \right)^2 2^{-k(2\alpha+d)} k^{-2}$$

$$\leq \sum_{k=0}^{K} 2^{(k-K)d} \cdot 2^{-k(2\alpha+d)} k^{-2}$$

$$\lesssim 2^{-Kd} \cdot \sum_{k=0}^{K} 2^{-2k\alpha} k^{-2} \asymp N^{-1},$$

from which we conclude that $\mathrm{Tr}(\Sigma_{\bar{\beta},N}) = \sum_{\ell \in I_K^d} \mathbb{E}_\beta[\bar{\beta}_{K,\ell}^2] \lesssim N \cdot N^{-1}$ is uniformly bounded.

Finally for the verification of Assumption 3, see Appendix C.1.2.

### C.1.2  Proof of Lemma 4.4

We use the following result to approximate each wavelet $\omega_{K,\ell}^d$ at resolution $K$ with the class (6). The proof, in turn, relies on the construction by Yarotsky (2016) of DNNs which efficiently approximates the multiplication operation.

**Lemma C.5** (Suzuki (2019), Lemma 1). *For all $\delta > 0$, there exists a ReLU neural network $\tilde{\omega} \in \mathcal{F}_{\mathrm{DNN}}(L, W, S, M)$ with*

$$L = 3 + 2 \left\lceil \log_2 \left( 3^{d \vee m} (1 + dm^{-1/2}(2e)^{m+1}) \delta^{-1} \right) + 5 \right\rceil \lceil \log_2(d \vee m) \rceil,$$

$$W = W_0 = 6dm(m+2) + 2d, \qquad S = LW^2, \qquad M = 2(m+1)^m$$

*satisfying* $\mathrm{supp}\, \tilde{\omega} \subseteq [0, m+1]^d$ *and* $\|\omega_{0,0}^d - \tilde{\omega}\|_{L^\infty(\mathcal{X})} \leq \delta$.

Here, $\delta$ is also dependent on $N$.

Now consider $N$ identical copies of $\tilde{\omega}$ in parallel, where each module is preceded by the scaling $(x_i)_{i=1}^d \mapsto (2^K x_i - \ell_i)_{i=1}^d$ for $\ell \in I_K^d$ and whose output is scaled by $2^{Kd/2}$. In particular, these operations can be implemented by $K \lesssim \log N$ consecutive additional layers with norm bounded by a constant. Hence each module $\phi_{\underline{N}}^*, \cdots, \phi_{\bar{N}}^*$ approximates the basis $2^{Kd/2} \omega_{K,\ell}^d$ with $2^{Kd/2}\delta \lesssim \sqrt{N}\delta$ accuracy, and substituting $\delta_N = \sqrt{N}\delta$ gives that

$$\|\psi_j^\circ - \phi_j^*\|_{L^\infty(\mathcal{P}_\mathcal{X})} \leq \delta_N, \quad \underline{N} \leq j \leq \bar{N},$$

with $L \lesssim \log \delta^{-1} + \log N \lesssim \log \delta_N^{-1} + \log N$. Note that the sparsity $S$ is only multiplied by a factor of $N$ since different modules do not share any connections. Moreover the target basis has 2-norm bounded as

$$\|\psi_{\underline{N}:\bar{N}}^\circ(x)\|_2 \lesssim \left( \sum_{\ell \in I_K^d} 2^{Kd} \omega_{K,\ell}(x)^2 \right)^{1/2} \leq \left( 2^{Kd} \sum_{j \in \mathbb{Z}^d} \omega_{K,\ell}(x) \right)^{1/2} \asymp \sqrt{N},$$

where we have again used the sparsity of $\omega_{k,\ell}^d$ at each resolution. Hence we may clip the magnitude of the vector output $\phi$ by $B_N'$ and the approximation guarantee remains unchanged.

To bound the covering number of $\mathcal{F}_N$, we directly apply the following result.

**Lemma C.6** ([Suzuki (2019)](), Lemma 3). *The covering number of $\mathcal{F}_{\mathrm{DNN}}$ is bounded as*

$$\mathcal{N}(\mathcal{F}_{\mathrm{DNN}}(L, W, S, M), \|\cdot\|_{L^\infty}, \epsilon) \leq \left( \frac{L(M \vee 1)^{L-1}(W+1)^{2L}}{\epsilon} \right)^S.$$

Since clipping the magnitude of the outputs does not increase the covering number, we conclude:

$$
\begin{aligned}
\mathcal{V}(\mathcal{F}_N, \|\cdot\|_{L^\infty}, \epsilon) &\leq N \cdot \mathcal{V}(\mathcal{F}_{\mathrm{DNN}}(L, W, S, M), \|\cdot\|_{L^\infty}, \epsilon) \\
&\leq SN \log L + SLN \log M + 2SLN \log(W+1) + SN \log \frac{1}{\epsilon} \\
&\lesssim N \log \frac{N}{\delta_N} + N \log \frac{1}{\epsilon}.
\end{aligned}
$$

$\square$

### C.1.3 Proof of Theorem 4.5

By Lemma 3.3 and Lemma 4.4, the metric entropy of $\mathcal{T}_N$ is bounded as

$$\mathcal{V}(\mathcal{T}_N, \|\cdot\|_{L^\infty}, \epsilon) \lesssim N^2 \log \frac{N}{\epsilon} + N \log \frac{N^2}{\delta_N \epsilon}.$$

Combining with Theorem 3.1 and Proposition 3.2 with $r = 1/2$ and $s = \alpha/d$ gives

$$\bar{R}(\widehat{\Theta}) \lesssim \frac{N}{n} \log N + \frac{N^2}{n^2} \log^2 N + N^{-2\alpha/d} + N^2 \delta_N^2 + \frac{1}{T} \left( N^2 \log \frac{N}{\epsilon} + N \log \frac{N^2}{\delta_N \epsilon} \right) + \epsilon.$$

Substituting $\delta_N \asymp N^{-1-\alpha/d}$ and $\epsilon \asymp N^{-2\alpha/d}$ yields the desired bound. $\square$

## C.2 Anisotropic Besov Space

### C.2.1 Definitions and Results

For $1 \leq p, q \leq \infty$, directional smoothness $\alpha = (\alpha_1, \cdots, \alpha_d) \in \mathbb{R}_{>0}^d$ and $r = \max_{i \leq d} \lfloor \alpha_i \rfloor + 1$, we define $\|\cdot\|_{B_{p,q}^\alpha} = \|\cdot\|_{L^p} + |\cdot|_{B_{p,q}^\alpha}$ where

$$|f|_{B_{p,q}^\alpha} := \begin{cases} \left( \sum_{k=0}^\infty \left[ 2^k w_{r,p}(f, (2^{-k/\alpha_1}, \cdots, 2^{-k/\alpha_d})) \right]^q \right)^{1/q} & q < \infty \\ \sup_{k \geq 0} 2^k w_{r,p}(f, (2^{-k/\alpha_1}, \cdots, 2^{-k/\alpha_d})) & q = \infty. \end{cases}$$

The anisotropic Besov space is defined as

$$B_{p,q}^\alpha(\mathcal{X}) = \{ f \in L^p(\mathcal{X}) \mid \|f\|_{B_{p,q}^\alpha} < \infty \}.$$

The definition reduces to the usual Besov space if $\alpha_1 = \cdots = \alpha_d$; see [Vybíral (2006)](); [Triebel (2011)]() for details. We also write $\overline{\alpha} = \max_i \alpha_i$, $\underline{\alpha} = \min_i \alpha_i$ and the harmonic mean smoothness as

$$\widetilde{\alpha} := \left( \sum_{i=1}^d \alpha_i^{-1} \right)^{-1}.$$

For the anisotropic Besov space, we need to redefine the wavelet basis so that the sensitivity to resolution $k \in \mathbb{Z}_{\geq 0}$ differs for each component depending on $\alpha$. Define the quantities

$$\|k\|_{\underline{\alpha}/\alpha} := \sum_{i=1}^d \lfloor k\underline{\alpha}/\alpha_i \rfloor, \quad I_k^{d,\alpha} := \prod_{i=1}^d \{-m, -m+1, \cdots, 2^{\lfloor k\underline{\alpha}/\alpha_i \rfloor}\} \subset \mathbb{Z}^d.$$

We then set for each $k \geq 0$ and $\ell \in I_k^{d,\alpha}$

$$\omega_{k,\ell}^{d,\alpha}(x) := \omega_{(\lfloor k\underline{\alpha}/\alpha_1 \rfloor, \cdots, \lfloor k\underline{\alpha}/\alpha_d \rfloor), \ell}^d(x) = \prod_{i=1}^d \iota_m(2^{\lfloor k\underline{\alpha}/\alpha_i \rfloor} x_i - \ell_i),$$

and take the scaled basis

$$\{\psi_j^\circ \mid j \in \mathbb{N}\} = \{2^{\|k\|_{\underline{\alpha}/\alpha}/2} \omega_{k,\ell}^{d,\alpha} \mid k \in \mathbb{Z}_{\geq 0}, \ell \in I_k^{d,\alpha}\}$$

with the natural hierarchy induced by $k$.

The minimax optimal rate for the anisotropic Besov space is equal to $n^{-\frac{2\widetilde{\alpha}}{2\widetilde{\alpha}+1}}$ (Suzuki and Nitanda, 2021, Theorem 4). Our result for in-context learning is as follows.

**Theorem C.7** (minimax optimality of ICL in anisotropic Besov space). *Let $\alpha \in \mathbb{R}_{>0}^d$ with $\widetilde{\alpha} > 1/p$ and $\mathcal{F}^\circ = \mathbb{U}(B_{p,q}^\alpha(\mathcal{X}))$. Suppose that $\mathcal{P}_\mathcal{X}$ has positive Lebesgue density $\rho_\mathcal{X}$ bounded above and below on $\mathcal{X}$. Also suppose that all coefficients are independent and*

$$\mathbb{E}_\beta[\beta_{k,\ell}] = 0, \quad \mathbb{E}_\beta[\beta_{k,\ell}^2] \lesssim 2^{-k\underline{\alpha}(2+1/\widetilde{\alpha})}k^{-2}, \quad \forall k \geq 0, \ \ell \in I_k^{d,\alpha}. \tag{20}$$

*Then for $n \gtrsim N \log N$ we have*

$$\bar{R}(\widehat{\Theta}) \lesssim N^{-2\widetilde{\alpha}} + \frac{N \log N}{n} + \frac{N^2 \log N}{T}.$$

*Hence if $T \gtrsim nN$ and $N \asymp n^{\frac{1}{2\widetilde{\alpha}+1}}$, in-context learning achieves the rate $n^{-\frac{2\widetilde{\alpha}}{2\widetilde{\alpha}+1}} \log n$ which is minimax optimal up to a polylog factor.*

### C.2.2 Proof of Theorem C.7

The overall approach is similar to Appendix C.1. The decay rate of functions in the anisotropic Besov space is characterized by the following result which extends Lemma C.1.

**Lemma C.8** (Suzuki and Nitanda (2021), Lemma 2). *If $\widetilde{\alpha} > 1/p$ and $m > \overline{\alpha} + 1 - 1/p$, a function $f \in L^p(\mathcal{X})$ is in $MB_{p,q}^\alpha(\mathcal{X})$ if and only if $f$ can be represented as*

$$f = \sum_{k \in \mathbb{Z}_{\geq 0}^d} \sum_{\ell \in I_k^{d,\alpha}} \tilde{\beta}_{k,\ell} \omega_{k,\ell}^{d,\alpha}(x)$$

*such that the coefficients satisfy*

$$\|\tilde{\beta}\|_{b_{p,q}^\alpha} := \left[ \sum_{k=0}^\infty \left[ 2^{k\underline{\alpha} - \|k\|_{\underline{\alpha}/\alpha}/p} \left( \sum_{\ell \in I_k^{d,\alpha}} |\tilde{\beta}_{k,\ell}|^p \right)^{1/p} \right]^q \right]^{1/q} \lesssim \|f\|_{B_{p,q}^\alpha}.$$

*Moreover, the two norms $\|\tilde{\beta}\|_{b_{p,q}^\alpha}$ and $\|f\|_{B_{p,q}^\alpha}$ are equivalent.*

We again select all B-splines $(\omega_{K,\ell}^{d,\alpha})_{\ell \in I_K^d}$ at each resolution $K$ to approximate the target functions. By repeatedly applying the refinement equation for one-dimensional wavelets as many times as needed to each dimension separately, we may express any B-spline at a lower resolution $k < K$ as a linear combination of $(\omega_{K,\ell}^{d,\alpha})_{\ell \in I_K^d}$ similarly to Lemma C.2. See Proposition C.10 for details. We thus have

$$N = |I_K^{d,\alpha}| = \prod_{i=1}^d (m + 1 + 2^{\lfloor K\underline{\alpha}/\alpha_i \rfloor}) \asymp 2^{\|K\|_{\underline{\alpha}/\alpha}}.$$

Since $\|k\|_{\underline{\alpha}/\alpha} = k\underline{\alpha}/\widetilde{\alpha} + O_k(1)$ always holds, it also follows that

$$\bar{N} = \sum_{k=0}^K |I_k^{d,\alpha}| + 1 \asymp \sum_{k=0}^K 2^{\|k\|_{\underline{\alpha}/\alpha}} \asymp \sum_{k=0}^K (2^{\underline{\alpha}/\widetilde{\alpha}})^k \asymp 2^{K\underline{\alpha}/\widetilde{\alpha}} \asymp N$$

and similarly $\underline{N} \asymp N$. Therefore,

$$\sum_{j=\underline{N}}^{\bar{N}} \psi_j^\circ(x)^2 \leq 2^{\|K\|_{\underline{\alpha}/\alpha}} \sum_{\ell \in I_k^{d,\alpha}} \omega_{K,\ell}^{d,\alpha}(x)^2 \leq 2^{\|K\|_{\underline{\alpha}/\alpha}} \asymp N$$

and the scaled coefficients decay in average as

$$\left(\frac{1}{|I_k^{d,\alpha}|}\sum_{\ell\in I_k^{d,\alpha}}|\beta_{k,\ell}|^p\right)^{1/p} \lesssim \left(\prod_{i=1}^d 2^{\lfloor k\underline{\alpha}/\alpha_i\rfloor}\right)^{-1/p} 2^{-\|k\|_{\underline{\alpha}/\alpha}/2}\left(\sum_{\ell\in I_k^{d,\alpha}}|\tilde\beta_{k,\ell}|^p\right)^{1/p}$$

$$\lesssim 2^{-\|k\|_{\underline{\alpha}/\alpha}/2-k\underline{\alpha}}\|f\|_{B_{p,q}^\alpha}$$

$$\asymp N^{-(\tilde\alpha+1/2)}\|f\|_{B_{p,q}^\alpha}.$$

For Assumption 2, we can check that

$$\|F_\beta^\circ\|_{L^\infty(\mathcal{P}_\mathcal{X})} \leq \sum_{k=0}^\infty\left\|\sum_{\ell\in I_k^{d,\alpha}}\beta_{k,\ell}\cdot 2^{\|k\|_{\underline{\alpha}/\alpha}/2}\omega_{k,\ell}^{d,\alpha}\right\|_{L^\infty(\mathcal{P}_\mathcal{X})}$$

$$\leq \sum_{k=0}^\infty 2^{(1/2+1/p)\|k\|_{\underline{\alpha}/\alpha}}\left(\frac{1}{|I_k^{d,\alpha}|}\sum_{\ell\in I_k^{d,\alpha}}|\beta_{k,\ell}|^p\right)^{1/p}$$

$$\lesssim \sum_{k=0}^\infty 2^{(1/2+1/p)\|k\|_{\underline{\alpha}/\alpha}}\cdot 2^{-\|k\|_{\underline{\alpha}/\alpha}/2-k\underline{\alpha}}$$

$$\lesssim \left(1-2^{\underline{\alpha}/\tilde\alpha(1/p-\tilde\alpha)}\right)^{-1} =: B$$

and for a resolution cutoff $K>0$, $\bar N\asymp 2^{\|K\|_{\underline{\alpha}/\alpha}}$ the truncation error satisfies

$$\|F_\beta^\circ - F_{\beta,\bar N}^\circ\|_{L^2(\mathcal{P}_\mathcal{X})}^2 \lesssim \sum_{k=K+1}^\infty\sum_{\ell\in I_k^{d,\alpha}}\beta_{k,\ell}^2 \lesssim \sum_{k=K+1}^\infty |I_k^{d,\alpha}|^{1-2/p}\left(\sum_{\ell\in I_k^{d,\alpha}}|\beta_{k,\ell}|^p\right)^{2/p}$$

$$\lesssim \sum_{k=K+1}^\infty 2^{\|k\|_{\underline{\alpha}/\alpha}}\cdot 2^{-\|k\|_{\underline{\alpha}/\alpha}-2k\underline{\alpha}} \asymp 2^{-2K\underline{\alpha}} \asymp N^{-2\tilde\alpha}.$$

Thus we may set $r=1/2$, $s=\tilde\alpha$ and take the variance decay rate (20) as

$$\mathbb{E}_\beta[\beta_{k,\ell}^2] \lesssim 2^{-\|k\|_{\underline{\alpha}/\alpha}(2\tilde\alpha+1)}k^{-2} \asymp 2^{-k\underline{\alpha}(2+1/\tilde\alpha)}k^{-2}.$$

For boundedness of $\mathrm{Tr}(\Sigma_{\bar\beta,N})$, we use the following result which is also obtained from Proposition C.10 by considering resolution vectors $(\lfloor k\underline{\alpha}/\alpha_1\rfloor,\cdots,\lfloor k\underline{\alpha}/\alpha_d\rfloor)$ and $(\lfloor k'\underline{\alpha}/\alpha_1\rfloor,\cdots,\lfloor k'\underline{\alpha}/\alpha_d\rfloor)$.

**Corollary C.9.** *For any $0\leq k<k'$ there exists constants $\gamma_{k,k',\ell,\ell'}\geq 0$ for $\ell\in I_k^{d,\alpha}$, $\ell'\in I_{k'}^{d,\alpha}$ such that*

$$\sum_{\ell\in I_k^{d,\alpha}}\beta_{k,\ell}2^{\|k\|_{\underline{\alpha}/\alpha}/2}\omega_{k,\ell}^{d,\alpha} = \sum_{\ell'\in I_{k'}^{d,\alpha}}\bar\beta_{k',\ell'}2^{\|k'\|_{\underline{\alpha}/\alpha}/2}\omega_{k',\ell'}^{d,\alpha}, \quad \bar\beta_{k',\ell'} = \sum_{\ell\in I_k^{d,\alpha}}\gamma_{k,k',\ell,\ell'}\beta_{k,\ell}$$

*holds for all $(\beta_{k,\ell})_{\ell\in I_k^{d,\alpha}}$. Moreover, it holds that*

$$\sum_{\ell\in I_k^{d,\alpha}}\gamma_{k,k',\ell,\ell'} \leq 2^{(\|k\|_{\underline{\alpha}/\alpha}-\|k'\|_{\underline{\alpha}/\alpha})/2}, \quad \sum_{\ell'\in I_{k'}^{d,\alpha}}\gamma_{k,k',\ell,\ell'} \leq 2^{(\|k'\|_{\underline{\alpha}/\alpha}-\|k\|_{\underline{\alpha}/\alpha})/2}.$$

We apply Corollary C.9 to refine all components of $F_{\beta,\bar N}^\circ$ to resolution $K$:

$$F_{\beta,\bar N}^\circ = \sum_{k=0}^K\sum_{\ell\in I_k^{d,\alpha}}\beta_{k,\ell}2^{\|k\|_{\underline{\alpha}/\alpha}/2}\omega_{k,\ell}^{d,\alpha} = \sum_{k=0}^K\sum_{\ell'\in I_K^{d,\alpha}}\sum_{\ell\in I_k^{d,\alpha}}\gamma_{k,K,\ell,\ell'}\beta_{k,\ell}2^{\|K\|_{\underline{\alpha}/\alpha}/2}\omega_{K,\ell'}^{d,\alpha}.$$

Hence it follows that

$$\mathbb{E}_\beta[\bar\beta_{K,\ell}^2] = \sum_{k=0}^K\sum_{\ell\in I_k^{d,\alpha}}\gamma_{k,K,\ell,\ell'}^2\mathbb{E}_\beta[\beta_{k,\ell}^2] \lesssim \sum_{k=0}^K\left(\sum_{\ell\in I_k^{d,\alpha}}\gamma_{k,K,\ell,\ell'}\right)^2 2^{-k\underline{\alpha}(2+1/\tilde\alpha)}k^{-2}$$

$$\leq \sum_{k=0}^{K} 2^{\|k\|_{\underline{\alpha}/\alpha} - \|K\|_{\underline{\alpha}/\alpha}} \cdot 2^{-k\underline{\alpha}(2+1/\widetilde{\alpha})} k^{-2}$$

$$\lesssim 2^{-\|K\|_{\underline{\alpha}/\alpha}} \cdot \sum_{k=0}^{K} 2^{-2k\underline{\alpha}} k^{-2} \asymp N^{-1},$$

and we again conclude that $\mathrm{Tr}(\Sigma_{\bar{\beta},N})$ is uniformly bounded.

The rest of the proof proceeds similarly to the ordinary Besov space. $\qquad\square$

### C.3 Wavelet Refinement

In this subsection, we present and prove an auxiliary result concerning the refinement of B-spline wavelets and the recurrence relations satisfied by their coefficient sequences.

**Proposition C.10.** *For any $k, k' \in \mathbb{Z}_{\geq 0}^d$ such that $k' - k \in \mathbb{Z}_{\geq 0}^d$ there exists constants $\gamma_{k,k',\ell,\ell'} \geq 0$ for $\ell \in I_k^d$, $\ell' \in I_{k'}^d$ such that*

$$\sum_{\ell \in I_k^d} \beta_{k,\ell} 2^{\|k\|_1/2} \omega_{k,\ell}^d = \sum_{\ell' \in I_{k'}^d} \bar{\beta}_{k',\ell'} 2^{\|k'\|_1/2} \omega_{k',\ell'}^d, \quad \bar{\beta}_{k',\ell'} = \sum_{\ell \in I_k^d} \gamma_{k,k',\ell,\ell'} \beta_{k,\ell} \tag{21}$$

*holds for all $(\beta_{k,\ell})_{\ell \in I_k^d}$. Moreover, it holds that*

$$\sum_{\ell \in I_k^d} \gamma_{k,k',\ell,\ell'} \leq 2^{-\|k'-k\|_1/2}, \quad \sum_{\ell' \in I_{k'}^d} \gamma_{k,k',\ell,\ell'} \leq 2^{\|k'-k\|_1/2}.$$

*Proof.* We proceed by induction on $\|k' - k\|_1$. When $k' = k + e_j$ for some $1 \leq j \leq d$, we can refine each $\omega_{k,\ell}^d$ using equation (2.21) of Dũng (2011b) as

$$\omega_{k,\ell}^d(x) = \prod_{i=1}^{d} \iota_m(2^{k_i} x_i - \ell_i)$$

$$= 2^{-m+1} \prod_{i\neq j} \iota_m(2^{k_i} x_i - \ell_i) \sum_{r=0}^{m} \binom{m}{r} \iota_m\left(2^{k_j+1} x_j - 2\ell_j - r + \frac{m}{2}\right)$$

$$= 2^{-m+1} \sum_{r=0}^{m} \binom{m}{r} \omega_{k+e_j, \ell+(\ell_j+r-\frac{m}{2})e_j}^d(x).$$

Since $\ell + (\ell_j + r - \frac{m}{2})e_j$ matches a given location vector $\ell' \in I_{k+e_j}^d$ if and only if $\ell_i = \ell'_i$ $(i \neq j)$ and $\ell'_j = 2\ell_j + r - \frac{m}{2}$, comparing coefficients in (21) yields

$$\gamma_{k,k+e_j,\ell,\ell'} = 2^{-m+1/2} \mathbf{1}_{\{\ell_i = \ell'_i \, (i \neq j)\}} \binom{m}{\ell'_j - 2\ell_j + \frac{m}{2}}.$$

Here, $\mathbf{1}_A$ denotes the indicator function for condition $A$. It follows that $\gamma_{k,k+e_j,\ell,\ell'} \geq 0$ and

$$\sum_{\ell \in I_k^d} \gamma_{k,k+e_j,\ell,\ell'} \leq \sum_{\ell_j \in \mathbb{Z}} 2^{-m+1/2} \binom{m}{\ell'_j - 2\ell_j + \frac{m}{2}} \leq 2^{-1/2},$$

$$\sum_{\ell' \in I_{k+e_j}^d} \gamma_{k,k+e_j,\ell,\ell'} \leq \sum_{\ell'_j \in \mathbb{Z}} 2^{-m+1/2} \binom{m}{\ell'_j - 2\ell_j + \frac{m}{2}} \leq 2^{1/2},$$

by considering parities.

Now suppose the claim holds for a fixed difference $\|k' - k\|_1$. Applying the above derivation to further refine resolution $k'$ to $k'' = k' + e_j$ for arbitrary $j$ gives

$$\sum_{\ell \in I_k^d} \beta_{k,\ell} 2^{\|k\|_1/2} \omega_{k,\ell}^d = \sum_{\ell' \in I_{k'}^d} \bar{\beta}_{k',\ell'} 2^{\|k'\|_1/2} \omega_{k',\ell'}^d = \sum_{\ell'' \in I_{k'+1}^d} \bar{\bar{\beta}}_{k'+1,\ell''} 2^{(\|k'\|_1+1)/2} \omega_{k'+e_j,\ell''}^d$$

where

$$\bar{\bar{\beta}}_{k'+e_j,\ell''} = \sum_{\ell' \in I_{k'}^d} 2^{-m+1/2} \mathbf{1}_{\{\ell'_i = \ell''_i \, (i \neq j)\}} \binom{m}{\ell''_j - 2\ell'_j + \frac{m}{2}} \bar{\beta}_{k',\ell'}$$

$$= \sum_{\ell \in I_k^d} \sum_{\ell' \in I_{k'}^d} 2^{-m+1/2} \mathbf{1}_{\{\ell'_i = \ell''_i \, (i \neq j)\}} \binom{m}{\ell''_j - 2\ell'_j + \frac{m}{2}} \gamma_{k,k',\ell,\ell'} \beta_{k,\ell}.$$

Hence we obtain the recurrence relation

$$\gamma_{k,k'+e_j,\ell,\ell''} = \sum_{\ell' \in I_{k'}^d} 2^{-m+1/2} \mathbf{1}_{\{\ell'_i = \ell''_i \, (i \neq j)\}} \binom{m}{\ell''_j - 2\ell'_j + \frac{m}{2}} \gamma_{k,k',\ell,\ell'},$$

from which we verify that $\gamma_{k,k'+e_j,\ell,\ell''} \geq 0$ and

$$\sum_{\ell \in I_k^d} \gamma_{k,k'+e_j,\ell,\ell''} = \sum_{\ell' \in I_{k'}^d} 2^{-m+1/2} \mathbf{1}_{\{\ell'_i = \ell''_i \, (i \neq j)\}} \binom{m}{\ell''_j - 2\ell'_j + \frac{m}{2}} \sum_{\ell \in I_k^d} \gamma_{k,k',\ell,\ell'}$$

$$\leq 2^{-m+1/2 - \|k'-k\|_1/2} \sum_{\ell'_j \in \mathbb{Z}} \binom{m}{\ell''_j - 2\ell'_j + \frac{m}{2}} = 2^{-(\|k'-k\|_1+1)/2},$$

and furthermore

$$\sum_{\ell'' \in I_{k'+e_j}^d} \gamma_{k,k'+e_j,\ell,\ell''} = \sum_{\ell' \in I_{k'}^d} \sum_{\ell'' \in I_{k'+e_j}^d} 2^{-m+1/2} \mathbf{1}_{\{\ell'_i = \ell''_i \, (i \neq j)\}} \binom{m}{\ell''_j - 2\ell'_j + \frac{m}{2}} \gamma_{k,k',\ell,\ell'}$$

$$\leq 2^{-m+1/2} \sum_{\ell' \in I_{k'}^d, \, \ell''_j \in \mathbb{Z}} \binom{m}{\ell''_j - 2\ell'_j + \frac{m}{2}} \gamma_{k,k',\ell,\ell'}$$

$$\leq 2^{1/2} \sum_{\ell' \in I_{k'}^d} \gamma_{k,k',\ell,\ell'} \leq 2^{(\|k'-k\|_1+1)/2}.$$

This concludes the proof. $\qquad\square$

## C.4 Proof of Corollary 4.8

In order to approximate arbitrary $\psi_j^\circ \in \mathbb{U}(B_{p,q}^\tau(\mathcal{X}))$, we need the following construction instead of Lemma C.5. Note that $N'$ corresponds to the number of B-splines used to approximate the target function and can be freely chosen to match the desired error, which however affects the covering number of $\mathcal{F}_N$.

**Lemma C.11** (Suzuki (2019), Proposition 1). *Set $m \in \mathbb{N}$, $m > \tau + 2 - 1/p$ and $\nu = (p\tau - d)/2d$. For all $N' \in \mathbb{N}$ sufficiently large and $\epsilon = N'^{-\tau/d}(\log N')^{-1}$, for any $f^\circ \in \mathbb{U}(B_{p,q}^\tau(\mathcal{X}))$ there exists a ReLU network $\tilde{f} \in \mathcal{F}_{\mathrm{DNN}}(L, W, S, M)$ with*

$$L = 3 + 2\left\lceil \log_2\left(3^{d \vee m}(1 + dm^{-1/2}(2e)^{m+1})\epsilon^{-1}\right) + 5\right\rceil \lceil \log_2(d \vee m) \rceil,$$

$$W = N'W_0, \qquad S = ((L-1)W_0^2 + 1)N', \qquad M = O(N'^{1/\nu + 1/d})$$

*satisfying $\|f^\circ - \tilde{f}\|_{L^\infty(\mathcal{X})} \leq N'^{-\tau/d}$.*

Also note that from Assumption 1 it follows that $\|\psi_j\|_{L^\infty(\mathcal{P}_\mathcal{X})} \leq C_\infty N^{1/2}$. Setting $N' \asymp \delta_N^{-d/\tau}$ and applying the covering number bound in Lemma C.6, after some algebra we obtain the following counterpart to Lemma 4.4.

**Corollary C.12.** *For any $\delta_N > 0$, Assumption 3 is satisfied by taking*

$$\mathcal{F}_N = \{\Pi_{B'_N} \circ \phi \mid \phi = (\phi_j)_{j=1}^N, \phi_j \in \mathcal{F}_{\mathrm{DNN}}(L, W, S, M)\}$$

*where $B'_N = C_\infty N^{1/2}$ and*

$$L = O(\log \delta_N^{-1}), \quad W = O(\delta_N^{-d/\tau}), \quad S = O(\delta_N^{-d/\tau} \log \delta_N^{-1}), \quad \log M = O(\log \delta_N).$$

*Also, the metric entropy of $\mathcal{F}_N$ is bounded as*

$$\mathcal{V}(\mathcal{F}_N, \|\cdot\|_{L^\infty}, \epsilon) \lesssim N\delta_N^{-d/\tau} \log\frac{1}{\delta_N}\left(\log\frac{1}{\epsilon} + \log^2\frac{1}{\delta_N}\right).$$

Then by combining with Lemma 3.3 and Proposition 3.2 via Theorem 3.1, it follows that

$$\mathcal{V}(\mathcal{T}_N, \|\cdot\|_{L^\infty}, \epsilon) \lesssim N^2 \log\frac{N}{\epsilon} + N\delta_N^{-d/\tau}\log\frac{1}{\delta_N}\left(\log\frac{N}{\epsilon} + \log^2\frac{1}{\delta_N}\right).$$

and

$$\bar{R}(\widehat{\Theta}) \lesssim \frac{N}{n}\log N + \frac{N^2}{n^2}\log^2 N + N^{-2\alpha/d} + N^2\delta_N^2$$
$$+ \frac{N^2}{T}\log\frac{N}{\epsilon} + \frac{N}{T}\delta_N^{-d/\tau}\log\frac{1}{\delta_N}\left(\log\frac{N}{\epsilon} + \log^2\frac{1}{\delta_N}\right) + \epsilon.$$

Substituting $\delta_N \asymp N^{-1-\alpha/d}$ and $\epsilon \asymp N^{-2\alpha/d}$ concludes the desired bound. $\qquad\square$

# D   Details on Sequential Input

## D.1   Definitions and Results

**$\gamma$-smooth class.**   We first define the $\gamma$-smooth function class introduced by Okumoto and Suzuki (2022). Let $r \in \mathbb{Z}_0^{d\times\infty}$ and $s \in \bar{\mathbb{N}}_0^{d\times\infty}$, where $\bar{\mathbb{N}} = \mathbb{N}\cup\{0\}$ and the subscript $0$ indicates restriction to the subset of elements with a finite number of nonzero components. Consider the orthonormal basis $(\psi_r)_r$ of $L^2([0,1]^{d\times\infty})$ given as

$$\psi_r(x) = \prod_{i\in\mathbb{Z}}\prod_{j=1}^d \psi_{r_{ij}}(x_{ij}), \quad \psi_{r_{ij}}(x_{ij}) = \begin{cases} \sqrt{2}\cos(2\pi r_{ij}x_{ij}) & r_{ij} < 0 \\ 1 & r_{ij} = 0 \\ \sqrt{2}\sin(2\pi r_{ij}x_{ij}) & r_{ij} > 0 \end{cases}.$$

The frequency $s$ component $\delta_s(f)$ of $f \in L^2([0,1]^{d\times\infty})$ is defined as

$$\delta_s(f) := \sum_{\lfloor 2^{s_{ij}-1}\rfloor \leq |r_{ij}| < 2^{s_{ij}}} \langle f, \psi_r\rangle\psi_r.$$

For a monotonically nondecreasing function $\gamma : \bar{\mathbb{N}}_0^{d\times\infty} \to \mathbb{R}$ and $p \geq 2, q \geq 1$, the $\gamma$-smooth norm and function class are defined as

$$\|f\|_{\mathcal{F}_{p,q}^\gamma(\mathcal{P}_\mathcal{X})} := \left(\sum_{s\in\bar{\mathbb{N}}_0^{d\times\infty}} 2^{q\gamma(s)}\|\delta_s(f)\|_{p,\mathcal{P}_\mathcal{X}}^q\right)^{1/q}$$

and

$$\mathcal{F}_{p,q}^\gamma(\mathcal{P}_\mathcal{X}) := \{f \in L^2([0,1]^{d\times\infty}) \mid \|f\|_{\mathcal{F}_{p,q}^\gamma(\mathcal{P}_\mathcal{X})} < \infty\}.$$

The $\gamma$-smooth class over finite-dimensional input space $[0,1]^{d\times m}$ is similarly defined.

In particular, we consider two specific cases of $\gamma$ for the component-wise smoothness parameter $\alpha \in \mathbb{R}_{>0}^{d\times\infty}$, for which we also define the corresponding *degrees of smoothness* $\alpha^\dagger \in \mathbb{R}_{>0}$. Denote by $(\tilde{\alpha}_j)_{j=1}^\infty$ all components of $\alpha$ sorted by ascending magnitude.

- Mixed smoothness: $\gamma(s) = \langle\alpha, s\rangle$, $\alpha^\dagger = \tilde{\alpha}_1 = \max_{i,j}\alpha_{ij}$.
- Anisotropic smoothness: $\gamma(s) = \max_{i,j}\alpha_{ij}s_{ij}$, $\alpha^\dagger = (\sum_{i,j}\alpha_{ij}^{-1})^{-1}$.

Furthermore, the weak $l^\eta$-norm of $\alpha$ is defined as $\|\alpha\|_{wl^\eta} := \sup_j j^\eta\tilde{\alpha}_j^{-1}$ for $\eta > 0$.

**Piecewise $\gamma$-smooth class.** The piecewise $\gamma$-smooth class is an extension of the $\gamma$-smooth class allowing for arbitrary bounded permutations of the tokens of an input (Takakura and Suzuki, 2023). For a threshold $V \in \mathbb{N}$ and an index set $\Lambda$, let $\{\Omega_\lambda\}_{\lambda \in \Lambda}$ be a disjoint partition of $\operatorname{supp} \mathcal{P}_\mathcal{X}$ and $\{\pi_\lambda\}_{\lambda \in \Lambda}$ a set of bijections from $[2V+1]$ to $[-V:V]$. Further define the permutation operator $\Pi : \operatorname{supp} \mathcal{P}_\mathcal{X} \to \mathbb{R}^{d \times (2V+1)}$ as

$$\Pi(x) = (x_{\pi_\lambda(1)}, \cdots, x_{\pi_\lambda(2V+1)}), \quad \text{if } x \in \Omega_\lambda.$$

Then the piecewise $\gamma$-smooth function class is defined as

$$\mathscr{P}^\gamma_{p,q}(\mathcal{P}_\mathcal{X}) := \big\{ g = f \circ \Pi \mid f \in \mathcal{F}^\gamma_{p,q}(\mathcal{P}_\mathcal{X}), \|g\|_{\mathscr{P}^\gamma_{p,q}(\mathcal{P}_\mathcal{X})} < \infty \big\},$$

where

$$\|g\|_{\mathscr{P}^\gamma_{p,q}(\mathcal{P}_\mathcal{X})} := \left( \sum_{s \in \bar{\mathbb{N}}_0^{d \times [-V:V]}} 2^{q\gamma(s)} \|\delta_s(f) \circ \Pi\|_{p,\mathcal{P}_\mathcal{X}}^q \right)^{1/q}.$$

Next, we state the set of assumptions inherited from Takakura and Suzuki (2023). In particular, the *importance function* makes precise a notion of relative importance between tokens which is preserved by permutations.

**Assumption 5** (smoothness and importance function). $1 < q \leq 2$ *and:*

1. *The smoothness parameter $\alpha$ satisfies $\|\alpha\|_{wl^\eta} \leq 1$ and $\alpha_{ij} = \Omega(|i|^\eta)$ for some $\eta > 1$. For mixed smoothness, we also require $\tilde{\alpha}_1 < \tilde{\alpha}_2$.*

2. *There exists a shift-equivariant map $\mu : \operatorname{supp} \mathcal{P}_\mathcal{X} \to \mathbb{R}^\infty$ such that $\mu_0 \in \mathbb{U}(\mathcal{F}^\gamma_{\infty,q})$, $\|\mu_0\| \leq 1$ and $\Omega_\lambda = \{x \in \operatorname{supp} \mathcal{P}_\mathcal{X} \mid \mu(x)_{\pi_\lambda(1)} > \cdots > \mu(x)_{\pi_\lambda(2V+1)}\}$ for all $\lambda \in \Lambda$. $\mu$ is moreover well-separated, that is $\mu(x)_{\pi_\lambda(v)} - \mu(x)_{\pi_\lambda(v+1)} \geq C_\mu v^{-\varrho}$ for $C_\mu, \varrho > 0$.*

We focus on parameter ranges $1 < q \leq 2$ and $\eta > 1$ strictly for simplicity of presentation, but the cases $q = 1, q > 2$ and $\eta > 0$ can be handled with some more analysis. Note that $\eta > 1$ ensures $\alpha^\dagger > 0$ for anisotropic smoothness.

Additionally, the assumption pertaining to our ICL setup is stated as follows.

**Assumption 6.** *For $r \in \mathbb{Z}_0^{d \times \infty}$ the coefficients $\beta_r$ corresponding to $\psi_r$ are independent and satisfy for $s \in \bar{\mathbb{N}}_0^{d \times \infty}$ such that the frequency component $\delta_s(f)$ contains the element $\psi_r$,*

$$\mathbb{E}_\beta[\beta_r] = 0, \quad \mathbb{E}_\beta[\beta_r^2] \lesssim 2^{-(2+1/\alpha^\dagger)\gamma(s)}\gamma(s)^{-2}. \tag{22}$$

*Also $\Sigma_{\Psi,N} \asymp \mathbf{I}_N$ holds, for example $\mathcal{P}_\mathcal{X}$ is bounded above and below with respect to the product measure $\lambda^{d \times \infty}$ on $\mathscr{B}([0,1]^{d \times \infty})$ of the uniform measure $\lambda$ on $\mathscr{B}([0,1])$.*

We then obtain the following result for ICL with transformers:

**Theorem D.1** (minimax optimality of ICL for sequential input). *Let $\mathcal{F}^\circ = \{f \in \mathbb{U}(\mathscr{P}^\gamma_{p,q}(\mathcal{P}_\mathcal{X})) \mid \|f\|_{L^\infty(\mathcal{P}_\mathcal{X})} \leq B\}$ for some $B > 0$ where $\gamma$ corresponds to mixed or anisotropic smoothness. Suppose Assumptions 5 and 6 hold. Then for $n \gtrsim N \log N$ we have*

$$\bar{R}(\widehat{\Theta}) \lesssim N^{-2\alpha^\dagger} + \frac{N \log N}{n} + \frac{N^{2 \vee (1+1/\alpha^\dagger)} \operatorname{polylog}(N)}{T}.$$

*Hence if $T \gtrsim nN^{1 \vee 1/\alpha^\dagger}$ and $N \asymp n^{\frac{1}{2\alpha^\dagger + 1}}$, ICL achieves the rate $n^{-\frac{2\alpha^\dagger}{2\alpha^\dagger + 1}} \operatorname{polylog}(n)$.*

### D.2   Proof of Theorem D.1

Since the system $(\psi_r)_r$ is orthonormal, we may take $\underline{N} = 1, \bar{N} = N$ following Remark 2.1. We mainly utilize the following approximation and covering number bounds.

**Theorem D.2** (Takakura and Suzuki (2023), Theorem 4.5). *For a function $F^\circ \in \mathbb{U}(\mathscr{P}^\gamma_{p,q}(\mathcal{P}_\mathcal{X}))$, $\|F^\circ\|_{L^\infty(\mathcal{P}_\mathcal{X})} \leq B$ and any $K > 0$, there exists a transformer $\widehat{F} \in \mathcal{F}_{\mathrm{TF}}(J, U, D, H, L, W, S, M)$ such that*

$$\|\widehat{F}_0 - F^\circ\|_{L^2(\mathcal{P}_\mathcal{X})} \lesssim 2^{-K},$$

*where*

$$J \lesssim K^{1/\eta}, \quad \log U \lesssim \log K \vee \log V, \quad D \lesssim K^{2(1+\varrho)/\eta} \log V, \quad H \lesssim (\log K)^{1/\eta},$$

$$L \lesssim K^2, \quad W \lesssim 2^{K/\alpha^\dagger} K^{1/\eta}, \quad S \lesssim 2^{K/\alpha^\dagger} K^{2+2/\eta}, \quad \log M \lesssim K \vee \log \log V.$$

**Theorem D.3** (Takakura and Suzuki (2023), Theorem 5.3). *For $\epsilon > 0$ and $B \geq 1$ it holds that*

$$\log \mathcal{N}(\mathcal{F}_{\mathrm{TF}}(J, U, D, H, L, W, S, M), \|\cdot\|_{L^\infty}, \epsilon) \lesssim J^3 L(S + HD^2) \log\left(\frac{DHLWM}{\epsilon}\right).$$

To analyze the decay rate in the $\gamma$-smooth class, we approximate a function $f \in L^2([0,1]^{d\times\infty})$ by the partial sum of its frequency components up to 'resolution' $K$, measured via the $\gamma$ function:

$$R_K(f) := \sum_{\gamma(s) < K} \delta_s(f).$$

The basis functions $\psi_r$ are thus ordered primarily ordered by increasing $\gamma(s)$.

**Lemma D.4** (Okumoto and Suzuki (2022), Lemma 17). *For $1 \leq q \leq 2$ it holds that*

$$\|f - R_K(f)\|_{L^2(\mathcal{P}_{\mathcal{X}})} \lesssim 2^{-K} \|f\|_{\mathcal{F}^\gamma_{p,q}(\mathcal{P}_{\mathcal{X}})}.$$

Note that if $\gamma(s) < K$ then $s_{ij} < K/a_{ij} \lesssim K/|i|^\eta$ for all $i, j$ for both types of smoothness and so $\|s\|_0 \lesssim dK^{1/\eta}$. In addition, the number of basis functions $\psi_r$ used in the sum for $\delta_s(f)$ is exactly $2^{\|s\|_1} = \prod_{i,j} 2^{s_{ij}}$. Theorem D.3 of Takakura and Suzuki (2023) shows that the number of basis elements used in the sum $R_K(f)$ satisfies

$$N \asymp \sum_{\gamma(s) < K} 2^{\|s\|_1} \lesssim 2^{K/\alpha^\dagger}$$

for both mixed and anisotropic smoothness. Hence the $N$-term approximation error decays as $N^{-\alpha^\dagger}$ so that the choice $s = \alpha^\dagger$ leading to the assumed variance bound $N^{-2\alpha^\dagger - 1} \asymp 2^{-(2+1/\alpha^\dagger)K}$ in (22) is justified. Moreover for large $K$,

$$\sum_{\gamma(s) < K} \sum_{\lfloor 2^{s_{ij}-1} \rfloor \leq |r_{ij}| < 2^{s_{ij}}} \|\psi_r\|^2_{L^\infty(\mathcal{P}_{\mathcal{X}})} \leq \sum_{\gamma(s) < K} 2^{\|s\|_1} (\sqrt{2}^{\|s\|_0})^2 \lesssim 2^{K/\alpha^\dagger + O(K^{1/\eta})}$$

since $\|r\|_0 = \|s\|_0$, so that (2) of Assumption 1 is satisfied with $r = 1/2$. The second part of Assumption 1 holds since $(\psi_r)_{r \in \mathbb{Z}_0^{d\times\infty}}$ is orthonormal w.r.t. $\lambda^{d\times\infty}$. Furthermore, the discussion thus far immediately extends to the piecewise $\gamma$-smooth class for any partition $\{\Omega_\lambda\}_{\lambda \in \Lambda}$ by composing with the permutation operator $\Pi$.

We proceed to use Theorem D.2 to approximate each basis function $\psi_r \circ \Pi$ up to resolution $K$. Moreover, we can see from the proof of Lemma 17 of Okumoto and Suzuki (2022) that we do not need to account for the sup-norm scaling of $\psi_r$ and thus it suffices to find the parameter $K' \in \mathbb{N}$ such that the approximation error $2^{-K'} \asymp \delta_N$. Hence combining Theorems D.2 and D.3, we conclude that

$$\mathcal{V}(\mathcal{F}_{\mathrm{TF}}(J, U, D, H, L, W, S, M), \|\cdot\|_{L^\infty}, \epsilon) \lesssim K'^{3/\eta} K'^2 \cdot 2^{K'/\alpha^\dagger} K'^{2+2/\eta} \cdot K' \log \frac{1}{\epsilon}$$

$$\lesssim \left(\frac{1}{\delta_N}\right)^{1/\alpha^\dagger} \mathrm{polylog}\left(N, \frac{1}{\delta_N}\right) \log \frac{1}{\epsilon}$$

is sufficient to satisfy Assumption 3. Therefore, we can now apply our framework with $B'_N \asymp N$ to obtain the bound

$$\bar{R}(\widehat{\Theta}) \lesssim \frac{N}{n} \log N + \frac{N^2}{n^2} \log^2 N + N^{-2\alpha^\dagger} + N^2 \delta_N^2$$

$$+ \frac{N^2}{T} \log \frac{N}{\epsilon} + \frac{1}{T} \delta_N^{-1/\alpha^\dagger} \mathrm{polylog}\left(N, \frac{1}{\delta_N}\right) \log \frac{1}{\epsilon} + \epsilon.$$

Substituting $\delta_N \asymp N^{-1-\alpha^\dagger}$ and $\epsilon \asymp N^{-2\alpha^\dagger}$ concludes the theorem. $\qquad\square$

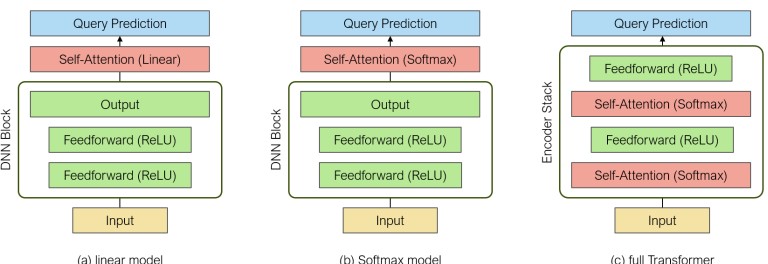

Figure 1: Architecture of the compared models. Each model contains two MLP components, all attention layers are single-head and LayerNorm is not included. (a),(b) implement the simplified reparametrization for attention, while all layers in (c) utilize the full embeddings. The input dimension is 8 and all hidden layer and DNN output widths are 32. The query prediction is read off the last entry of the output at the query position.

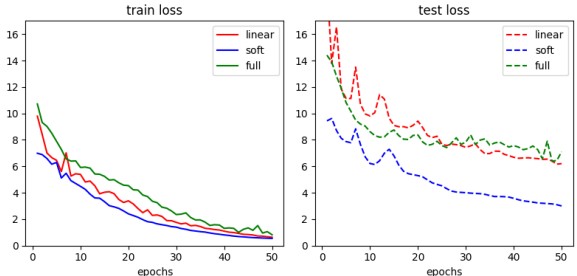

Figure 2: Training and test curves for the ICL pretraining objective. We use the Adam optimizer with a learning rate of 0.02 for all layers. For the task class we take $\alpha = 1$, $p = q = \infty$, $T = n = 512$ and generate samples from random combinations of order 2 wavelets.

# E    Numerical Experiments

In this section, we connect our theoretical contributions to practical transformers by conducting experiments verifying our results as well as justifying the simplified model setup and the empirical risk minimization assumption. We implement and compare the following toy models: (a) the simplified architecture studied in our paper; (b) the same model with linear attention replaced by softmax; and (c) a full transformer with 2 stacked encoder layers. The number of feedforward layers, widths of hidden layers, learning rate, etc. are set to equal for a fair comparison, see Figure 1 for details.

Figure 2 shows training (solid) and test loss curves (dashed) during pretraining. All 3 architectures exhibit similar behavior and converge to near zero training loss, justifying the use of our simplified model and also supporting the assumption that the empirical risk is minimized. Moreover, Figure 3 shows the converged losses over a wide range of $N, n, T$ values. We verify that increasing $N, n$

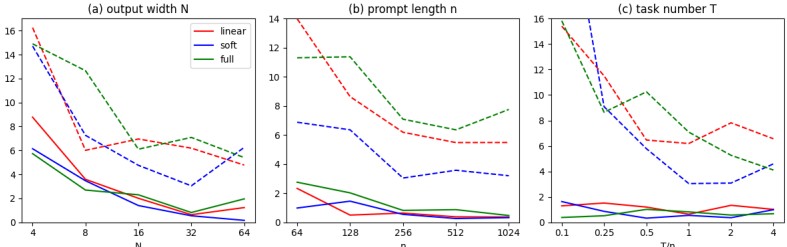

Figure 3: Training and test losses of the three models after 50 epochs while varying (a) DNN width $N$; (b) number of in-context samples $n$; (c) number of tasks $T$. For (a), the widths of all hidden layers also vary with $N$. We take the median over 5 runs for robustness.

leads to decreasing train and test error, corresponding to the approximation error of Theorem 3.1. We also observe that increasing $T$ tends to improve the pretraining generalization gap up to a threshold, confirming our theoretical analysis of task diversity. Again, this behavior is consistent across the 3 architectures. We note that in the overparametrized regime when the number of total parameters $\gtrsim nT$, the trained model is likely not the empirical risk minimizer, which may also contribute to the large error for large $N$ or small $n, T$.

# F  Proofs of Minimax Lower Bounds

## F.1  Proof of Proposition 5.1

In this section, we develop our framework for obtaining minimax lower bounds in the ICL setup by adapting the information-theoretic approach of Yang and Barron (1999).

Let $\{(\psi^{(j)}, \beta_{T+1}^{(j)})\}_{j=1}^{M}$ be a $\delta_n$-packing of the class $\mathcal{F}^\circ$ with respect to the $L^2(\mathcal{X})$-norm such that

$$\|\beta_{T+1}^{(j)\top}\psi^{(j)} - \beta_{T+1}^{(j')\top}\psi^{(j')}\|_{L^2(\mathcal{X})}^2 \geq \delta_n^2, \quad 1 \leq j < j' \leq M,$$

where $M$ is the corresponding packing number. Then we have the following proposition as an application of Fano's inequality (Yang and Barron, 1999).

**Proposition F.1.** *Let $\Theta$ be a random variable uniformly distributed over $\{(\psi^{(j)}, \beta_{T+1}^{(j)})\}_{j=1}^{M}$. Then, it holds that*

$$\inf_{\widehat{f}_n : \mathcal{D}_{n,T} \to \mathbb{R}} \sup_{f^\circ \in \mathcal{F}^\circ} \mathbb{E}_{\mathcal{D}_{n,T}}[\|\widehat{f} - f^\circ\|_{L^2(\mathcal{P}_\mathcal{X})}^2] \geq \frac{\delta_n^2}{2}\left(1 - \frac{\mathbb{E}_{\mathbf{X}}[I_{\mathbf{X}^{(1:T+1)}}(\Theta, \mathbf{y}^{(1:T+1)})] + \log 2}{\log M}\right),$$

*where $I_{\mathbf{X}^{(1:T+1)}}(\Theta, \mathbf{y}^{(1:T+1)})$ is the mutual information between $\Theta, \mathbf{y}^{(1:T+1)}$ for given $\mathbf{X}^{(1:T+1)}$.*

The mutual information $I_{\mathbf{X}^{(1:T+1)}}(\Theta, \mathbf{y}^{(1:T+1)})$ is formulated more concretely as

$$\sum_{\theta \in \text{supp}\,\Theta} w(\theta) \int p(\mathbf{y}^{(1:T+1)}|\theta, \mathbf{X}^{(1:T+1)}) \log\left(\frac{p(\mathbf{y}^{(1:T+1)}|\theta, \mathbf{X}^{(1:T+1)})}{p_w(\mathbf{y}^{(1:T+1)}|\mathbf{X}^{(1:T+1)})}\right) \mathrm{d}\mathbf{y}^{(1:T+1)},$$

where $p(\mathbf{y}|\theta, \mathbf{X})$ is the probability density of $\mathbf{y}$ conditioned on $\theta, \mathbf{X}$ and $p_w$ is the marginal distribution of $\mathbf{y}^{(1:T+1)}$ where $w(\cdot) \equiv \frac{1}{M}$ is the probability mass function of $\Theta$ (i.e., $p_w(\cdot|\mathbf{X}^{(1:T+1)}) = \sum_{\theta \in \text{supp}\,\Theta} w(\theta)p(\cdot|\theta, \mathbf{X}^{(1:T+1)})$). We let $P_{\mathbf{y}^{(t)}|\theta}$ (and $P_{\mathbf{y}^{(1:t)}|\theta}$) be the distribution of $\mathbf{y}^{(t)}$ conditioned on $\theta, \mathbf{X}^{(t)}$ (and $\mathbf{X}^{(1:t)}$) respectively, and let

$$\bar{P}_{\mathbf{y}^{(1:T+1)}} = \frac{1}{M}\sum_{j=1}^{M} P_{\mathbf{y}^{(1:T+1)}|\theta^{(j)}}$$

be the marginal distribution of $\mathbf{y}^{(1:T+1)}$ conditioned on $\mathbf{X}^{(1:T+1)}$.

Next, we define the set $\{\tilde{\psi}^{(j)}\}_{j=1}^{Q_1}$ to be a $\varepsilon_{n,1}$-covering of $\mathcal{F}_N$ w.r.t. the norm $d(\psi, \psi') := \sqrt{\mathbb{E}_x[\|\psi(x) - \psi'(x)\|^2]}$ with $\varepsilon_{n,1}$-covering number $Q_1$, and $\{\tilde{\beta}^{(j)}\}_{j=1}^{Q_2}$ to be a $\varepsilon_{n,2}$-covering of $\mathcal{B}$ w.r.t. the $L^2$ norm with $\varepsilon_{n,2}$-covering number $Q_2$. By taking all combinations of $(\tilde{\psi}^{(j)}, \tilde{\beta}^{(j')})$ for $1 \leq j \leq Q_1$ and $1 \leq j' \leq Q_2$, we obtain the covering $\{\tilde{\theta}^{(j)}\}_{j=1}^{Q}$ with respect to the quantity $\varepsilon_n^2 = \sigma_\beta^2 \varepsilon_{n,1}^2 + C_2 \varepsilon_{n,2}^2$ where $Q = Q_1 Q_2$ and each $\tilde{\theta}^{(j)}$ is given by $\tilde{\theta}^{(j)} = (\tilde{\psi}^{(j_1)}, \tilde{\beta}^{(j_2)})$ for some indices $j_1$ and $j_2$.

Then as in the discussion of Yang and Barron (1999), the mutual information is bounded by

$$I_{\mathbf{X}^{(1:T+1)}}(\Theta, \mathbf{y}^{(1:T+1)}) = \frac{1}{M}\sum_{j=1}^{M} D(P_{\mathbf{y}^{(1:T+1)}|\theta^{(j)}} \| \bar{P}_{\mathbf{y}^{(1:T+1)}})$$

$$\leq \frac{1}{M}\sum_{j=1}^{M} D(P_{\mathbf{y}^{(1:T+1)}|\theta^{(j)}} \| \tilde{P}_{\mathbf{y}^{(1:T+1)}}),$$

where $D(\cdot\|\cdot)$ is the Kullback-Leibler divergence and $\tilde{P}_{\boldsymbol{y}^{(1:T+1)}} = \frac{1}{Q}\sum_{j=1}^{Q} P_{\boldsymbol{y}^{(1:T+1)}|\tilde{\theta}^{(j)}}$ because $\bar{P}_{\boldsymbol{y}^{(1:T+1)}}$ minimizes the right hand side. If we let

$$\kappa(j) := \arg\min_{1\leq k\leq Q} D(P_{\boldsymbol{y}^{(1:T+1)}|\theta^{(j)}}\|P_{\boldsymbol{y}^{(1:T+1)}|\tilde{\theta}^{(k)}}),$$

then each summand of the right-hand side is further bounded by

$$\log Q + D(P_{\boldsymbol{y}^{(1:T+1)}|\theta^{(j)}}\|P_{\boldsymbol{y}^{(1:T+1)}|\tilde{\theta}^{\kappa(j)}}).$$

Moreover, for $\theta = (\psi, \beta^{(T+1)})$ it holds that

$$p(\boldsymbol{y}^{(1:T+1)}|\theta, \mathbf{X}^{(1:T+1)})$$

$$= \prod_{t=1}^{T} p(\boldsymbol{y}^{(t)}|\psi, \mathbf{X}^{(t)}) \cdot p(\boldsymbol{y}^{(T+1)}|\psi, \beta^{(T+1)}, \mathbf{X}^{(T+1)})$$

$$= \prod_{t=1}^{T} \int p(\boldsymbol{y}^{(t)}|\psi, \beta^{(t)}, \mathbf{X}^{(t)})p_\beta(\beta^{(t)})\mathrm{d}\beta^{(t)} \cdot p(\boldsymbol{y}^{(T+1)}|\psi, \beta^{(T+1)}, \mathbf{X}^{(T+1)}).$$

Then the KL-divergence can be bounded as

$$D(P_{\boldsymbol{y}^{(1:T+1)}|\theta^{(j)}}\|P_{\boldsymbol{y}^{(1:T+1)}|\tilde{\theta}^{\kappa(j)}})$$

$$= \sum_{t=1}^{T} D\left(P_{\boldsymbol{y}^{(t)}|\psi^{(j)}}\|P_{\boldsymbol{y}^{(t)}|\tilde{\psi}^{(\kappa(j))}}\right) + D\left(P_{\boldsymbol{y}^{(T+1)}|\psi^{(j)},\beta_{T+1}^{(j)}}\|P_{\boldsymbol{y}^{(T+1)}|\tilde{\psi}^{(\kappa(j))},\tilde{\beta}_{T+1}^{(\kappa(j))}}\right)$$

$$\leq \sum_{t=1}^{T} \int D\left(P_{\boldsymbol{y}^{(t)}|\psi^{(j)},\beta^{(t)}}\|P_{\boldsymbol{y}^{(t)}|\tilde{\psi}^{(\kappa(j))},\beta^{(t)}}\right) p_\beta(\beta^{(t)})\mathrm{d}\beta^{(t)}$$

$$+ D\left(P_{\boldsymbol{y}^{(T+1)}|\psi^{(j)},\beta_{T+1}^{(j)}}\|P_{\boldsymbol{y}^{(T+1)}|\tilde{\psi}^{(\kappa(j))},\tilde{\beta}_{T+1}^{(\kappa(j))}}\right),$$

where the joint convexity of KL-divergence was used for the last inequality. Since the observation noise is assumed to be normally distributed, the integrand KL-divergence can be bounded as

$$D\left(P_{\boldsymbol{y}^{(t)}|\psi^{(j)},\beta}\|P_{\boldsymbol{y}^{(t)}|\tilde{\psi}^{\kappa(j)},\beta}\right) = \sum_{i=1}^{n} \frac{1}{2\sigma^2}\left(\beta^\top\psi^{(j)}(x_i^{(t)}) - \beta^\top\tilde{\psi}^{(\kappa(j))}(x_i^{(t)})\right)^2.$$

Hence, its expectation with respect to $\beta, \mathbf{X}^{(t)}$ becomes

$$\mathbb{E}_{\mathbf{X}^{(t)},\beta}\left[D\left(P_{\boldsymbol{y}^{(t)}|\psi^{(j)},\beta}\|P_{\boldsymbol{y}^{(t)}|\tilde{\psi}^{\kappa(j)},\beta}\right)\right] = \frac{n\sigma_\beta^2}{2\sigma^2}\|\psi^{(j)} - \tilde{\psi}^{\kappa(j)}\|_{L^2(\mathcal{P}_\mathcal{X})}^2 \leq \frac{n\sigma_\beta^2}{2\sigma^2}\varepsilon_{n,1}^2,$$

In the same manner, we have that

$$D\left(P_{\boldsymbol{y}^{(T+1)}|\psi^{(j)},\beta_{T+1}^{(j)}}\|P_{\boldsymbol{y}^{(T+1)}|\tilde{\psi}^{\kappa(j)},\tilde{\beta}_{T+1}^{(\kappa(j))}}\right)$$

$$= \frac{1}{2\sigma^2}\sum_{i=1}^{n}\left(\beta_{T+1}^{(j)\top}\psi^{(j)}(x_i^{(t)}) - \tilde{\beta}_{T+1}^{(\kappa(j))\top}\tilde{\psi}^{(\kappa(j))}(x_i^{(t)})\right)^2$$

$$\leq \sum_{i=1}^{n} \frac{1}{2\sigma^2}\left[\left(\beta_{T+1}^{(j)} - \tilde{\beta}_{T+1}^{(\kappa(j))}\right)^\top \tilde{\psi}^{(\kappa(j))}(x_i^{(t)})\right]^2 + \sum_{i=1}^{n} \frac{1}{2\sigma^2}\left[\beta_{T+1}^{(j)\top}(\psi^{(j)}(x_i^{(t)}) - \tilde{\psi}^{(\kappa(j))}(x_i^{(t)}))\right]^2.$$

The expectation of the right-hand side with respect to $\mathbf{X}^{(T+1)}, \beta_{T+1}^{(j)}$ is bounded as

$$\mathbb{E}_{\mathbf{X}^{(T+1)},\beta_{T+1}^{(j)}}\left[D\left(P_{\boldsymbol{y}^{(T+1)}|\psi^{(j)},\beta_{T+1}^{(j)}}\|P_{\boldsymbol{y}^{(t)}|\tilde{\psi}^{\kappa(j)},\tilde{\beta}_{T+1}^{(\kappa(j))}}\right)\right]$$

$$\leq \frac{C_2 n}{2\sigma^2}\|\beta_{T+1}^{(j)} - \tilde{\beta}_{T+1}^{(\kappa(j))}\|^2 + \frac{n}{2\sigma^2}\sigma_\beta^2\|\psi^{(j)} - \tilde{\psi}^{(\kappa(j))}\|_{L^2(\mathcal{P}_\mathcal{X})}^2$$

$$\leq \frac{C_2 n}{2\sigma^2}\varepsilon_{n,2}^2 + \frac{n}{2\sigma^2}\sigma_\beta^2\varepsilon_{n,1}^2 = \frac{n}{2\sigma^2}\varepsilon_n^2.$$

Therefore, the expected mutual information can be bounded as

$$\mathbb{E}_X[I_{\mathbf{X}^{(1:T+1)}}(\Theta, \boldsymbol{y}^{(1:T+1)})] \leq \log Q_1 + \log Q_2 + \frac{nT}{2\sigma^2}\sigma_\beta^2\varepsilon_{n,1}^2 + \frac{n}{2\sigma^2}\varepsilon_n^2.$$

Applying Proposition F.1 together with (7) concludes the proof. $\qquad\square$

## F.2 Lower Bound in Besov Space

Here, we derive the minimax lower bound when $\mathcal{F}^\circ = \mathbb{U}(B^\alpha_{p,q}(\mathcal{X}))$. Recall that in this setting $s = \alpha/d$. We fix a resolution $K$ and then consider the set of B-splines $\omega^d_{K,\ell}, \ell \in I^d_K$ of cardinality $N' \asymp 2^{Kd}$. Considering the basis pairs $(\omega^d_{K,1}, \omega^d_{K,2}), \ldots, (\omega^d_{K,N'-1}, \omega^d_{K,N'})$, we can determine which one is employed to construct the basis $\psi^{(j)}$. The Varshamov-Gilbert bound yields that for $\Omega = \{0,1\}^{N'/2}$, we can construct a subset $\Omega' = \{w_1, \ldots, w_{2^{N'/16}}\} \subset \Omega$ such that $|\Omega| = 2^{N'/16}$ and $w \neq w' \in \Omega'$ has a Hamming distance not less than $N'/16$. Using this $\Omega'$, we set $N = N'/2$ and $M = 2^{N'/16}$ and define $(\psi^{(j)})^M_{j=1}$ as $\psi^{(j)}_i = \omega^d_{K,2i-1}$ if $w_{j,i} = 0$ and $\psi^{(j)}_i = \omega^d_{K,2i}$ if $w_{j,i} = 1$. We use the same B-spline bases with resolution more than $K$ for $\psi^{(j)}_i$ ($i \geq N$) across all $j$.

By the construction of $(\psi^{(j)})$, if we set $\beta^{(1)} = (\sigma_\beta, \ldots, \sigma_\beta, 0, 0, \ldots)$, then

$$\|\beta^{(1)\top}\psi^{(j)} - \beta^{(1)\top}\psi^{(j')}\|^2_{L^2(\mathcal{P}_\mathcal{X})} \geq \sigma^2_\beta N/8.$$

Hence, for $\delta^2_n \leq \sigma^2_\beta N/8 \lesssim 1$, the $\delta_n$-packing number is not less than $2^{N/8}$. Moreover, the logarithmic $\delta_n$-packing number of $\{\beta^\top\psi^{(j)} \mid \beta \in \mathcal{B}\}$ for a fixed $j$ is $\Theta(\min\{\delta^{-1/s}_n, N\log(1/\delta_n)\})$ by the standard argument.

Hence taking $\delta_n = N^{-s}$, we obtain $\log M \gtrsim N$ and the upper bound of the covering numbers $\log Q_1 + \log Q_2 \lesssim N$ for $\sigma^2_\beta \varepsilon^2_{n,1} \leq \delta^2_n$ and $\varepsilon^2_{n,2} = C\delta^2_n$ where $C$ is a constant. Then, by choosing $C$ appropriately and $\varepsilon_{n,1} \lesssim N^{-1-s}$ (so that $\log Q_1 \lesssim N$), as long as

$$nT\sigma^2_\beta\varepsilon^2_{n,1} + n\delta^2_n \lesssim \log Q_1 + \log Q_2 \lesssim N$$

is satisfied, the minimax rate is lower bounded by $\delta^2_n$. Taking $N \asymp n^{\frac{1}{2s+1}}$, we obtain the lower bound

$$\delta^2_n \gtrsim n^{-\frac{2s}{2s+1}}. \tag{23}$$

## F.3 Lower Bound with Coarser Basis

We consider a generalized setting where $\mathcal{X} = \underbrace{\mathbb{R}^d \times \mathbb{R}^d \times \cdots \times \mathbb{R}^d}_{(N+1)\text{times}}$ and take $\psi^{(j)}_i \in \mathbb{U}(B^\tau_{p,q}(\mathbb{R}^d))$ and assume that $\beta_1 \in [-1,1]$ and $\beta_j \in [-\sigma_\beta, \sigma_\beta]$ where $\sigma^2_\beta = \tilde{\Theta}(N^{-2s-1})$. Since the logarithmic $\tilde{\varepsilon}_1$-covering and packing numbers of $\mathbb{U}(B^\tau_{p,q}(\mathbb{R}^d))$ are $\Theta(\tilde{\varepsilon}^{-d/\tau}_1)$, those for the basis functions on $j = 2, \ldots, N+1$ become $\Theta(N\tilde{\varepsilon}^{-d/\tau}_1)$, and those for $\mathcal{B}$ are $\Theta(N\log(1 + \frac{N\sigma^2_\beta}{\varepsilon^2_{n,2}}))$. Therefore, by taking $\varepsilon^2_{n,1} = N\tilde{\varepsilon}^2_1$ we see that

$$nT(\varepsilon^2_{n,1} + \sigma^2_\beta\varepsilon^2_{n,1}) + n\varepsilon^2_{n,2} \lesssim \varepsilon^{-d/\tau}_{n,1} + N\left(\frac{\varepsilon_{n,1}}{\sqrt{N}}\right)^{-d/\tau} + N\log\left(1 + \frac{N\sigma^2_\beta}{\varepsilon^2_{n,2}}\right)$$

should be satisfied. Moreover, by taking $\varepsilon^{(2\tau+d)/\tau}_{n,1} \asymp 1/nT$ and $\varepsilon^2_{n,2} \asymp N\log(1 + N^{-2s}/\varepsilon^2_{n,2})/n$ we can balance both sides. In particular, we may set

$$\varepsilon^2_{n,1} \asymp (nT)^{-\frac{2\tau}{2\tau+d}}, \quad \varepsilon^2_{n,2} \asymp \frac{N}{n} \wedge N^{-2s}.$$

Taking the balance with respect to $N$ to maximize $\varepsilon^2_{n,2}$, we have $N \asymp n^{\frac{d}{2\alpha+d}}$ and $\varepsilon^2_{n,1} \asymp (nT)^{-\frac{2\tau}{2\tau+d}}$. Therefore, the minimax rate is lower bounded as

$$\delta^2_n \simeq (1 + \sigma^2_\beta)\varepsilon^2_{n,1} + \varepsilon^2_{n,2} \simeq n^{-\frac{2\alpha}{2\alpha+d}} + (nT)^{-\frac{2\tau}{2\tau+d}}. \tag{24}$$

## F.4 Lower Bound in Piecewise $\gamma$-smooth Class

Suppose that we utilize the basis functions up to resolution $K$. Then, by the argument by Nishimura and Suzuki (2024), the number of basis functions $\psi_r$ in the $K$-th resolution is $N' \asymp 2^{K/a^\dagger}$. Moreover, the $\delta_n$-packing number of the $\gamma$-smooth class is also lower bounded by

$$\log M \geq N' \operatorname{polylog}(\delta_n, N'). \tag{25}$$

Here, by noticing the approximation error bound in Appendix D.2, we take $N' \asymp \delta_n^{-1/a^\dagger}$ where the basis functions are chosen from the $K$th resolution. As in the case of the Besov space, we construct $(\psi^{(j)})_{j=1}^{M'}$ where $M' = 2^{N'/16}$ and $\psi^{(j)}(x) \in \mathbb{R}^N$ for $N = N'/2$ and $\|\psi^{(j)} - \psi^{(j')}\|_{L^2(P_{\mathcal{X}})}^2 \geq N/8$ for $j \neq j'$. Following the same argument as in the Besov case, we need to take $\varepsilon_{n,1}$ and $\varepsilon_{n,2}$ as

$$nT\sigma_\beta^2\varepsilon_{n,1}^2 + n\varepsilon_{n,2}^2 \lesssim \delta_n^{-1/a^\dagger}$$

up to logarithmic factors. This is satisfied by taking $\varepsilon_{n,2}^2 = C\delta_n^2 \asymp n^{-\frac{2a^\dagger}{2a^\dagger+1}}$ with a constant $C$ and balancing $N$ so that $\sigma_\beta^2\varepsilon_{n,1}^2 = (nT)^{-\frac{2a^\dagger}{2a^\dagger+1}}\sigma_\beta^{\frac{2}{2a^\dagger+1}} \asymp T^{-1}n^{-\frac{2a^\dagger}{2a^\dagger+1}}$. Combining this evaluation and (25) yields that the minimax lower bound is given by

$$\delta_n^2 \gtrsim n^{-\frac{2a^\dagger}{2a^\dagger+1}}. \tag{26}$$

