# OpenReview forum: "Transformers are Minimax Optimal Nonparametric In-Context Learners"
_NeurIPS.cc/2024/Conference — NeurIPS 2024 poster_

### Official Review · Reviewer_eg5H · 2024-07-09

**Soundness:** 3
**Presentation:** 4
**Contribution:** 3
**Rating:** 7
**Confidence:** 4

**Summary:**

This paper analyzed in-context learning of a transformer consisting of a DNN and a linear attention layer pretrained on nonparametric regression tasks.  The authors derived a general bound on the generalization error consisting of the approximation error, in-context generalization error and pertaining generalization error. They also showed that the icl prediction given by the pretrained TF is minimax optimal when the nonparametric regression functions are from the Besov space or  piecewise  $\gamma$-smooth class.

**Strengths:**

The paper is well-written and the theoretical results are solid. The authors made reasonable assumptions on the regression tasks and the TF function class that do not significantly simplify the problem. From a technical perspective, this work leverages the approximation ability of DNNs and the ICL ability of single-layer linear attention to derive the near-optimal generalization error bounds. A few generalizations are also considered, e.g., anisotropic Besov space and piecewise  $\gamma$-smooth class.

**Weaknesses:**

The paper considered a model different from the standard TF models in practice, in the sense that a trainable DNN-based feature map is applied to the tokens before the attention layer. The proof in this works heavily rely on this feature map, as it is used to approximate the basis functions of the Besov space.

Lack of simulation results. The work showed that the empirical risk minimizer is minimax-optimal but didn't analyze the training dynamics. So it would be good to have empirical evidence showing that pretraining indeed finds an empirical risk minimizer.

**Questions:**

1. The functions in the Besov space are well approximated by linear combinations of finite number of basis functions in the space. Suppose now the feature map $\phi$ can well-approximate the basic functions (assumption 3) and is fixed to be $\phi^*$, and only the linear weight in the attention layer is trainable. I wonder how much differences there are between this simplified setting and previous works on ICL for linear regression.

2. I wonder if it is possible to theoretically analyze the training dynamics under the setting where the feature map is given and fixed and only the linear attention is trainable.

**Limitations:**

As the author already mentioned, this work only analyzed the generalization error of the ERM but not the training dynamics. The transformer model in this work is limited to a single layer of self-attention and does not include the softmax activation and the MLP layer.

---

> ### Author Rebuttal · Authors · 2024-08-06
>
> Thank you for your detailed review and positive assessment of our theoretical contributions! Here are our responses to the comments.
>
> **Weakness 1 & Limitations.**
>
> While the reviewer mentioned in the Limitations section that our model does not include the MLP layer, we find it illuminating to consider our setup as a deep Transformer with all attention layers (except the last one) and layernorm/output layers removed, so that the MLP layers have combined into a feedforward DNN. Since this simplified model already achieves optimal ICL, we can expect a full Transformer to achieve the same. Moreover, our techniques can be partially applied to settings with multiple attention layers or nonlinear (e.g. softmax) attention; please see **Item 1** of the global response where we address this in depth.
>
> **Weakness 2.**
>
> Motivated by the reviewers' comments, we have conducted new numerical experiments justifying our simplified model setup (Weakness 1) and the assumption of empirical risk minimization (Weakness 2). Please see **Item 3** of the global response for details.
>
> **Question 1.**
>
> Indeed, if we completely remove the DNN and set both the features and basis to be simply the coordinate mappings $x_1,\cdots,x_d$, this reduces to the class of linear maps in prior works, e.g. Zhang et al. (2023). In a nutshell, our contribution is extending this to infinite-dimensional nonparametric task classes via learnable representations, enabling fine-grained sample complexity analysis and guaranteeing optimality of ICL with deep architectures.
>
> **Question 2.**
>
> If the features are fixed, the dynamics are arguably less interesting. Since the attention layer output is linear in the attention matrix $\Gamma$, the pretraining loss function is always convex (ignoring clipping). Hence gradient descent is easily shown to converge exponentially fast to a minimizer regardless of the number of samples, random initialization, etc. which can also be written down explicitly by computing the matrix derivative of line 145 to zero and solving for $\Gamma$.
>
> The linear regression setup of the previous work by Zhang is not quite as simple, in that they also include a scalar multiplier $\sigma$ representing the value matrix and study the joint dynamics. In that paper, convergence is proved by showing a PL inequality under restrictive assumptions on initialization. Rough calculations show that a similar result can be obtained in the fixed feature setting as well, justifying the fixed value matrix assumption in our model. Some other works also indicate that the attention mechanism may possess structures favorable for optimization; see our discussion in Section 1.1. Nevertheless, the main novelty of our work comes from incorporating the representational capability of the MLP layers, for which a dynamical analysis is unfortunately still out of reach.

---

> ### Author Response · Authors · 2024-08-11
>
> Thank you again for your time and effort in reviewing our paper! As the discussion period is ending soon, we are following up to see if our response was satisfactory in addressing the reviewer's questions. If not, we would be happy to discuss any remaining concerns.

---

> > ### Comment · Reviewer_eg5H · 2024-08-13
> >
> > Thanks for the response. I will maintain my positive score.

---

### Official Review · Reviewer_YwVZ · 2024-07-11

**Soundness:** 3
**Presentation:** 2
**Contribution:** 3
**Rating:** 5
**Confidence:** 4

**Summary:**

This paper explores the ICL capabilities of transformers from a statistical learning theory perspective. It focuses on transformers with a deep neural network and a linear attention layer, pretrained on nonparametric regression tasks from function spaces like the Besov space and piecewise gamma-smooth class. The authors demonstrate that sufficiently trained transformers can achieve or even surpass the minimax optimal estimation risk by encoding relevant basis representations during pretraining. The paper also establish result that explains the pretraining and in-context generalization gaps, which is essential for understanding ICL.

**Strengths:**

1. The topic of this work is both interesting and important, addressing key aspects of ICL in transformers.

2. The results developed, especially Theorem 4.5, are very encouraging and could help the community better understand LLMs and their ICL capacity.

3. The analysis provided is rigorous and sound, offering new tools and methodologies for theoretical analysis in the ICL literature.

**Weaknesses:**

1. The writing and presentation of the paper need improvement. For instance, in Section 4, the flow feels disjointed, with several results appearing to be stitched together without a clear, organized connection.

2. Although the ICL capabilities of transformers are powerful and of great interest, the simplified transformer model presented in this paper is not very realistic. While it is common to use simplified models for analyzing LLMs, the paper would greatly benefit from numerical experiments demonstrating the claimed theoretical findings.

**Questions:**

While reading this paper, I feel it provides new insights into understanding ICLs. However, I cannot find concrete evidence of how this theoretical framework can offer supportive insights for practitioners using ICL. The simplified model without numerical experiments cannot indicate how real LLMs behave.

Could the authors provide more detailed explanations on how this work connects to the mechanisms of ICL and LLMs in practice?

---

> ### Author Rebuttal · Authors · 2024-08-06
>
> Thank you for your through review and helpful suggestions, which have helped us greatly to improve our paper!
>
> **Response to Weakness 1.**
>
> Unfortunately, Section 5 was completed last minute which resulted in the flow of the paper being rather disjoint. Together with improved lower bounds, we restate our take-home message (besides in-context optimality in $n$) as follows. We believe this novel understanding will be helpful to both theoreticians and practitioners.
>
> * (In the Besov space setting) The obtained upper bound $n^{-\frac{2\alpha}{2\alpha+d}}$ when $T\gtrsim nN$ which is minimax optimal in $n$, has also been shown to be jointly optimal in $n,T$. Hence **ICL is provably optimal in the large $T$ regime.**
> * (In the coarser basis setting) We obtained an improved lower bound $n^{-\frac{2\alpha}{2\alpha+d}} + (nT)^{-\frac{2\tau}{2\tau+d}}$ using the method in Appendix E.3. If $T=O(1)$, this gives the slower $\tau$ rate, while if $T$ is larger this gives the faster $\alpha$ rate. This aligns with Corollary 4.9 (which only gave an upper bound), hence **ICL is provably suboptimal in the small $T$ regime.**
>
> When combined, these results are stronger than optimality in only $n$. (Since the Transformer requires $n$ in-context samples plus $n\times T$ pretraining samples, one could argue that standard minimax rates do not apply. However, the above lower bounds do apply rigorously to any meta-learning algorithm in the $n\times T$ samples.) They also align with experimental observations of task diversity threshold (Raventos 2023), rigorously supporting the importance of varied tasks over increasing prompt length.
>
> **Response to Weakness 2 & Question 1.**
>
> Motivated by your comments on practical relevance, we have conducted new numerical experiments justifying our simplified model setup and the assumption of empirical risk minimization. Figures can be found in the attached PDF of the global response. We implement and compare the following toy models:
>
> * Linear Transformer: the model studied in our paper, where a DNN (2 hidden layers) feeds into a simplified linear attention layer
> * Softmax Transformer: the same model as (a) but with softmax attention
> * Full Transformer: an ordinary transformer with 2 stacked encoder layers
>
> Architectures are compared in Figure 1. The number of feedforward layers, width of hidden layers, learning rate etc. are set to equal for a fair comparison.
>
> * Figure 2 shows training (solid lines) and test loss curves (dashed lines) during pretraining. All 3 architectures exhibit similar behavior and converge to near zero training loss, justifying the use of our simplified model and also supporting the assumption that the empirical risk can be minimized.
> * Moreover, Figure 3 shows the converged losses over a wide range of $N,n,T$ values (median over 5 runs). We verify that increasing $N,n$ leads to decreasing train and test error, corresponding to the approximation error of Theorem 3.1. We also observe that increasing $T$ tends to improve the pretraining generalization gap up to a threshold, confirming our theoretical analyses. Again, this behavior is consistent across the 3 architectures.
> * Although the dimensions and architectures are relatively small due to limited time and compute, we plan to scale up our experiments and add them to the final version of the paper.
>
> Please also see **Item 1** of the global response where we discuss how our theoretical results may be extended to more complex transformer architectures. We also humbly ask the reviewer to consider raising their score if their concerns were addressed.

---

> > ### Comment · Reviewer_YwVZ · 2024-08-09
> >
> > Thanks for the reply. My questions are partly addressed and I'll maintain my rating.

---

> > > ### Author Response · Authors · 2024-08-09
> > >
> > > Thank you for taking the time to read our rebuttal! As your criticisms were generally on the lack of numerical experiments, if there are any other types of experiments which you believe will further benefit our paper, please inform us and we will strive to implement them to gain more insight into our analyses.

---

### Official Review · Reviewer_1n5m · 2024-07-17

**Soundness:** 3
**Presentation:** 2
**Contribution:** 3
**Rating:** 5
**Confidence:** 3

**Summary:**

This work shows that ICL can perform non-parametric regression at an optimal rate.

Section 3 gives an upper bound on ICL error in terms of a metric entropy of the representation class. Section 4 instantiates the bound for DNN representations and shows it to be optimal. Section 4.3 explores ways to reduce the dependence on dimension. Section 5 gives lower bounds for any ICL (not just transformers).

There is a sufficiently expressive representation class $\mathcal{F}$ (here we take DNNs) from which we can draw a $\phi^*$ such that linear attention on top of this solves regression well. In that sense, this paper generalizes the idea from linear attention for linear regression that the attention inverts the data-covariance.

**Strengths:**

The paper contains a general analysis of ICL error for regression problems. The comparison to non-parametric rates is interesting and possibly the right way to extend the so far mostly linear analysis in the literature to problems beyond linear regression.

**Weaknesses:**

This is general analysis of a very specific simplification of the transformer. The setting is that there are trainable layers before the attention layer, and then exactly one linear attention layer. Is there any hope to extend this analysis to more attention layers/ non-linear attention/ constant $N$? When $N$ is constant, the error is constant, which seems vacuous considering everything is bounded. Is that an artifact of analysis?

Is it possible to have a setting where we removed the DNN and force the features themselves to satisfy Assumption 3? Would that reduce to the linear regression setting of prior work? If so, it seems very important to justify how we can find a $\phi^*$ satisfying Assumption 3 for DNNs.

**Questions:**

A matter of notation: in assumption 1, why aren’t the basis functions the first N functions, rather than the functions from $\underline{N}$ to $\overline{N}$? Also, are the functions past $\overline{N}$ not spanned by the basis? Remark 2.1 offers some explanation. I think it would be helpful to write the main paper in the setting where $\psi$'s span the space and leave the generalization for the appendix.

What role does the sparsity $S$ play, why is it set to be $O(1)$? Isn't is unrealistic that $N$ should scale in $n$ for a real transformer (one of the key selling points about transformers are how the parameters dont scale in the context length)? Do the authors think this is due to the analysis or is it fundamental for attention?

It would be helpful to instantiate a target function class in the main paper, specify what $\alpha$ is, etc.

**Limitations:**

Yes

---

> ### Author Rebuttal · Authors · 2024-08-06
>
> Thank you for the detailed review and insightful questions! Our responses are as follows.
>
> * **It would be helpful to instantiate a target function class in the main paper, specify what $\alpha$ is, etc.**
>
> (We answer this question first to help in understanding our paper.) **Throughout Section 4, we consider concrete task classes including the Besov space, anisotropic Besov space, and piecewise $\gamma$-smooth function class.** Our main result for DNNs (Theorem 4.5) is stated when $\mathcal{F}^\circ$ is the unit ball in Besov space $B_{p,q}^\alpha(\mathcal{X})$, which is a very wide function class generalizing Holder and Sobolev spaces; please see Section 4.1 for details. Here $\alpha$ is the smoothness and $p,q$ are additional parameters controlling spatial inhomogeneity. These relate to the decay rate $s$ of Assumption 2 by $s=\alpha/d$; the high-level Assumptions 1-3 all follow from Assumption 4 in this setting.
>
> We also show the curse of dimensionality can be avoided when $\mathcal{F}^\circ$ is extended to the *anisotropic* Besov space in Section 4.3. Furthermore, for the multilayer Transformer setting (Section 4.5) the inputs $x$ are sequences and $\mathcal{F}^\circ$ is the piecewise $\gamma$-smooth class, which is a more realistic model for text by allowing the position of important tokens to depend on input.
>
> * **Is there any hope to extend this analysis to more attention layers/non-linear attention/constant $N$?**
>
> Please see Item 1 of the global response where we address this question in depth.
>
> * **When $N$ is constant, the error is constant, which seems vacuous.**
>
> While our bounds (e.g. Theorem 4.5) hold for all $N$, the risk dependency is not an artifact but a natural result of how our task class is set up. After pretraining our transformer has learned $N$ feature maps $\phi_1,\cdots,\phi_N$ and its output $f_\Theta(\mathbf{X},\mathbf{y},\cdot)$ is always contained in the span of these $N$ functions. However, the true test task $F_\beta^\circ$ is randomly chosen from an infinite-dimensional space, and so the $L^2$ regression error is fundamentally lower bounded by its **Kolmogorov width** -- the minimum approximation error by an $N$-dimensional linear subspace -- which scales as $N^{-\alpha/d}$ for the Besov space. Hence this error is unavoidable for *any* fixed-basis approximation scheme; an implication of our analysis is that Transformers attain this optimal rate (the first term in Theorem 4.5).
>
> Besides the approximation error, the pretraining and in-context generalization errors decrease as $O(1/T)$ and $O(1/n)$, which is still a useful result if one is concerned with improving generalization.
>
> * **Is it possible to have a setting where we removed the DNN?**
>
> Indeed, if we completely remove the DNN and set both the features and basis to be simply the coordinate mappings $x_1,\cdots,x_d$, this reduces to the class of linear maps in prior works. In a nutshell, our contribution is extending this to infinite-dimensional nonparametric task classes via learnable representations, enabling fine-grained sample complexity analysis and guaranteeing optimality of ICL with deep architectures.
>
> * **It seems very important to justify how we can find a $\phi$ satisfying Assumption 3 for DNNs.**
>
> That is exactly what Appendix C.1.1 (Verification of Assumptions) and Lemma 4.4 are for! We show that the abstract Assumptions 1,2,3 are all replaced by the single concrete Assumption 4 when the target class is set to the Besov space. For the multilayer Transformer setting (Section 4.5), Assumption 3 instead follows from Theorem D.2. We will make this point clearer in the paper.
>
> * **Why aren’t the basis functions the first N functions?**
>
> In fact, this is to address a **unique technical difficulty** that we newly uncover: the necessity of the inverse covariance matrix $\Psi_N^{-1}$ to approximate the attention matrix. Wavelet systems of Besov-type spaces are fundamentally co-dependent and form a multi-resolution scheme: there are $O(2^k)$ independent basis elements at each resolution $k\in\mathbb{N}$, which can always be refined as a linear combination of higher resolution wavelets. This makes $\Psi_N$ singular and affects various decay rates, so we cannot simply take the first $N$ elements. This was not a problem in prior attention-only works ($\text{Var}(x)$ was simply assumed to be positive definite) nor any existing minimax optimality works (which don't have a product structure). Nonetheless, we were still able to prove optimality by decomposing all wavelets to the finest resolution -- numbered from $\underline{N}$ to $\overline{N}$ -- via the refinement analysis in Appendix C.3, hence the additional steps to account for the aggregated coefficients $\bar{\beta}$. While this additional notation may cause confusion upon first reading, we have left it in since specializing to the wavelet system to obtain optimality is an important part of our work.
>
> * **What role does the sparsity $S$ play, why is it set to be $O(1)$?**
>
> The sparsity bound $S=O(1)$ is merely to clarify the minimum number of essential parameters and can be completely removed. Recall that a fully connected network has $S\sim LW^2$. The dominating term from the DNN class entropy (line 675) is $SLN\log W$, so that letting $S=O(\log N)$ (fully connected depth $L$ network) would only incur an additional log factor to yield $N(\log N)^2$. Since the $N^2\log N$ term from the attention matrix entropy dominates this, the overall bound remains unchanged.
>
> (continued in comment)

---

> ### Author Response · Authors · 2024-08-06
>
> (continued from rebuttal)
>
> * **Isn't is unrealistic that $N$ should scale in $n$ for a real transformer?**
>
> In practice we agree that the architecture should not depend on $n$, especially if prompts of any length are allowed. However since we assume **all** prompts (during both pretraining and ICL) are of fixed length $n$, it is reasonable to choose a more powerful architecture for larger $n$.
>
> Strictly speaking, $N$ does not need to scale in $n$ since our bounds hold for all $N,n,T$. But the bound itself is natural: as we mentioned, approximation error must decrease in $N$ due to the infinite dimensionality of the target class, and generalization error should of course decrease in $n,T$. For example if $N$ is considered fixed, the bound is interpreted as an excess risk of $O(1/n+1/T)$. However we want to obtain the *overall* sample complexity rate in $n$, which necessitates $N$ to scale in $n$. The rate itself is less important than the fact that it is the *best rate attainable by any estimator*, and the optimality result should be viewed more as guaranteeing tightness of the derived bounds (further justified by Section 5) than a prescription of how $N,T$ should scale.
>
> We also mention that this approach is standard in minimax analyses. For example, rate-optimal scaling for ordinary supervised regression with DNNs (Suzuki, 2019) also requires the width to scale as $n^{\frac{1}{2s+1}}$ even though the smoothness of the target is unknown.
>
> * **Rebuttal End**
>
> Finally, we have also obtained **improved lower bounds** which reinforce the message of our paper, and newly conducted **numerical experiments** justifying our assumptions and results. Please see Item 2 and Item 3 of the global response for details. We also humbly ask the reviewer to consider raising their score if some of their concerns were addressed.

---

> ### Author Response · Authors · 2024-08-11
>
> Thank you again for your time and effort in reviewing our paper! As the discussion period is ending soon, we are following up to see if our response addressed the reviewer's questions. If so, we hope that they would be willing to increase their score. If not, we would be happy to discuss any remaining concerns.

---

> ### Comment · Reviewer_1n5m · 2024-08-13
>
> Thank you for your response. I will raise my score to a 5.

---

### Official Review · Reviewer_KRWK · 2024-07-29

**Soundness:** 4
**Presentation:** 4
**Contribution:** 2
**Rating:** 6
**Confidence:** 3

**Summary:**

This paper studies in-context nonparametric learning using transformers. In the setting used in the main result of the paper, the transformer is trained on a dataset consisting of multiple sequences/tasks. For each task, the target function $F_\beta$ is drawn from the span of a certain countable set of functions - in the setting used for the main result of the paper, the target function is drawn from a Besov space spanned by a B-spline wavelet basis. The task then consists of multiple in-context pairs $(x_k, y_k)$, where $y_k$ is $F_\beta(x_k) + \xi$, where $\xi$ is random noise. Certain technical assumptions are made on the basis functions $\psi_j$ and the coefficients $\beta_j$: the $\psi_j$ are assumed to have a property that is similar to linear independence/orthonormality, while the $\beta_j$ are assumed to decay at a certain rate.

The model in this paper's setting consists of a feature map $\phi$ applied to all of the $x_k$, followed by a linear attention layer applied to the $\phi(x_k)$ and the $y_k$. The feature map $\phi$ is assumed to be expressive enough to approximate the $\psi_j$. In the main result of this paper, $\phi$ is chosen from the class of deep neural networks (DNNs) with a logarithmic number of layers (logarithmic in the size of the feature map), and $O(1)$ width and $O(1)$ entries per layer. The class of feature maps $\phi$ is denoted as $\mathcal{F}_N$, and the overall model class is denoted as $\mathcal{T}_N$.

The main result of the paper, Theorem 4.5, is that with a sufficient number of tasks during pretraining, in-context learning with transformers will achieve the optimal minimax risk. This is shown as follows. Theorem 3.1 first gives a bound on the expected test loss in terms of the covering number of the model class $\mathcal{T}_N$, and the minimum test loss achievable by some member of $\mathcal{T}_N$. Next, the paper gives a particular construction for a set of parameters which achieves low test loss. The weight matrix for the linear attention layer is chosen similarly to the optimal weight matrix given in Zhang, et al. (2023) and nonlinear feature map outputs a subset of the basis functions.

Additional results are also given. This paper shows how the curse of dimensionality can be avoided when the target function is drawn from an anisotropic Besov space. Also, under certain assumptions, even if the target function is drawn from a Besov space with smoothness $\tau < \alpha$, it is possible to achieve the same minimax rate as in the case where the target function is drawn from a Besov space with smoothness $\alpha$. Finally, the paper gives similar results in the setting where the tokens are themselves sequences of unbounded length, and gives minimax lower bounds that match the upper bounds.

**Strengths:**

- This work seems to be the first to study in-context learning using transformers in Besov spaces (prior work studied settings such as regression with DNNs).
- The paper is very well-written.

**Weaknesses:**

I think the main weakness is that the techniques used to show Theorem 4.5 are somewhat standard.
- Theorem 3.1, which shows that the empirical risk minimizer achieves good test loss, seems to be using relatively standard arguments from previous work, based on covering numbers.
- The construction used to show that low approximation error can be achieved also seems somewhat straightforward. The weight matrix for the attention layer is chosen similarly to Zhang, et al. (2023), and the feature map outputs a subset of the basis vectors.

**Questions:**

- In Lemma 4.4, how important is the assumption that the width/sparsity of the deep neural network is $O(1)$? Would the guarantee obtained in Theorem 4.5 change if the deep neural network is allowed to have more nonzero entries?
- Could the setting in this paper be considered similar to the linear regression setting studied by previous works such as Zhang, et al. (2023), with the difference being that the linear regression weight vector is replaced by the vector $\beta$ of coefficients for the basis functions?

**Limitations:**

Yes.

---

> ### Author Rebuttal · Authors · 2024-08-06
>
> Thank you for your through review of our work! We hope our responses can help clarify some important points.
>
> **Weakness 1.**
>
> Theorem 3.1 is indeed a standard result, as we have indicated it is a straightforward adaptation of Lemma 4 of Schmidt-Hieber (2020). While it is only the first step of the proof of the main results, we included the statement to explicitly show how the in-context and pretraining generalization gaps arise from different elements of the risk: the former manifests when evaluating the approximation error w.r.t. $n$, while the latter is the usual entropy term.
>
> **Weakness 2.**
>
> The attention matrix construction can be seen as an extension of Zhang (2023), however we urge the reviewer to take the following points into consideration.
>
> * The optimal construction is completely determined for a linear attention layer since the L2 linear regression loss is convex; even the construction in Zhang et al. (2023) is stated to be a generalization of Oswald et al. (2023). So while we agree the construction is not surprising, it is an important step allowing us to reduce to the analysis of the DNN module which facilitates the main risk analysis. Conversely, Zhang and Oswald do not consider the MLP at all and only study the attention matrix.
> * The given form (line 174) is only an example construction to upper bound the empirical loss. Unlike Zhang, we do **not** claim the ERM or optimization dynamics must result in such a simplified form, which may not be true in a deep Transformer setting.
> * The idea of approximating the target function with a well-chosen basis is of course fundamental to all functional analysis and is not unique to our approach. Indeed, our goal is to **establish a strong connection between ICL and ordinary supervised regression** by isolating the pretraining generalization gap. We dive deeper into this idea by pointing out a new limitation of ICL stemming from non-adaptive function approximation (Remark 4.6).
> * Moreover, a **unique technical difficulty** that we newly uncover and address is the necessity of the inverse covariance matrix $\Psi_N^{-1}$ to approximate the attention matrix. Wavelet systems of Besov-type spaces are fundamentally co-dependent and form a multi-resolution scheme which makes $\Psi_N$ singular and affects various decay rates. This was not a problem in Zhang's work ($\Lambda$ was simply assumed to be positive definite) nor any existing minimax optimality works (which don't have a product structure). Nonetheless, we were still able to prove optimality by decomposing all splines to the finest resolution and performing the analysis strictly at that scale, hence the additional steps to account for the aggregated coefficients $\bar{\beta}$.
> * We have added experiments justifying the use of our simplified transformer model; please see Item 3 of the global response.
>
> We also implore the reviewer to judge our result not based on the perceived difficulty level of the proof, but its implications towards understanding the effectiveness of in-context learning.
>
> **Question 1.**
>
> The assumption can be substantially relaxed; the stated form is mostly to clarify the minimum number of essential parameters. Recall that a fully connected network has $S\sim LW^2$. The dominating term from the DNN entropy (line 675) is $SLN\log W$, so that letting $S=O(\log N)$ (fully connected depth $L$ network) would only incur an additional log factor to yield $N(\log N)^2$. Since the $N^2\log N$ term from the attention matrix entropy dominates this, the overall bound remains unchanged. The same holds even if width is relaxed to $W=O(N)$, and furthermore even if $S=O(N)$ in this case.
>
> However if $W=O(N)$ *and* the network is fully connected, that is $S=O(N^2\log N)$, the pretraining gap will worsen to around $N^3/T$. This reflects the fact that wavelets are 'easier' to approximate with DNNs than Besov functions are with wavelets.
>
> **Question 2.**
>
> That is indeed the setup we were aiming for, however please consider our response to Weakness 2 as well as the following points.
>
> * Linear attention has been studied in many other papers including Zhang, Oswald, Mahankali, and Ahn's works (cited in the paper), all yielding similar 'optimal matrix' constructions. As mentioned before, we do not claim originality in this regard. However, our focus is on reduction to the **representational capability of the MLP layers**. Except constructive works such as Bai et al. (2023), The NN+attention setup had only been studied before very recently by Kim and Suzuki (2024), and they only considered a shallow MLP.
> * We also find it illuminating to consider our setup as a deep Transformer with all but the last attention layer removed. Since this simplified model is already in-context sample optimal, we can expect a full Transformer to be the same. Moreover, we indicate how to extend our results to nonlinear attention or multiple attention layers in Item 1 of the global response.
>
> We have also obtained **improved lower bounds** which reinforce the message of our paper, and newly conducted **numerical experiments** justifying our assumptions and results. Please see Item 2 and Item 3 of the global response for details. We also humbly ask the reviewer to consider raising their score if some of their concerns were addressed.

---

> > ### Author Response · Authors · 2024-08-11
> >
> > Thank you again for your time and effort in reviewing our paper! As the discussion period is ending soon, we are following up to see if our response addressed the reviewer's questions. If so, we hope that they would be willing to increase their score. If not, we would be happy to discuss any remaining concerns.

---

> > ### Comment · Reviewer_KRWK · 2024-08-13
> >
> > Thank you for the detailed reply. I am not familiar with the literature on wavelet systems, but since the covariance matrix may not be invertible, this suggests that the analysis in this work is more than a direct extension of Zhang, et al. The lower bound in the coarser basis setting is also an interesting takeaway. I will increase my score to 6.
> >
> > Could you please clarify the following:
> > - For your main setting, you mention in the global response that your upper bound is jointly optimal in $n, T$. In what sense is it optimal in $T$?
> > - How do you select $N$ in Theorem 4.5 and Corollary 4.9? Would it be possible to optimize the bound in Corollary 4.9 further by selecting a different $N$?

---

> > > ### Author Response · Authors · 2024-08-13
> > >
> > > We are very grateful to the reviewer for re-evaluating our contributions and raising their score! Here are some further clarifications:
> > >
> > > * Wavelet systems form a hierarchy ordered by resolution, and wavelets with lower resolution can always be written as a combination of those with higher resolution (forming the basis of *multiresolution analysis* theory). Hence the covariance matrix is indeed never invertible, necessitating our new 'aggregation' techniques.
> > > * ICL can be viewed as a meta-learner which takes as input not just $n$ samples from the current task but also $(n+1)T$ samples from related tasks. Now also consider $T$ as a variable. It could be the case that a well-designed meta-estimator (possibly not constrained to be fixed at inference time) can achieve a rate faster than $n^{-\frac{2\alpha}{2\alpha+d}}$ by taking all these samples as input, as the ordinary minimax rate only applies to estimators which only learn from $n$ ground-truth samples. Our lower bound shows that this is impossible: *any* estimator that takes the $n$ ground-truth \& $(n+1)T$ related samples as input is still lower bounded by $n^{-\frac{2\alpha}{2\alpha+d}}$. Since our upper bound matches this when $T$ is sufficiently large, we conclude that ICL is **optimal in both** $n,T$. Conversely, the new lower bound $n^{-\frac{2\alpha}{2\alpha+d}} + (nT)^{-\frac{2\tau}{2\tau+d}}$ in the coarser basis setting allows us to conclude that ICL (or any other meta-learner) is **suboptimal in both** $n,T$ when $T<n^{\frac{(\alpha-\tau)d}{\tau(2\alpha+d)}}$, supplementing our upper bound in Corollary 4.9.
> > > * The selection of $N$ is standard rate analysis, except that we also have to consider the effect of $T$ through the pretraining gap. Let's look at Theorem 4.5. First assume $T\gtrsim nN$ so that the 3rd term can be ignored compared to the 2nd term. Then we want to choose $N$ such that $N^{-2\alpha/d}+\frac{N\log N}{n}$ is minimized, which can be found by balancing $N^{-2\alpha/d}\asymp\frac{N}{n}$ and thus $N\asymp n^\frac{d}{2\alpha+d}$ (ignoring log terms). However if $T<nN$ then the 3rd term dominates and $N^{-2\alpha/d}+\frac{N^2\log N}{T}$ is minimized when $N\asymp T^\frac{d}{2\alpha+2d}$, so the overall risk is now bounded suboptimally as $T^{-\frac{\alpha}{\alpha+d}}$.
> > > * Corollary 4.9 is similarly tuned and the bound cannot be optimized further as we also proved it is optimal in both $n,T$ when $T$ is large. Again there is some suboptimal scaling when $T$ is small, but -- as we indicate in our global response -- for this regime it is better to look at the information-theoretic lower bound which gives a rigorous proof of suboptimality.

---

### Author Rebuttal · Authors · 2024-08-06

## 1. Extending to Multiple/Nonlinear Attention

As reviewers have noted, our setup builds on previous works which establish ICL of a single attention layer as linear regression in order to extend to complexity analysis of nonparametric regression. ICL of multiple/nonlinear attention layers can be fundamentally different, possibly exhibiting complex behavior such as induction heads or data clustering which are not yet well understood. Nonetheless, our techniques may be partially applied to these settings as follows.

**Multiple attention layers:** We consider a setup with multiple attention layers in Section 4.5, where the preceding Transformer module performs dynamical feature extraction and the final attention layer performs regression based on the learned representations. This aligns with the empirical observation by Guo et al. (2024) that lower layers of Transformers learn representations of the input and upper layers perform linear ICL.

We also find it illuminating to consider our setup as a deep Transformer with all but the last attention layer removed, so that the feedforward layers combine into a deep MLP. Since this simplified model already achieves sample optimal ICL, we can expect a full Transformer to achieve the same.

**Nonlinear attention:** The simplest approach is to linearize the kernel, for example by taking a small rescaling factor $\gamma$ and approximating $e^{\gamma x^\top\Gamma x'}\approx 1+\gamma x^\top\Gamma x'$. Ignoring the normalizing factor, this shows that nonlinear attention has at least as much expressivity as linear attention. Moreover, linear attention has been empirically shown to capture many other prominent aspects of softmax attention (Ahn et al. 2024).

However ideally we want to characterize the full expressive capability of nonlinear attention. To this end, we can take the following kernel-based approach. By introducing an additional kernel transformation $K(x,x')$ such as the RBF kernel $e^{x^\top\Gamma x'}e^{-\lVert x\rVert_\Gamma^2/2}e^{-\lVert x'\rVert_\Gamma^2/2}$ for (rescaled) softmax attention, we can use Mercer's theorem to decompose $K(x,x') = \sum_{j=0}^\infty \lambda_j e_j(x)e_j(x')^\top$ and approximate $K$ using the top few eigenfunctions $e_j$. For example for the RBF kernel, $e_j$ is an appropriately scaled Gaussian density times the $j$th Hermite polynomial.

Then the main problem is to evaluate the approximation capability of the maps $e_j\circ\phi$ composed with the DNN module. This can be done assuming some compatibility conditions between the kernel eigenfunctions and target class basis functions. (This is particularly straightforward if e.g. the kernel is such that $\lambda_j$ is learnable, so that the terms can be considered separately in $j$.)

## 2. Take-home Message

Together with improved lower bounds, we restate our take-home message (besides in-context optimality in $n$) as follows. We believe this novel understanding will be helpful to both theoreticians and practitioners.

* (In the Besov space setting) The obtained upper bound $n^{-\frac{2\alpha}{2\alpha+d}}$ when $T\gtrsim nN$ which is minimax optimal in $n$, has also been shown to be jointly optimal in $n,T$. Hence **ICL is provably optimal in the large $T$ regime.**
* (In the coarser basis setting) We obtained an improved lower bound $n^{-\frac{2\alpha}{2\alpha+d}} + (nT)^{-\frac{2\tau}{2\tau+d}}$ using the method in Appendix E.3. If $T=O(1)$, this gives the slower $\tau$ rate, while if $T$ is larger this gives the faster $\alpha$ rate. This aligns with Corollary 4.9 (which only gave an upper bound), hence **ICL is provably suboptimal in the small $T$ regime.**

When combined, these results are stronger than optimality in only $n$. They also align with experimental observations of task diversity threshold (Raventos 2023), rigorously supporting the importance of varied tasks over increasing prompt length.

## 3. Numerical Experiments

Following comments by reviewers on the lack of experiments connecting our analysis to practical transformers, we have conducted new simulations justifying our simplified model setup and the assumption of empirical risk minimization. Figures can be found in the attached PDF. We implement and compare the following toy models:

* Linear transformer: the model studied in our paper, where a DNN (2 hidden layers) feeds into a simplified linear attention layer
* Softmax transformer: the same model as (a) but with softmax attention
* Full transformer: an ordinary transformer with 2 stacked encoder layers

Architectures are compared in Figure 1. The number of feedforward layers, width of hidden layers, learning rate etc. are set to equal for a fair comparison.

* Figure 2 shows training (solid lines) and test loss curves (dashed lines) during pretraining. All 3 architectures exhibit similar behavior and converge to near zero training loss, **justifying the use of our simplified model** and also supporting the assumption that the **empirical risk can be minimized**.
* Moreover, Figure 3 shows the converged losses over a wide range of $N,n,T$ values (median over 5 runs). We verify that increasing $N,n$ leads to decreasing train and test error, corresponding to the approximation error of Theorem 3.1. We also observe that increasing $T$ tends to improve the pretraining generalization gap up to a threshold, confirming our theoretical analyses. Again, this behavior is consistent across the 3 architectures.
* Although the dimensions and architectures are relatively small due to limited time and compute, we plan to scale up our experiments and add them to the final version of the paper.

---

### Decision · Program_Chairs · 2024-09-25

**Decision:**

Accept (poster)

**Comment:**

This paper develops generalization bounds for 1-layer transformers for nonparametric regression tasks.  The authors show that one can achieve optimal rates through the forward pass of a simple 1-layer linear transformer provided the inputs are "tokenized" correctly via their feature maps.  I think this work provides a nice contribution to our understanding of the power of the forward pass of transformers to implement complex algorithms and the statistical complexity of a natural task.  I recommend acceptance.